# Deep Learning Methods for Proximal Inference via Maximum Moment Restriction

**Benjamin Kompa**[*]
Department of Biomedical Informatics
Harvard Medical School
benjamin_kompa@hms.harvard.edu

**David R. Bellamy**[*]
Department of Epidemiology, CAUSALab
Harvard School of Public Health
david_bellamy@g.harvard.edu

**Thomas Kolokotrones**
Department of Epidemiology
Harvard School of Public Health
thomas_kolokotrones@hms.harvard.edu

**James M. Robins**
Department of Epidemiology, CAUSALab
Harvard School of Public Health
robins@hsph.harvard.edu

**Andrew L. Beam**
Department of Epidemiology, CAUSALab
Harvard School of Public Health
andrew_beam@hms.harvard.edu

## Abstract

The No Unmeasured Confounding Assumption is widely used to identify causal effects in observational studies. Recent work on proximal inference has provided alternative identification results that succeed even in the presence of unobserved confounders, provided that one has measured a sufficiently rich set of *proxy variables*, satisfying specific structural conditions. However, proximal inference requires solving an ill-posed integral equation. Previous approaches have used a variety of machine learning techniques to estimate a solution to this integral equation, commonly referred to as the *bridge function*. However, prior work has often been limited by relying on pre-specified kernel functions, which are not data adaptive and struggle to scale to large datasets. In this work, we introduce a flexible and scalable method based on a deep neural network to estimate causal effects in the presence of unmeasured confounding using proximal inference. Our method achieves state of the art performance on two well-established proximal inference benchmarks. Finally, we provide theoretical consistency guarantees for our method.

## 1 Introduction

Causal inference is concerned with estimating the effect of a treatment $A$ on an outcome $Y$ from either observational data or the results of a randomized experiment. An estimand of primary importance is the *average causal effect* (ACE), which is the expected difference in $Y$ caused by changing the treatment from value $a$ to $a'$ for each unit in the study population, and is defined as a contrast between the expected value of the potential outcomes at the two levels of the treatment: $\mathbb{E}[Y^{a'}] - \mathbb{E}[Y^a]$. However, in observational settings, the ACE is rarely equal to the observed difference in conditional means, $\mathbb{E}[Y|A = a'] - \mathbb{E}[Y|A = a]$ due to confounding. In an attempt to eliminate the influence of confounding, investigators measure putative confounders $X$ and subsequently make adjustments for $X$ in their analyses.

---

[*]Denotes equal contribution

36th Conference on Neural Information Processing Systems (NeurIPS 2022).

Given $X$, common approaches, such as standardization and inverse probability weighting (Hernán and Robins [1]), obtain valid estimates of the ACE given that the following assumptions hold: i) Positivity: $Pr[A = a|X = x] > 0$ for all $x$ in the population, ii) Consistency: $Y^a = Y$ for all individuals with $A = a$, iii) No unmeasured confounding which results in conditional exchangeability: $Y^a \perp\!\!\!\perp A|X$., and iv) No model misspecification.

The assumptions are typically unverifiable for continuous data. While model misspecification is likely in all real-world scenarios, flexible models and doubly robust estimators have been developed to mitigate the effect of this assumption[2]. Therefore, the assumption of conditional exchangeability, or equivalently, the No Unmeasured Confounding Assumption (NUCA), is the defining characteristic of this broad set of approaches to causal effect estimation (Hernán and Robins [3]). However, in many settings, it is unrealistic to assume that we are able to measure a sufficient set of confounders for $A$ and $Y$ such that conditional exchangeability holds.

*Proximal inference* is a recently introduced framework that allows for the identification of causal effects even in the presence of unmeasured confounders [4, 5]. Proximal inference requires categorizing the measured covariates into three groups: treatment-inducing proxy variables $Z$, outcome-inducing proxy variables $W$, and "backdoor" variables $X$ that affect both $A$ and $Y$ (i.e. typical confounders). See Figure 1 for an example of a directed acyclic graph (DAG) that admits identification under the assumptions of proximal inference. The proxy sets $W$ and $Z$ must contain sufficient information about the remaining unobserved confounders $U$, a condition that can be formalized by completeness assumptions. Under these and several other conditions, one can estimate average potential outcomes from data even in the presence of unmeasured confounding. Proximal inference has potential applications in medical settings, where a natural question is the effect of a treatment on an outcome in the presence of unmeasured confounding. Before applying proximal inference to real world problems, more validation is required before they can be used safely to inform medical decision-making.

Existing methods for proximal inference can be divided into two categories: two-stage regression procedures and methods that impose a maximum moment restriction (MMR). In two-stage regression procedures, the first stage aims to predict outcome-inducing proxy variables $W$ as a function of $A$, $X$, and $Z$. Then, the second stage regression estimates outcomes $Y$ as a function of the predicted $\hat{W}$ and the treatment $A$, and measured confounders $X$. Tchetgen Tchetgen et al. [5] introduced the first estimation technique for proximal inference which was a two-stage procedure that used a model based on ordinary least squares regression. Mastouri et al. [6] extended this framework by replacing simple linear regression with kernel ridge regression. Xu et al. [7] increased feature flexibility further by incorporating neural networks as feature maps instead of kernels.

In contrast, MMR methods are single-stage procedures to estimate average potential outcomes. Muandet et al. [8] introduced MMR for reproducing kernel Hilbert spaces (RKHS). MMR critically relies on the optimization of a V-statistic or U-statistic for learning a function needed to calculate the ACE. Zhang et al. [9] used an MMR method to obtain point identification of the ACE in the instrumental variable (IV) setting and incorporated neural networks into their method trained with the V-statistic as a loss function and optimized using stochastic gradient descent. Mastouri et al. [6] demonstrated that the MMR framework with kernel functions can be used for proximal inference as well as IV regression.

In this work, we introduce a new method, *Neural Maximum Moment Restriction* (NMMR) which is a flexible neural network approach that is trained to minimize a loss function derived from either a U-statistic or V-statistic to satisfy MMR in the proximal setting. The method introduced in this work makes several novel contributions to the proximal inference literature:

- We introduce a new, single stage method based on neural networks for estimating potential outcomes and the ACE in the presence of unmeasured confounding.
- We provide new theoretical consistency guarantees for our method.
- We demonstrate state-of-the-art (SOTA) performance on two well-established proximal inference benchmark tasks.
- We show for the first time how to incorporate domain-specific inductive biases using a convolutional model on a proximal inference task that uses images.
- We provide the first unbiased estimate of the MMR risk function using the U-statistic rather than V-statistic in the proximal setting.

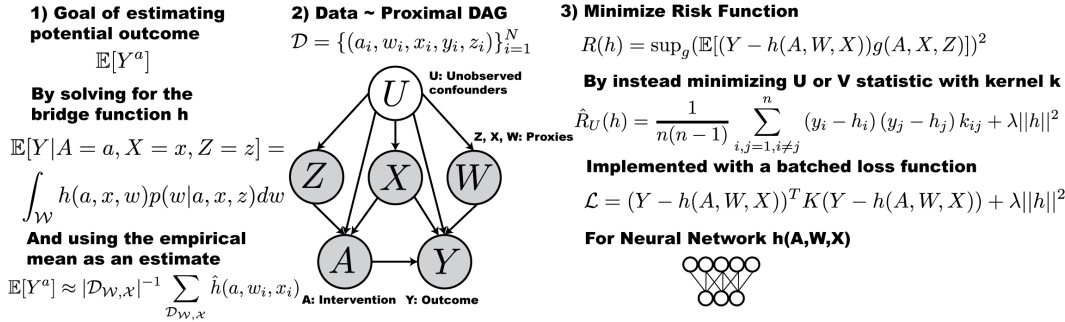

Figure 1: A summary of NMMR. Our method estimates the bridge function $h$, which can be used to compute the average potential outcome $\mathbb{E}[Y^a]$. We rely on structural assumptions for the causal DAG generating the data. NMMR uses a U or V statistic to train a neural network that solves a risk function that reflects a maximum moment restriction function.

## 2 Background: Proximal Inference

In this manuscript, capital, caligraphic, and lowercase letters (e.g. $A$, $\mathcal{A}$, and $a$) denote random variables and their corresponding ranges and realizations, respectively. Estimates of random variables and functions will be indicated by hats, e.g. $\hat{Y}$ and $\hat{h}$ are estimates of $Y$ and $h$, respectively.

Our goal is to estimate, for each level of treatment, $a$, the expected potential outcome $\mathbb{E}[Y^a]$. Without loss of generality, we refer to $A$ as a treatment, though it could refer to any (possibly continuous) intervention. Proximal inference allows us to do this, in the presence of unobserved confounders, $U$, provided that we have a sufficiently rich set of proxies, $(W, Z)$, that obey certain structural assumptions. We may also include observed confounders, $X$. We also require the following:

**Assumption 1.** *Given* $(A, U, W, X, Y, Z)$, $Y \perp\!\!\!\perp Z | A, U, X$ *and* $W \perp\!\!\!\perp (A, Z) | U, X$.

Figure 1 provides an example of a DAG that satisfies these assumptions. The following completeness conditions formalize the notion that the proxies are "sufficiently rich":

**Assumption 2.** *For all* $f \in L^2$ *and all* $a \in \mathcal{A}, x \in \mathcal{X}$, $\mathbb{E}[f(U)|A = a, X = x, Z = z] = 0$ *for all* $z \in \mathcal{Z}$ *if and only if* $f(U) = 0$ *almost surely.*

**Assumption 3.** *For all* $f \in L^2$ *and all* $a \in \mathcal{A}, x \in \mathcal{X}$, $\mathbb{E}[f(Z)|A = a, W = w, X = x] = 0$ *for all* $w \in \mathcal{W}$ *if and only if* $f(Z) = 0$ *almost surely.*

We will use two other assumptions at various points in the paper. The first guarantees the uniqueness of the bridge function, while the second ensures the risk function does not have false zeros.

**Assumption 4.** $\mathbb{E}\left[f(A, W, X)|A, X, Z\right] = 0$ $\mathrm{P}_{A,X,Z}$*-almost surely if and only if* $f(A, W, X) = 0$ $\mathrm{P}_{A,W,X}$*-almost surely.*

**Assumption 5.** $k : (\mathcal{A} \times \mathcal{X} \times \mathcal{Z})^2 \to \mathbb{R}$ *is continuous, bounded, and Integrally Strictly Positive Definite (ISPD), so that* $\int f(\xi) k(\xi, \xi') f(\xi') d\xi d\xi' > 0$ *if and only if* $f \neq 0$ $\mathrm{P}_{A,Z,X}$*-almost surely.*

Assumptions 1-3 together with several regularity assumptions (see assumptions (v)-(vii) in [4]) ensure that there exists a function $h$ such that:

$$\mathbb{E}[Y|A = a, X = x, Z = z] = \int_{\mathcal{W}} h(a, w, x)p(w|a, x, z)dw \tag{1}$$

Equation 1 is a Fredholm integral equation of the first kind; its solution, $h$, is often called the "bridge function." Theorem 1 of Miao et al. [4] shows that the expected potential outcomes are given by:

$$\mathbb{E}[Y^a] = \int_{\mathcal{W},\mathcal{X}} h(a, w, x)p(w, x)dwdx = \mathbb{E}_{W,X}[h(a, W, X)] \tag{2}$$

We can obtain unbiased estimates of the expected potential outcomes, $\mathbb{E}[Y^a]$, by splitting the sample, using the first part of the data to estimate the bridge function $h$, by some $\hat{h}$, and using the

second part of the data to compute the empirical mean of $\hat{h}$ with $a$ fixed to the value of interest, $\hat{\mathbb{E}}[Y^a] = \frac{1}{M} \sum_{i=1}^{M} \hat{h}(a, w_i, x_i)$. From these, we can obtain other quantities of interest, like the ACE.

In what follows, all norms will be $L^2$ with respect to the relevant probability measure, unless otherwise noted. If necessary, we will explicitly denote the norm of $f$ with respect to a probability measure $\mathcal{P}_{U,V,\dots}$ by $\|f\|_{\mathcal{P}_{U,V,\dots}}$ or $\|f\|_{2,\mathcal{P}_{U,V,\dots}}$.

## 3 Related Work

Kuroki and Pearl [10] first established identification of a causal effect in the setting of unobserved confounders by leveraging noisy proxy variables, $W$, to "recover" the distribution of $U$, potentially using external datasets to estimate $p(w|u)$. Tchetgen-Tchetgen and colleagues extended these results to allow for identification without recovery of $U$ in Miao et al. [4] and Tchetgen Tchetgen et al. [5], also providing a 2-Stage Least Squares (2SLS) method to identify and estimate causal effects under the assumption that the bridge function is linear. Cui et al. [11] introduced a bridge function for Inverse Probability Weighting (IPW), which enabled IPW and Doubly Robust (DR) proximal estimators, for which they presented influence functions under binary treatment. This was extended in Ghassami et al. [12] and further explored by Kallus et al. [13], who provided alternative identification assumptions as well as results for general treatments. The latter two works also consider the use of adversarial methods for estimation, which were previously utilized by Lewis and Syrgkanis [14] and Bennett et al. [15] in the Instrumental Variable (IV) setting and Dikkala et al. [16] in conditional moment models.

Other early investigators include Deaner [17], who developed machine learning techniques for proximal inference introducing a method based on a two-stage penalized sieve distance minimization. Several later works similarly employed two-stage regressions with increasingly flexible basis functions to estimate potential outcomes. Mastouri et al. [6] developed a two-stage kernel ridge regression (Kernel Proxy Variables "KPV") to estimate the bridge function $h$, allowing more flexibility than the linear basis of Tchetgen Tchetgen et al. [5]. Xu et al. [7] further improved upon this with an adaptive basis derived from neural networks. Their two stage regression method, Deep Feature Proxy Variables (DFPV), established the previous SOTA performance on the proximal benchmark tasks that we consider in our work. Singh and colleagues also considered two stage kernel models in the IV setting [18] and RKHS techniques for proximal inference [19].

An alternative approach based on maximum moment restriction (MMR) uses single-stage estimators of the bridge function. MMR-based methods were established in Muandet et al. [8] as a way to enforce conditional moment restrictions [20]. Zhang et al. [9] introduced the MMR framework to the IV setting, which can be considered a subset of proximal inference without outcome-inducing proxies $W$ and with additional exclusion restrictions [5]. There are now several machine learning methods that can be applied in the IV setting [21, 14, 18, 16, 22]

Of note, Zhang et al. [9] introduced MMR-IV, which is related to work by Lewis and Syrgkanis [14] and Dikkala et al. [16]. MMR-IV involves optimizing a family of risk functions based on U or V-statistics [23]. However, Zhang et al. [9] only considered IV, rather than proximal, inference and only optimized neural networks by a loss that corresponds to the V-statistic. The V-statistic provides a biased estimate [23] of its corresponding risk function, such as $R(h)$ in Equation 3.

Finally, Mastouri et al. [6] introduced an MMR-based method for proximal inference called Proximal Maximum Moment Restriction (PMMR). PMMR extends the MMR framework to the proximal setting through the use of kernel functions and also optimizes Equation 3 via a V-statistic. For a comparison of our model to PMMR and MMR-IV, see Table 1.

## 4 Our Method: Neural Maximum Moment Restriction (NMMR)

In this work we propose *Neural Maximum Moment Restriction* (NMMR): a method to estimate expected potential outcomes $\mathbb{E}[Y^a]$ in the presence of unmeasured confounding. We use deep neural networks due to their flexibility, scalability, and adaptability to diverse data types (e.g. using convolutions for images). By rewriting maximum moment restrictions [8] as U and V-statistics [6, 9], we show how a single-stage neural network procedure can be used to estimate the bridge function.

Following Muandet et al. [8] and Zhang et al. [9], we rewrite the integral equation (1) as a *conditional moment restriction*: $\mathbb{E}[Y - h(A, W, X)|A, X, Z] = 0$. Then, for any measurable $g : \mathcal{A} \times \mathcal{X} \times \mathcal{Z} \to \mathbb{R}$, $\mathbb{E}[(Y - h(A, W, X))g(A, X, Z)|A, X, Z] = 0$, and, thus, $\mathbb{E}[(Y - h(A, W, X))g(A, X, Z)] = 0$, so we can use unconditional expectations instead of conditional ones. This is the basis of the MMR framework of Muandet et al. [8]. Since this yields an infinite number of moment restrictions we employ a minimax strategy to estimate $h$ by minimizing the risk $R(h)$ for the worst-case value of $g$:

$$R(h) = \sup_{\|g\| \leq 1} (\mathbb{E}[(Y - h(A, W, X))g(A, X, Z)])^2 \tag{3}$$

Following Zhang et al. [9]'s work in the IV setting, Mastouri et al. [6] (Lemma 2) showed that, if $g$ is an element of an RKHS, $R(h)$ can be rewritten in the form

$$R_k(h) = \mathbb{E}[(Y - h(A, W, X))(Y' - h(A', W', X'))k((A, X, Z), (A', X', Z'))]$$

where $(A', W', X', Y', Z')$ are independent copies of the random variables $(A, W, X, Y, Z)$ and $k : (\mathcal{A} \times \mathcal{Z} \times \mathcal{X})^2 \to \mathbb{R}$ is a continuous, bounded, and Integrally Strictly Positive Definite (ISPD) kernel. Then, if $h$ satisfies $R_k(h) = 0$, $\mathbb{E}[Y - h(A, W, X)|A, X, Z] = 0$ $P_{A,X,Z}$-almost surely. Thus, if we can find a neural network $h$ that satisfies $R_k(h) = 0$, we will have obtained a $P_{A,X,Z}$-almost sure solution to Equation 1 and can compute any expected potential outcome by using Equation 2.

The empirical risk $\hat{R}_{k,n}$ given data $\mathcal{D} = \{(a_i, w_i, x_i, y_i, z_i)\}_{i=1}^{N}$ can be written as either a U or V-statistic, respectively [23]:

$$\hat{R}_{k,U,n}(h) = \frac{1}{n(n-1)} \sum_{i,j=1, i \neq j}^{n} (y_i - h_i)(y_j - h_j) k_{ij}$$

$$\hat{R}_{k,V,n}(h) = \frac{1}{n^2} \sum_{i,j=1}^{n} (y_i - h_i)(y_j - h_j) k_{ij}$$

where $h_i = h(a_i, w_i, x_i)$ and $k_{ij} = k((a_i, z_i, x_i), (a_j, z_j, x_j))$. $\hat{R}_{k,U,n}(h)$ is the minimum variance unbiased estimator of $R_k(h)$ [23], while $\hat{R}_{k,V,n}(h)$ is a biased estimator of $R_k(h)$. In order to prevent overfitting, we add an additional penalty to our risk function $\Lambda : \mathcal{H} \times \Theta_{\mathcal{H}} \to \mathbb{R}_+$, which is a function of $h$ as well as, possibly, its parameters, $\theta_h$ (e.g. network weights). Specifically, we take $\Lambda$ to be an $L^2$ penalty on the weights so $\Lambda[h, \theta_h] = \sum_i \theta_{h,i}^2$. We then denote the penalized risk functions by $\hat{R}_{k,U,\lambda,n}(h) = \hat{R}_{k,U,n}(h) + \lambda \Lambda[h, \theta_h]$ and $\hat{R}_{k,V,\lambda,n}(h) = \hat{R}_{k,V,n}(h) + \lambda \Lambda[h, \theta_h]$, respectively. In practice, $\hat{R}_{k,U,\lambda,n}(h)$ is slightly biased, but, in simulations, is much less biased than even the unpenalized $\hat{R}_{k,V,n}(h)$. Previous work either did not consider the U-statistic [6], or did not utilize the U-statistic [9]. In our work, we introduce two variants of our method, NMMR-U and NMMR-V, where the former is optimized with a U-statistic and the latter a V-statistic. We train the neural networks in both variants with the regularized loss function:

$$\mathcal{L} = (Y - h(A, W, X))^t K(Y - h(A, W, X)) + \lambda \Lambda[h, \theta_h]$$

where $(Y - h(A, W, X))$ is a vector of residuals from the neural network's predictions and $K$ is a kernel matrix with entries $k_{ij}$. We choose $k$ to be an RBF kernel (see Appendix B). If $\mathcal{L}$ represents a V-statistic, we include the main diagonal elements of $K$, while if $\mathcal{L}$ represents a U-statistic, we set the main diagonal to be 0. Once we've obtained an optimal neural network $\hat{h}$, we can compute an estimate of the expected potential outcome with data from a held-out dataset with $M$ data points

$$D_{\mathcal{W},\mathcal{X}} = \{(w_i, x_i)\}_{i=1}^{M}, \quad \mathbb{E}[\hat{Y}^a] = \frac{1}{M} \sum_{i=1}^{M} \hat{h}(a, w_i, x_i)$$

In contrast to PMMR [6], which uses kernels as feature maps for proxy and treatment variables, NMMR uses adaptive feature maps from neural networks. NMMR is similar to MMR-IV [9], but MMR-IV is restricted to the instrumental variable (IV) setting rather than the proximal inference setting. Table 1 places NMMR in context with existing methods for proximal inference and IV regression.

Table 1: Comparison of the most related methods to NMMR.

| Method | Setting | # of Stages | Hypothesis Class | Optimization Objective |
|---|---|---|---|---|
| KPV [6] | Proximal | 2 | Kernels | 2-stage least squares |
| DFPV [7] | Proximal | 2 | Neural Networks | 2-stage least squares |
| MMR-IV [9] | IV | 1 | Neural Networks | V-statistic |
| PMMR [6] | Proximal | 1 | Kernels | V-statistic |
| **NMMR-V (ours)** | Proximal | 1 | Neural Networks | V-statistic |
| **NMMR-U (ours)** | Proximal | 1 | Neural Networks | U-statistic |

# 5 Consistency of NMMR

In this section we provide a probabilistic bound on the distance of the estimated bridge function, $\hat{h}_{k,\lambda,n}$, from the true bridge function, $h^*$, in terms of the Radamacher complexity $\mathcal{R}_n(\mathcal{F})$ of a class of functions $\mathcal{F}$ derived from elements of the hypothesis space $\mathcal{H}$ and the fixed kernel, $k$. Note that $\hat{R}_{k,\lambda,n}(h) = \hat{R}_{k,n}(h) + \lambda\Lambda[h, \theta_h]$ (see Section 4). We use this bound to demonstrate that, under mild conditions, $\hat{h}_{k,\lambda,n}$ converges in probability to $h^*$, and that, under an additional completeness assumption, $h^*$ is unique $P_{A,W,X}$-almost surely. This provides a consistent estimate of $\mathbb{E}[Y^a]$.

**Theorem 1.** *Let $\tilde{h}_k$ minimize $R_k(h)$ and $\hat{h}_{k,U,\lambda,n}$ minimize $\hat{R}_{k,U,\lambda,n}(h)$ for $h \in \mathcal{H}$, $k : (\mathcal{A} \times \mathcal{X} \times \mathcal{Z})^2 \to [-M_k, M_k]$, $\Lambda : \mathcal{H} \times \Theta_h \to [0, M_\lambda]$, and let $h^* : \mathcal{A} \times \mathcal{W} \times \mathcal{X} \to \mathbb{R}$ satisfy $\mathbb{E}[Y - h^*(A, W, X)|A, X, Z] = 0$ $P_{A,X,Z}$-almost surely, where*

$$R_k(h) = \mathbb{E}\left[(Y - h(A, W, X))(Y' - h(A', W', X'))k((A, X, Z), (A', X', Z'))\right]$$

$$\hat{R}_{k,U,\lambda,n}(h) = \frac{1}{n(n-1)} \sum_{i,j=1, i \neq j}^{n} [(y_i - h(a_i, w_i, x_i))(y_j - h(a_j, w_j, x_j))$$

$$\times k((a_i, x_i, z_i), (a_j, x_j, z_j))] + \lambda\Lambda[h, \theta_h]$$

*Also let,*

$$d_k^2(h, h') = \mathbb{E}[(h(A, W, X) - h'(A, W, X))(h(A', W', X') - h'(A', W', X'))$$
$$\times k((A, X, Z), (A', X', Z'))]$$

*Then, $d_k^2(h^*, h) = R_k(h)$ and, with probability at least $1 - \delta$,*

$$d_k^2\left(h^*, \hat{h}_{k,U,\lambda,n}\right) \leq d_k^2\left(h^*, \tilde{h}_k\right) + \lambda M_\lambda + 8M\mathbb{E}_{A,X,Z}\left(\mathcal{R}_{n-1}\left(\mathcal{F}'_{A,X,Z}\right) + \mathcal{R}_n\left(\mathcal{F}'_{A,X,Z}\right)\right)$$

$$+ 16M^2 M_k \left(\frac{2}{n} \log \frac{2}{\delta}\right)^{\frac{1}{2}} + 10(2\log 2)^{\frac{1}{2}} M^2 M_k n^{-\frac{1}{2}}$$

$$\leq d_k^2\left(h^*, \tilde{h}_k\right) + \lambda M_\lambda + 8M\left(\mathcal{R}_{n-1}\left(\mathcal{F}'\right) + \mathcal{R}_n\left(\mathcal{F}'\right)\right)$$

$$+ 16M^2 M_k \left(\frac{2}{n} \log \frac{2}{\delta}\right)^{\frac{1}{2}} + 10(2\log 2)^{\frac{1}{2}} M^2 M_k n^{-\frac{1}{2}}$$

*Further, if Assumption 5 holds, so $k$ is ISPD, then $d_k$ is a metric on $L^2_{\mathcal{A}\mathcal{X}\mathcal{Z}}$ and, if the right hand side of the inequality goes to zero as $n$ goes to infinity,*

$$d_k\left(\mathbb{E}[h^*|A, X, Z] - \mathbb{E}\left[\hat{h}_{k,\lambda,n}\Big|A, X, Z\right]\right) \xrightarrow{P} 0 \text{ so } \mathbb{E}\left[\hat{h}_{k,\lambda,n}\Big|A, X, Z\right] \xrightarrow{P} \mathbb{E}[h^*|A, X, Z]$$

*in $d_k$. Also, $\left\|\mathbb{E}[h^*|A, X, Z] - \mathbb{E}\left[\hat{h}_{k,\lambda,n}\Big|A, X, Z\right]\right\|_{P_{A,X,Z}} \xrightarrow{P} 0 \text{ so } \mathbb{E}\left[\hat{h}_{k,\lambda,n}\Big|A, X, Z\right] \xrightarrow{P}$
$\mathbb{E}[h^*|A, X, Z]$ in $L^2(P_{A,\mathcal{X},\mathcal{Z}}) - norm.*

$$\mathcal{F}'_{a,x,z} = \left\{f_{a,x,z} \mid \exists_{h\in\mathcal{H}}\forall_{a'\in\mathcal{A}, x'\in\mathcal{X}, z'\in\mathcal{Z}} f_{a,x,z}(a', w', x', z') = h(a', w', x') k((a', x', z'), (a, x, z))\right\}$$

$$\mathcal{F}' = \left\{f \mid \exists_{h\in\mathcal{H}, a\in\mathcal{A}, x\in\mathcal{X}, z\in\mathcal{Z}}\forall_{a'\in\mathcal{A}, x'\in\mathcal{X}, z'\in\mathcal{Z}} f(a', w', x', z') = h(a', w', x') k((a', x', z'), (a, x, z))\right\}$$

Corollary 8 provides a similar result for V-statistic estimators of $R(h)$, meaning we can choose to use either U or V-Statistics and have similar guarantees. In Theorem 1 if the quadratic form converges at

a particular rate, say $n^{-\frac{1}{2}}$, $\mathbb{E}\left[\hat{h}\Big|A, X, Z\right] \xrightarrow{\mathrm{P}} \mathbb{E}\left[h^*|A, X, Z\right]$, under the metric induced by the kernel, $d_k$, at half the rate, in this case $n^{-\frac{1}{4}}$. This is similar to Kallus et al. [13]'s findings in the unstabilized case.

**Theorem 2.** *Under Assumption 4, $h^*$ is the unique solution to the integral equation $\mathrm{P}_{A,W,X}$-almost surely. Further, if $\mathbb{E}\left[\hat{h}_{k,\lambda,n}\Big|A, X, Z\right] \xrightarrow{\mathrm{P}} \mathbb{E}\left[h^*|A, X, Z\right]$, $\hat{h}_n \xrightarrow{\mathrm{P}} h^*$.*

See Appendix A for proofs of Theorems 1 and 2. Taken together, these results tell us that, as long as our optimization algorithm is successful in estimating $\hat{h}_{k,\lambda,n}$, it will asymptotically approach the true bridge function, $h^*$. In order for this to occur, the right hand side of the inequalities in Theorem 1 must go to zero, which requires not only that the Rademacher terms vanish, but also that $\tilde{h}_k$ must approach $h^*$ arbitrarily closely as $n$ increases. In practice, this means increasing the complexity of the neural network, but doing so slowly enough the Rademacher complexity terms still decrease with sample size. Following Xu et al. [7], we note that recent results from Neyshabur et al. [24] suggest that the Rademacher complexity of a fixed network scales like $n^{-\frac{1}{2}}$ (similar to many other popular hypothesis classes) and that, although we cannot compute the scaling of the Rademacher terms directly due to the presence of the kernel function, we expect that they will decline with sample size and that, as the neural network becomes more complex, their scaling will more closely resemble terms derived from a pure neural network. Finally, we require that the regularization parameter decrease as sample size grows, which will, again, depend on the balance between increasing sample size, which tends to decrease the need for regularization, and increasing complexity, which tends to increase its importance. Thus, by choosing an appropriate growth rate for the network complexity, we expect the aforementioned terms to vanish as $n$ increases to infinity, and, with them, the entire right hand side, making $\hat{h}_{k,\lambda,n}$ a consistent (likely $\sqrt{n}$) estimator of $h^*$.

We can also compare the convergence of the estimated bridge function to that of its projection onto $L^2_{\mathcal{A}\mathcal{X}\mathcal{Z}}$. Prior literature has focused on a measure of "ill-posedness" $\tau = \sup_{h \in \mathcal{H}} \|h - h^*\|_2 \|\mathbb{E}\left[h - h^*|A, Z, X\right]\|_2^{-1}$. If $\tau$ is finite, then the rate of convergence of the estimated bridge function will be at worst $\tau$ times that of its projection (it will be slower by a factor of $\tau$). This will be the case whether we measure convergence using $\|\mathbb{E}\left[h - h^*|A, Z, X\right]\|_2$ or the metric induced by $k$.

## 6 Experiments

### 6.1 Overview of Baseline Models

We compare the performance of NMMR-U and NMMR-V to that of several previous approaches, which we describe briefly here. The baselines can be divided into two categories: structural and naive. Structural approaches leverage causal information about the data generating process. They include Kernel Proxy Variables (KPV) [6], Proximal Maximum Moment Restriction (PMMR) [6], Deep Feature Proxy Variables (DFPV) [25], Causal Effect Variational Autoencoder[26] (CEVAE), and the two-stage least squares model (2SLS) from Miao et al. [4]. For a review of KPV, PMMR, and DFPV, see Section 3. CEVAE is an autoencoder approach derived by Xu et al. [7] from Louizos et al. [26]. 2SLS is a two-stage least squares method which assumes that the bridge function $h$ is linear [5].

The naive approaches serve as baselines and do not use causal information, instead directly regressing $A$ and $W$ on the outcome $Y$. These methods include a naive neural network (Naive net), ordinary least squares regression (LS), and ordinary least squares with quadratic features (LS-QF). Naive Net is a neural network that has undergone the same architecture search as NMMR (described further in Appendix B) that is trained to predict $Y$ directly from $A$ and $W$ by minimizing observational MSE, $\frac{1}{n}\sum_{i=1}^{n}(y - \hat{y})^2$. Least Squares (LS) is the standard linear regression model that predicts $Y$ using a linear combination of $A$ and $W$. Least Squares with Quadratic Features (LS-QF) is the same as LS but with additional quadratic terms $A^2, W^2, AW$.

We evaluate NMMR-U, NMMR-V and baseline methods on two synthetic benchmark tasks from Xu et al. [7]. The first is a simulation of how ticket prices affect the number of tickets sold in the presence of a latent confounder: demand for travel (the Demand experiment). The second is an experiment where the goal is to recover a property of an image that is influenced by an unobserved confounder (the dSprite experiment). The Demand experiment is a low-dimensional estimation problem, whereas dSprite is high-dimensional as $A$ and $W$ are 64x64=4096-dimensional. dSprite leverages image-

specific models, which are rarely used in the causal inference literature. Neither task uses $X$. For the Demand experiment we evaluate all the methods mentioned above, whereas for the dSprite experiment, we omit 2SLS, LS, and LS-QF because of their lack of scalability to high-dimensional settings.

Experiments were conducted in PyTorch 1.9.0 (Python 3.9.7), using an A100 40GB or TitanX 12GB GPU and CUDA version 11.2. They can be run in minutes for simpler models (LS, LS-QF, 2SLS) and in several hours for the larger experiments and more complex models (DFPV, NMMR). The code to reproduce our experiments can be accessed on GitHub[2]

## 6.2 Demand Experiment

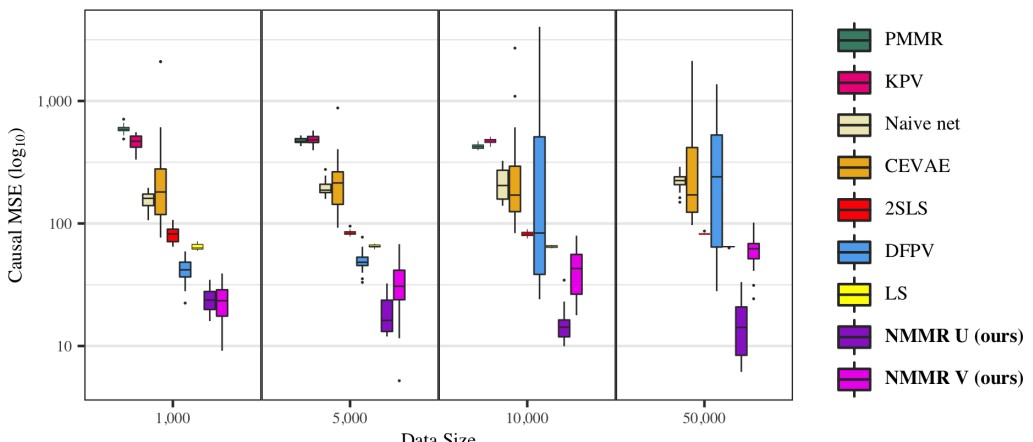

Figure 2: **NMMR-U and NMMR-V achieve state of the art performance across all sample sizes**. Causal MSE (c-MSE) of NMMR and baseline methods in the Demand experiment. Each method was replicated 20 times and evaluated on the same 10 test values of $\mathbb{E}[Y^a]$ each replicate. Each individual box plot represents 20 values of c-MSE. See Table S4 for the statistics of each boxplots

Hartford et al. [21] introduced a data generating process for studying instrumental variable regression, and Xu et al. [7] adapted it to the proximal setting. The goal is to estimate the effect of airline ticket price $A$ on sales $Y$, which is confounded by demand $U$ (e.g. seasonal fluctuations). We use the cost of fuel, $Z = (Z_1, Z_2)$, as a treatment-inducing proxy and number of views at a ticket reservation website, $W$, as an outcome-inducing proxy (Figure S1). Additional simulation details and the structural equations underlying the causal DAG can be found in Appendix C.1

Each method was trained on simulated datasets with sample sizes of 1000, 5000, 10,000, and 50,000. To assess the performance of each method, we evaluated $a$ at 10 equally-spaced intervals between 10 and 30. We compared each method's estimated potential outcomes, $\hat{E}[Y^a]$, against estimates of the truth, $E[Y^a]$, obtained from Monte Carlo simulations (10,000 replicates) of the data generating process for each $a$. The evaluation metric is the causal mean squared error (c-MSE) across the 10 evaluation points of $a$: $\frac{1}{10}\sum_{i=1}^{10}(\mathbb{E}[Y^{a_i}] - \hat{\mathbb{E}}[Y^{a_i}])^2$. For MMR-based methods, predictions are computed using a heldout dataset, $\mathcal{D}_{\mathcal{W}}$ with 1,000 draws from $W$ so $\hat{\mathbb{E}}[Y^{a_i}] = |D_{\mathcal{W}}|^{-1}\sum_j^{|D_{\mathcal{W}}|} \hat{h}(a_i, w_j)$, i.e. a sample average of the estimated bridge function over $W$. We performed 20 replicates for each method on each sample size, where a single replicate yields one c-MSE value. Figure 2 summarizes the c-MSE distribution for each method across the four sample sizes. NMMR-U has the lowest c-MSE across all sample sizes, with NMMR-V a close second. DFPV encounters difficulties with the larger sample sizes of 10,000 and 50,000, potentially due to convergence issues with its feature maps. Similarly, PMMR and KPV could not scale to $n = 50,000$.

For a more in-depth view of the potential outcome curve estimated by each method, we provide replicate-wise potential outcome prediction curves for each of the 4 sample sizes in Figures S3-S6. Least Squares estimates relatively unbiased prediction curves due to the nature of the data generating process and has very low variance. LS-QF matches some of the curvature, although its c-MSE

---

[2]https://github.com/beamlab-hsph/Neural-Moment-Matching-Regression

distribution (not shown) is not better than LS. Kernel-based methods, KPV and PMMR, are highly biased. DFPV is less biased, but still suffers from a lack of flexibility. Both NMMR variants demonstrate the benefit of added flexibility and have lower variance, resulting in a lower c-MSE.

Finally, we also varied the variance of the Gaussian noise terms in the structural equations for $Z$ and $W$, in order to examine how each method performs with varying quality proxies for $U$ (see Appendix E). Figure S10 shows that NMMR-V is more robust to proxy noise than NMMR-U. This could be because U-statistics yield unbiased, but higher variance, estimators than V-statistics, so, when proxies are less reliable, the estimated risk function $R_k(h)$ is less stable. Kernel-based methods (KPV and PMMR) perform increasingly well with noisier proxies, which is likely related to the fact that they are less data-adaptive. Figures S11 through S18 show replication-wise prediction curves across all 72 noise levels, with one grid plot per method.

### 6.3 dSprite Experiment

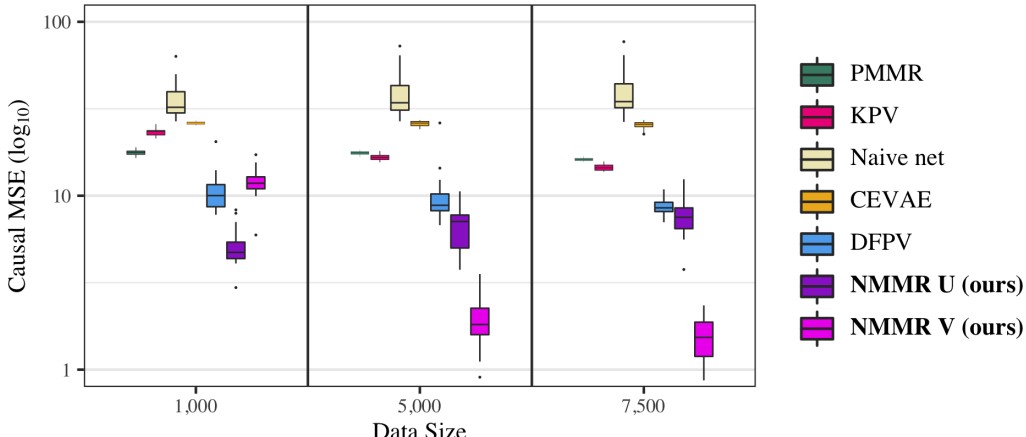

Figure 3: Causal MSE (c-MSE) of NMMR and baseline methods in the dSprite experiment. Each method was replicated 20 times and evaluated on the same 588 test images $A$ each replicate. Each individual box plot represents 20 values of c-MSE. See Table S5 for the statistics of each boxplots

The second benchmark uses the dSprite dataset from Matthey et al. [27], which was initially adapted to instrumental variable regression in Xu et al. [25], and repurposed for proximal inference in Xu et al. [7]. This image dataset consists of 2D shapes procedurally generated from 6 independent parameters: color, shape, scale, rotation, posX, and posY. All possible combinations of these parameters are present exactly once, generating 737,280 total images. In this experiment, we fix shape = heart, color = white, resulting in 245,760 images, each of which contains 64x64=4096 pixels. The causal DAG for this problem is shown in Figure S7. The structural equations and detailed data generating mechanism underlying the causal DAG can be found in Appendix C.6.

In the DAG, $Fig(\cdot)$ represents the act of retrieving the image from the dSprite dataset with the given arguments. $A$ and $W$ are vectors representing noised images of a heart shape, where the heart has a size (*scale*), orientation (*rotation*), horizontal position (*posX*), and vertical position (*posY*). For an exemplar image $A$ and $W$, see Figure S8. The benchmark computes

$$\mathbb{E}[Y^a] = \frac{\frac{1}{10}\left\|vec(a)^t B\right\|_2^2 - 5000}{1000}$$

where $B$ is a $4096 \times 10$ matrix of $\mathcal{U}(0,1)$ weights from Xu et al. [7]. The observed outcome is computed as

$$Y = \frac{\frac{1}{10}\left\|vec(A)^t B\right\|_2^2 - 5000}{1000} \times \frac{(31 \times U - 15.5)^2}{85.25} + \epsilon, \quad \epsilon \sim \mathcal{N}(0, 0.5)$$

$U$ is a discrete uniform random variable with

$$\mathbb{E}\left[\frac{(31 \times U - 15.5)^2}{85.25}\right] = 1$$

that dictates the vertical position of the shape in $A$, as well as the value of $Y$, making $U$ a confounder of $A, Y$. We hypothesized that a convolutional neural network would be exceptionally strong at recovering this information about $U$ from the images $A$ and $W$.

Similar to the Demand experiment, we trained each method on simulated datasets with sizes 1,000, 5,000, and 7,500, followed by an evaluation on the same test set as Xu et al. [7]. This test set contains 588 images $A$ that span the range of scale, rotation, posX and posY values (see Appendix C.9) and the 588 corresponding values of $\mathbb{E}[Y^a]$. The evaluation metric is again c-MSE:

$$\frac{1}{588} \sum_{i=1}^{588} \left( \mathbb{E}[Y^{a_i}] - \hat{\mathbb{E}}[Y^{a_i}] \right)^2$$

We performed 20 replicates for each method on each sample size. Figure 3 shows that NMMR-U or NMMR-V is consistently lowest in c-MSE, with NMMR-V showing substantial improvement with increasing sample size. Due to the high dimensionality of the images $A$ and $W$, we could not evaluate Least Squares, LS-QF or 2SLS on this experiment. KPV and PMMR do not improve much with increasing sample size. The Naive net, which uses the same underlying convolutional neural network architecture as NMMR but is trained using observational MSE, performs second-to-worst, with a much larger c-MSE than NMMR-U or NMMR-V. This reinforces the need to use causal knowledge in scenarios where it is available.

## 7 Conclusion

In this work we have presented a novel method to estimate potential outcomes in the presence of unmeasured confounding using deep neural networks. Though our method is promising, it has several limitations. For very high dimensional data, calculating the kernel matrix $K$ in the loss function can be computationally intensive (see Appendix D). Additionally, mapping real world scenarios to DAGs that satisfy Assumption 1 is non-trivial and technically unverifiable (e.g. we cannot be truly sure that $W$ has no impact on $A$), though unverifiable assumptions are inherent to causal inference.

Further, the present work focuses on methods that estimate only the outcome bridge function, rather than also estimating the IPW bridge function, which would permit us to construct a doubly robust estimator, as is done in Cui et al. [11] and Kallus et al. [13]. However, our method extends naturally to this setting and we expect to explore such estimators in future work.

In summary, we provide a new single stage estimator and show how it can be trained on a U-statistic based loss in addition to existing approaches based on V-statistics. We further prove theoretical convergence properties of our method. On established proximal inference benchmarks, our method achieves state of the art performance in estimating causal quantities. Finally, since our approach is a single-stage neural network, it potentially unlocks new domains for causal inference where deep learning has had success, such as imaging.

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
