# A Proofs

**Lemma 3.** *Let $X$ be a random variable taking values in $\mathcal{X}$ and let $\mathcal{F}$ be a family of measurable functions with $f \in \mathcal{F} : \mathcal{X}^2 \to [-M, M]$, with each $f \in \mathcal{F}$ optionally, and possibly not uniquely, (partially) parameterized by $\theta_f \in \Theta_{\mathcal{F}}$, $\Lambda : \mathcal{F} \times \Theta_{\mathcal{F}} \to [0, M_\Lambda]$, and $\lambda \geq 0$. Then, for any $\delta > 0$, with probability at least $1 - \delta$, any IID sample $S = \{x_i\}_{i=1}^n$ drawn from $\mathrm{P}_X$ satisfies*

$$\mathbb{E}f = \mathbb{E}_{X,X'} f(X, X') \leq \frac{1}{n(n-1)} \sum_{i,j=1, i \neq j}^n f(x_i, x_j) + \lambda \Lambda[f, \theta_f]$$

$$+ 2(\mathcal{R}_{n-1}(\mathcal{F}_1) + \mathcal{R}_n(\mathcal{F}_2)) + 2M\left(\frac{2}{n} \log \frac{1}{\delta}\right)^{\frac{1}{2}}$$

$$\frac{1}{n(n-1)} \sum_{i,j=1, i \neq j}^n f(x_i, x_j) + \lambda \Lambda[f, \theta_f] \leq \mathbb{E}f + \lambda M_\Lambda + 2(\mathcal{R}_{n-1}(\mathcal{F}_1) + \mathcal{R}_n(\mathcal{F}_2))$$

$$+ 2M\left(\frac{2}{n} \log \frac{1}{\delta}\right)^{\frac{1}{2}}$$

$$\mathbb{E}f = \mathbb{E}_{X,X'} f(X, X') \leq \frac{1}{n(n-1)} \sum_{i,j=1, i \neq j}^n f(x_i, x_j) + \lambda \Lambda[f, \theta_f]$$

$$+ 2\left(\hat{\mathcal{R}}_{n-1,S}(\mathcal{F}_1) + \mathcal{R}_S(\mathcal{F}_2)\right) + 6M\left(\frac{2}{n} \log \frac{2}{\delta}\right)^{\frac{1}{2}}$$

$$\frac{1}{n(n-1)} \sum_{i,j=1, i \neq j}^n f(x_i, x_j) + \lambda \Lambda[f, \theta_f] \leq \mathbb{E}f + \lambda M_\Lambda + 2\left(\hat{\mathcal{R}}_{n-1,S}(\mathcal{F}_1) + \mathcal{R}_S(\mathcal{F}_2)\right)$$

$$+ 6M\left(\frac{2}{n} \log \frac{2}{\delta}\right)^{\frac{1}{2}}$$

*where $\mathcal{R}_S$ is the empirical Rademacher Complexity, given by:*

$$\mathcal{R}_S(\mathcal{F}) = \mathbb{E}_\epsilon \sup_{f \in \mathcal{F}} \frac{1}{n} \sum_{i=1}^n \epsilon_i f(x_i)$$

*where $\epsilon$ is a Rademacher random vector taking values uniformly in $\{-1, 1\}^n$, $S_{-i} = \{x_j\}_{j=1, j \neq i}^n$, $\hat{\mathcal{R}}_{n-1,S}(\mathcal{F}) = n^{-1} \sum_{i=1}^n \mathcal{R}_{S_{-i}}(\mathcal{F})$, $\mathcal{F}_1 = \{g \mid \exists_{f \in \mathcal{F}, x' \in \mathcal{X}} \forall_{x \in \mathcal{X}} [g(x) = f(x', x)]\}$, $\mathcal{F}_2 = \{g \mid \exists_{f \in \mathcal{F}, x' \in \mathcal{X}} \forall_{x \in \mathcal{X}} [g(x) = f(x, x')]\}$, and $\mathcal{R}_n$ is the (expected) Rademacher complexity for a sample of size $n$, $\mathcal{R}_n = \mathbb{E}_S \mathcal{R}_S$, where the expectation is over all samples, $S$, of size $n$.*

*Finally, if the elements of $\mathcal{F}$ are symmetric, so that $\forall_{f \in \mathcal{F}, x, x' \in \mathcal{X}} (f(x, x') = f(x', x))$, $\mathcal{F}_1 = \mathcal{F}_2$.*

In particular, $\theta_f$ might be the weights associated with the neural network $f$. Note that, as is the case for neural networks, $\theta_f$ may not be uniquely determined by $f$, so that multiple $\theta$s may be associated with the same $f$. We may also take $\Theta_{\mathcal{F}} = \emptyset$, so that $f$ is regarded purely as an element of $\mathcal{F}$.

*Proof.* Let $\hat{E}_S f = \frac{1}{n(n-1)} \sum_{i,j=1, j \neq i} f(x_i, x_j)$, $\hat{E}_{S,\lambda} f = \frac{1}{n(n-1)} \sum_{i,j=1, j \neq i} f(x_i, x_j) + \lambda \Lambda[f, \theta_f]$, $\tilde{E}_S f = \frac{1}{n} \sum_{i=1}^n E_X f(x_i, X)$, $\phi(S) = \sup_{f \in \mathcal{F}} \left(\hat{E}_S f - \mathbb{E}f\right)$, $\phi^{(1)}(S) = \sup_{f \in \mathcal{F}} \left(\hat{E}_S f - \tilde{E}_S f\right)$, and $\phi^{(2)}(S) = \sup_{f \in \mathcal{F}} \left(\tilde{E}_S f - \mathbb{E}f\right)$. Note that $\mathbb{E}_S \hat{E}_S f = \mathbb{E}_{X,X'} f(X, X') = \mathbb{E}f = \mathbb{E}_{X,X'} f(X, X') = \mathbb{E}_S \tilde{E}_S f$. Also, let $S' = \{x_i'\}_{i=1}^n$ and let $S_i = \{x_{i,j}\}_{j=1}^n$ be obtained from $S$ by replacing $x_i$ by $x_i'$, so that $x_{i,j} = x_j$ for $j \neq i$ and $x_{i,i} = x_i'$. In order to apply McDiarmid's Inequality, we must find bounds, $c_i$ such that $|\phi(S_i) - \phi(S)| \leq c_i$ for $i = 1, \ldots, n$.

$$
\begin{aligned}
|\phi\left(S_i\right) - \phi(S)| &= \left| \sup_{f \in \mathcal{F}} \left( \hat{\mathbb{E}}_{S_i} f - \mathbb{E}f \right) - \sup_{f \in \mathcal{F}} \left( \hat{\mathbb{E}}_S f - \mathbb{E}f \right) \right| \\
&\leq \sup_{f \in \mathcal{F}} \left| \left( \hat{\mathbb{E}}_{S_i} f - \mathbb{E}f \right) - \left( \hat{\mathbb{E}}_S f - \mathbb{E}f \right) \right| \\
&= \sup_{f \in \mathcal{F}} \left| \hat{\mathbb{E}}_{S_i} f - \hat{\mathbb{E}}_S f \right| \\
&= \sup_{f \in \mathcal{F}} \left| \frac{1}{n(n-1)} \sum_{j,k=1,k \neq j}^{n} f\left(x_{i,j}, x_{i,k}\right) - \frac{1}{n(n-1)} \sum_{j,k=1,k \neq j}^{n} f\left(x_j, x_k\right) \right| \\
&\leq \frac{1}{n(n-1)} \sup_{f \in \mathcal{F}} \sum_{j,k=1,k \neq j}^{n} \left| f\left(x_{i,j}, x_{i,k}\right) - f\left(x_j, x_k\right) \right| \\
&= \frac{1}{n(n-1)} \left( \sum_{j=1,j \neq k, k=i}^{n} \sup_{f \in \mathcal{F}} \left| f\left(x_{i,j}, x_{i,i}\right) - f\left(x_j, x_i\right) \right| \right. \\
&\qquad\qquad\qquad \left. + \sum_{k=1,k \neq j, j=i}^{n} \sup_{f \in \mathcal{F}} \left| f\left(x_{i,i}, x_{i,k}\right) - f\left(x_i, x_k\right) \right| \right) \\
&= \frac{1}{n(n-1)} \left( \sum_{j=1,j \neq i}^{n} \sup_{f \in \mathcal{F}} \left| f\left(x_j, x_i'\right) - f\left(x_j, x_i\right) \right| \right. \\
&\qquad\qquad\qquad \left. + \sum_{k=1,k \neq i}^{n} \sup_{f \in \mathcal{F}} \left| f\left(x_i', x_k\right) - f\left(x_i, x_k\right) \right| \right) \\
&= \frac{1}{n(n-1)} \left( \sum_{j=1,j \neq i}^{n} \sup_{f \in \mathcal{F}} 2M + \sum_{k=1,k \neq i}^{n} \sup_{f \in \mathcal{F}} 2M \right) \leq \frac{2}{n(n-1)} \cdot (n-1) \cdot 2M \\
&= 4Mn^{-1}
\end{aligned}
$$

so we can choose $c_i = c = 4Mn^{-1}$. The exponent in McDiarmid's Inequality is then $-2\epsilon^2 \left( \sum_{i=1}^{n} c_i^2 \right)^{-1} = -2\epsilon^2 \left( n \cdot \left(4Mn^{-1}\right)^2 \right)^{-1} = -2\epsilon^2 \left(16M^2 n^{-1}\right)^{-1} = -\frac{1}{8} nM^{-2}\epsilon^2$. Setting $\frac{\delta}{2} = e^{-\frac{1}{8} nM^{-2}\epsilon^2}$ gives $\epsilon = \left( -\frac{8M^2}{n} \log \frac{\delta}{2} \right)^{\frac{1}{2}} = M \left( \frac{8}{n} \log \frac{2}{\delta} \right)^{\frac{1}{2}} = 2M \left( \frac{2}{n} \log \frac{2}{\delta} \right)^{\frac{1}{2}}$. Then, McDiarmid's Inequality yields,

$$
\mathrm{P}\left[\phi(S) - \mathbb{E}_S\phi(S) \geq \epsilon\right] \leq \frac{\delta}{2}, \quad \mathrm{P}\left[\mathbb{E}_S\phi(S) - \phi(S) \geq \epsilon\right] \leq \frac{\delta}{2}
$$

so that, with probability $1 - \frac{\delta}{2}$, $\phi(S) \leq \mathbb{E}_S\phi(S) + 2M \left( \frac{2}{n} \log \frac{2}{\delta} \right)^{\frac{1}{2}}$. We now need to compute $\mathbb{E}_S\phi(S)$. However, this is difficult to do directly, so we instead compute it separately for $\phi^{(1)}$ and $\phi^{(2)}$. Let $\epsilon$ be a Rademacher random vector taking values uniformly in $\{-1, 1\}^n$. Then,

$$\mathbb{E}_S \phi^{(1)}(S) = \mathbb{E}_S \sup_{f \in \mathcal{F}} \left( \hat{E}_S f - \tilde{E}_S f \right)$$

$$= \mathbb{E}_S \sup_{f \in \mathcal{F}} \left( \frac{1}{n(n-1)} \sum_{i,j=1, j \neq i}^{n} f(x_i, x_j) - \frac{1}{n} \sum_{i=1}^{n} E_X f(x_i, X) \right)$$

$$= \mathbb{E}_S \sup_{f \in \mathcal{F}} \left( \frac{1}{n(n-1)} \sum_{i,j=1, j \neq i}^{n} f(x_i, x_j) - \frac{1}{n} \sum_{i=1}^{n} E_{S'} \frac{1}{n-1} \sum_{j=1, j \neq i}^{n} f(x_i, x_j') \right)$$

$$= \mathbb{E}_S \sup_{f \in \mathcal{F}} E_{S'} \frac{1}{n(n-1)} \sum_{i,j=1, j \neq i}^{n} \left( f(x_i, x_j) - f(x_i, x_j') \right)$$

$$\leq \mathbb{E}_{S,S'} \sup_{f \in \mathcal{F}} \frac{1}{n(n-1)} \sum_{i,j=1, j \neq i}^{n} \left( f(x_i, x_j) - f(x_i, x_j') \right)$$

$$\leq \frac{1}{n} \sum_{i=1}^{n} \mathbb{E}_{S,S'} \sup_{f \in \mathcal{F}} \frac{1}{n-1} \sum_{j=1, j \neq i}^{n} \left( f(x_i, x_j) - f(x_i, x_j') \right)$$

$$= \frac{1}{n} \sum_{i=1}^{n} \mathbb{E}_{x_i, S_{-i}, S_{-i}'} \sup_{f \in \mathcal{F}} \frac{1}{n-1} \sum_{j=1, j \neq i}^{n} \left( f(x_i, x_j) - f(x_i, x_j') \right)$$

$$= \frac{1}{n} \sum_{i=1}^{n} \mathbb{E}_\epsilon \mathbb{E}_{x_i, S_{-i}, S_{-i}'} \sup_{f \in \mathcal{F}} \frac{1}{n-1} \sum_{j=1, j \neq i}^{n} \epsilon_j \left( f(x_i, x_j) - f(x_i, x_j') \right)$$

$$\leq \frac{1}{n} \sum_{i=1}^{n} \left[ \mathbb{E}_{x_i, S_{-i}, \epsilon} \sup_{f \in \mathcal{F}} \frac{1}{n-1} \sum_{j=1, j \neq i}^{n} \epsilon_j f(x_i, x_j) \right.$$

$$\left. + \mathbb{E}_{x_i, S_{-i}', \epsilon} \sup_{f \in \mathcal{F}} \frac{1}{n-1} \sum_{j=1, j \neq i}^{n} -\epsilon_j f(x_i, x_j') \right]$$

$$= \frac{1}{n} \sum_{i=1}^{n} \mathbb{E}_{x_i} \left[ \mathbb{E}_{S_{-i}, \epsilon} \sup_{f \in \mathcal{F}} \frac{1}{n-1} \sum_{j=1, j \neq i}^{n} \epsilon_j f(x_i, x_j) \right.$$

$$\left. + \mathbb{E}_{S_{-i}, \epsilon} \sup_{f \in \mathcal{F}} \frac{1}{n-1} \sum_{j=1, j \neq i}^{n} \epsilon_j f(x_i, x_j) \right]$$

$$= \frac{2}{n} \sum_{i=1}^{n} \mathbb{E}_{x_i} \mathbb{E}_{S_{-i}, \epsilon} \sup_{f \in \mathcal{F}} \frac{1}{n-1} \sum_{j=1, j \neq i}^{n} \epsilon_j f(x_i, x_j)$$

$$= \frac{2}{n} \sum_{i=1}^{n} \mathbb{E}_{x_n} \mathbb{E}_{S_{-n}, \epsilon} \sup_{f \in \mathcal{F}} \frac{1}{n-1} \sum_{j=1}^{n-1} \epsilon_j f(x_n, x_j)$$

$$= 2\mathbb{E}_{x_n} \mathbb{E}_{S_{-n}, \epsilon} \sup_{f \in \mathcal{F}} \frac{1}{n-1} \sum_{j=1}^{n-1} \epsilon_j f(x_n, x_j) = 2\mathbb{E}_X \mathbb{E}_{S_{-n}, \epsilon} \sup_{f \in \mathcal{F}} \frac{1}{n-1} \sum_{i=1}^{n-1} \epsilon_i f(X, x_i)$$

$$\leq 2\mathbb{E}_{S_{-n}, \epsilon} \sup_{f \in \mathcal{F}, x \in \mathcal{X}} \frac{1}{n-1} \sum_{i=1}^{n-1} \epsilon_i f(x, x_i) = 2\mathbb{E}_{S_{-n}} \mathbb{E}_\epsilon \sup_{f \in \mathcal{F}, x \in \mathcal{X}} \frac{1}{n-1} \sum_{i=1}^{n-1} \epsilon_i f(x, x_i)$$

$$= 2\mathcal{R}_{n-1}(\mathcal{F}_1)$$

where, in the seventh line, we note that the inner sum depends only on $S = \{x_i\} \cup S_{-i}$ and $S_{-i}'$, but not $x_i'$, in the eighth line, we introduce Rademacher variables because reversing the order of the difference is equivalent to swapping elements between $S_{-i}$ and $S_{-i}'$ and, since the expectation is over

all possible pairs of samples, its value is unchanged, in the tenth line, we note that negation simply interchanges pairs of Rademacher vectors, so the expectation is unchanged, and, in the final line, $\mathcal{F}_1 = \{g \mid \exists_{f \in \mathcal{F}, x' \in \mathcal{X}} \forall_{x \in \mathcal{X}} [g(x) = f(x', x)]\}$.

$$
\begin{aligned}
\mathbb{E}_S \phi^{(2)}(S) = \mathbb{E}_S \sup_{f \in \mathcal{F}} \left( \tilde{\mathbb{E}}_S f - \mathbb{E}f \right) &= \mathbb{E}_S \sup_{f \in \mathcal{F}} \left( \tilde{\mathbb{E}}_S f - \mathbb{E}_{S'} \tilde{\mathbb{E}}_{S'} f \right) = \mathbb{E}_S \sup_{f \in \mathcal{F}} \mathbb{E}_{S'} \left( \tilde{\mathbb{E}}_S f - \tilde{\mathbb{E}}_{S'} f \right) \\
&\leq \mathbb{E}_{S,S'} \sup_{f \in \mathcal{F}} \left( \tilde{\mathbb{E}}_S f - \tilde{\mathbb{E}}_{S'} f \right) \\
&= \mathbb{E}_{S,S'} \sup_{f \in \mathcal{F}} \left( \frac{1}{n} \sum_{i=1}^n E_X f(x_i, X) - \frac{1}{n} \sum_{i=1}^n E_X f(x_i', X) \right) \\
&= \mathbb{E}_{S,S'} \sup_{f \in \mathcal{F}} \left( \frac{1}{n} \sum_{i=1}^n \left( E_X f(x_i, X) - E_X f(x_i', X) \right) \right) \\
&= \mathbb{E}_\epsilon \mathbb{E}_{S,S'} \sup_{f \in \mathcal{F}} \left( \frac{1}{n} \sum_{i=1}^n \epsilon_i \left( E_X f(x_i, X) - E_X f(x_i', X) \right) \right) \\
&\leq \mathbb{E}_{S,\epsilon} \sup_{f \in \mathcal{F}} \frac{1}{n} \sum_{i=1}^n \epsilon_i E_X f(x_i, X) + \mathbb{E}_{S',\epsilon} \sup_{f \in \mathcal{F}} \frac{1}{n} \sum_{i=1}^n -\epsilon_i E_X f(x_i', X) \\
&= \mathbb{E}_{S,\epsilon} \sup_{f \in \mathcal{F}} \frac{1}{n} \sum_{i=1}^n \epsilon_i E_X f(x_i, X) + \mathbb{E}_{S,\epsilon} \sup_{f \in \mathcal{F}} \frac{1}{n} \sum_{i=1}^n \epsilon_i E_X f(x_i, X) \\
&= 2\mathbb{E}_{S,\epsilon} \sup_{f \in \mathcal{F}} \frac{1}{n} \sum_{i=1}^n \epsilon_i E_X f(x_i, X) \leq 2\mathbb{E}_{S,\epsilon} \sup_{f \in \mathcal{F}, x \in \mathcal{X}} \frac{1}{n} \sum_{i=1}^n \epsilon_i f(x_i, x) \\
&= 2\mathcal{R}_n(\mathcal{F}_2)
\end{aligned}
$$

where, in the fifth line, we introduce Rademacher variables because changing the order of the difference is equivalent to swapping elements between $S$ and $S'$, and, since the expectation is over all possible pairs of samples, its value is unchanged, in the seventh line, we note that negation simply interchanges pairs of Rademacher vectors leaving the expectation unchanged, and, in the final line, $\mathcal{F}_2 = \{g \mid \exists_{f \in \mathcal{F}, x' \in \mathcal{X}} \forall_{x \in \mathcal{X}} [g(x) = f(x, x')]\}$. Then,

$$
\begin{aligned}
\mathbb{E}_S \phi(S) = \mathbb{E}_S \sup_{f \in \mathcal{F}} \left( \hat{\mathbb{E}}_S f - \mathbb{E}f \right) &= \mathbb{E}_S \sup_{f \in \mathcal{F}} \left( \hat{\mathbb{E}}_S f - \tilde{\mathbb{E}}_S f + \tilde{\mathbb{E}}_S f - \mathbb{E}f \right) \\
&\leq \mathbb{E}_S \sup_{f \in \mathcal{F}} \left( \hat{\mathbb{E}}_S f - \tilde{\mathbb{E}}_S f \right) + \mathbb{E}_S \sup_{f \in \mathcal{F}} \left( \tilde{\mathbb{E}}_S f - \mathbb{E}f \right) = \mathbb{E}_S \phi^{(1)}(S) + \mathbb{E}_S \phi^{(2)}(S) \\
&= 2\mathcal{R}_{n-1}(\mathcal{F}_1) + 2\mathcal{R}_n(\mathcal{F}_2)
\end{aligned}
$$

Finally, combining the above results tells us that, with probability $1 - \frac{\delta}{2}$, $\phi(S) \leq 2(\mathcal{R}_{n-1}(\mathcal{F}_1) + \mathcal{R}_n(\mathcal{F}_2)) + 2M \left( \frac{2}{n} \log \frac{2}{\delta} \right)^{\frac{1}{2}}$ so $\hat{\mathbb{E}}_S f \leq \mathbb{E}f + 2(\mathcal{R}_{n-1}(\mathcal{F}_1) + \mathcal{R}_n(\mathcal{F}_2)) + 2M \left( \frac{2}{n} \log \frac{2}{\delta} \right)^{\frac{1}{2}}$. Replacing $\phi(S)$ by $\phi'(S) = \sup_{f \in \mathcal{F}} \left( \mathbb{E}f - \hat{\mathbb{E}}_S f \right)$ in the above proof yields $\mathbb{E}f \leq \hat{\mathbb{E}}_S f + 2(\mathcal{R}_{n-1}(\mathcal{F}_1) + \mathcal{R}_n(\mathcal{F}_2)) + 2M \left( \frac{2}{n} \log \frac{2}{\delta} \right)^{\frac{1}{2}}$. Since $\lambda \geq 0$, $0 \leq \Lambda \leq M_\Lambda$, $0 \leq \lambda \Lambda[f, \theta_f] \leq \lambda M_\Lambda$, so we also have,

$$
\mathbb{E}f \leq \hat{\mathbb{E}}_S f + \lambda \Lambda[f, \theta_f] + 2(\mathcal{R}_{n-1}(\mathcal{F}_1) + \mathcal{R}_n(\mathcal{F}_2)) + 2M \left( \frac{2}{n} \log \frac{2}{\delta} \right)^{\frac{1}{2}}
$$

$$
= \hat{\mathbb{E}}_{S,\lambda} f + 2(\mathcal{R}_{n-1}(\mathcal{F}_1) + \mathcal{R}_n(\mathcal{F}_2)) + 2M \left( \frac{2}{n} \log \frac{2}{\delta} \right)^{\frac{1}{2}}
$$

$$
\hat{\mathbb{E}}_{S,\lambda} f = \hat{\mathbb{E}}_S f + \lambda \Lambda[f, \theta_f] \leq \mathbb{E}f + \lambda M_\Lambda + 2(\mathcal{R}_{n-1}(\mathcal{F}_1) + \mathcal{R}_n(\mathcal{F}_2)) + 2M \left( \frac{2}{n} \log \frac{2}{\delta} \right)^{\frac{1}{2}}
$$

Using $2\delta$ in place of $\delta$, we see that, with probability at least $1 - \delta$, $\mathbb{E}f \leq \hat{\mathbb{E}}_{S,\lambda}f + 2\left(\mathcal{R}_{n-1}\left(\mathcal{F}_1\right) + \mathcal{R}_n\left(\mathcal{F}_2\right)\right) + 2M\left(\frac{2}{n}\log\frac{1}{\delta}\right)^{\frac{1}{2}}$ and $\hat{\mathbb{E}}_{S,\lambda}f \leq \mathbb{E}f + \lambda M_\Lambda + 2\left(\mathcal{R}_{n-1}\left(\mathcal{F}_1\right) + \mathcal{R}_n\left(\mathcal{F}_2\right)\right) + 2M\left(\frac{2}{n}\log\frac{1}{\delta}\right)^{\frac{1}{2}}$, yielding the first two inequalities.

In order to obtain results in terms of the empirical Rademacher complexity, $\mathcal{R}_S$, instead of the (expected) Rademacher complexity, $\mathcal{R}_n$, we need to apply McDiarmid's Inequality a second time. Let $\mathcal{G}$ be a family of measurable functions with $g \in \mathcal{G} : \mathcal{X} \to [-M, M]$, Let $\hat{\mathcal{R}}_{n-1,S}\left(\mathcal{G}\right) = n^{-1}\sum_{i=1}^n \mathcal{R}_{S_{-i}}\left(\mathcal{G}\right)$, so that $\mathbb{E}_S\hat{\mathcal{R}}_{n-1,S}\left(\mathcal{G}\right) = \mathcal{R}_{n-1}\left(\mathcal{G}\right)$. Then,

$$
\left|\hat{\mathcal{R}}_{n-1,S_i}\left(\mathcal{G}\right) - \hat{\mathcal{R}}_{n-1,S}\left(\mathcal{G}\right)\right|
$$

$$
= \left|n^{-1}\sum_{k=1}^n \mathbb{E}_\epsilon \sup_{g\in\mathcal{G}} \frac{1}{n-1}\sum_{j=1,j\neq k}^n \epsilon_j g\left(x_{i,j}\right) - n^{-1}\sum_{k=1}^n \mathbb{E}_\epsilon \sup_{g\in\mathcal{G}} \frac{1}{n-1}\sum_{j=1,j\neq k}^n \epsilon_j g\left(x_j\right)\right|
$$

$$
= \frac{1}{n(n-1)}\left|\sum_{k=1}^n \mathbb{E}_\epsilon \left(\sup_{g\in\mathcal{G}}\sum_{j=1,j\neq k}^n \epsilon_j g\left(x_{i,j}\right) - \sup_{g\in\mathcal{G}}\sum_{j=1,j\neq k}^n \epsilon_j g\left(x_j\right)\right)\right|
$$

$$
\leq \frac{1}{n(n-1)}\sum_{k=1}^n \mathbb{E}_\epsilon \sup_{g\in\mathcal{G}}\sum_{j=1,j\neq k}^n \left|\epsilon_j\left(g\left(x_{i,j}\right) - g\left(x_j\right)\right)\right|
$$

$$
= \frac{1}{n(n-1)}\sum_{k=1}^n \mathbb{E}_\epsilon \sup_{g\in\mathcal{G}}\left|g\left(x_i'\right) - g\left(x_i\right)\right| I(k \neq i)
$$

$$
\leq \frac{1}{n(n-1)}\sum_{k=1}^n I(k \neq i)\mathbb{E}_\epsilon 2M = \frac{1}{n(n-1)}\sum_{k=1,k\neq i}^n 2M
$$

$$
= 2Mn^{-1}
$$

where, in the fifth line, we note that the term inside the absolute value is potentially nonzero if and only if it involves $x_i'$, which occurs in each sum over $j$ exactly once, except in the case in which $k \neq i$, in which case it does not occur at all.

$$
\left|\mathcal{R}_{S_i}\left(\mathcal{G}\right) - \mathcal{R}_S\left(\mathcal{G}\right)\right| = \left|\mathbb{E}_\epsilon \sup_{g\in\mathcal{G}} n^{-1}\sum_{j=1}^n \epsilon_j g\left(x_{i,j}\right) - \mathbb{E}_\epsilon \sup_{g\in\mathcal{G}} n^{-1}\sum_{j=1}^n \epsilon_j g\left(x_j\right)\right|
$$

$$
\leq n^{-1}\mathbb{E}_\epsilon \sup_{g\in\mathcal{G}}\sum_{j=1}^n \left|\epsilon_j\left(g\left(x_{i,j}\right) - g\left(x_j\right)\right)\right| = n^{-1}\mathbb{E}_\epsilon \sup_{g\in\mathcal{G}}\left|g\left(x_i'\right) - g\left(x_i\right)\right|
$$

$$
\leq n^{-1}\mathbb{E}_\epsilon 2M
$$

$$
= 2Mn^{-1}
$$

Combining the above results gives,

$$
\left|\left(\hat{\mathcal{R}}_{n-1,S_i}\left(\mathcal{F}_1\right) + \mathcal{R}_{S_i}\left(\mathcal{F}_2\right)\right) - \left(\hat{\mathcal{R}}_{n-1,S}\left(\mathcal{F}_1\right) + \mathcal{R}_S\left(\mathcal{F}_2\right)\right)\right|
$$

$$
= \left|\left(\hat{\mathcal{R}}_{n-1,S_i}\left(\mathcal{F}_1\right) - \hat{\mathcal{R}}_{n-1,S}\left(\mathcal{F}_1\right)\right) + \left(\mathcal{R}_{S_i}\left(\mathcal{F}_2\right) - \mathcal{R}_S\left(\mathcal{F}_2\right)\right)\right|
$$

$$
\leq \left|\hat{\mathcal{R}}_{n-1,S_i}\left(\mathcal{F}_1\right) - \hat{\mathcal{R}}_{n-1,S}\left(\mathcal{F}_1\right)\right| + \left|\mathcal{R}_{S_i}\left(\mathcal{F}_2\right) - \mathcal{R}_S\left(\mathcal{F}_2\right)\right| = 2Mn^{-1} + 2Mn^{-1}
$$

$$
= 4Mn^{-1}
$$

This is the same value we obtained previously, so that the corresponding $\epsilon = 2M\left(\frac{2}{n}\log\frac{2}{\delta}\right)^{\frac{1}{2}}$. Then, McDiarmid's Inequality gives,

$$P\left[\left(\hat{\mathcal{R}}_{n-1,S}\left(\mathcal{F}_1\right)+\mathcal{R}_S\left(\mathcal{F}_2\right)\right)-\left(\mathcal{R}_{n-1}\left(\mathcal{F}_1\right)+\mathcal{R}_n\left(\mathcal{F}_2\right)\right)\geq\epsilon\right]\leq\frac{\delta}{2}$$

$$P\left[\left(\mathcal{R}_{n-1}\left(\mathcal{F}_1\right)+\mathcal{R}_n\left(\mathcal{F}_2\right)\right)-\left(\hat{\mathcal{R}}_{n-1,S}\left(\mathcal{F}_1\right)+\mathcal{R}_S\left(\mathcal{F}_2\right)\right)\geq\epsilon\right]\leq\frac{\delta}{2}$$

so that, with probability $1-\frac{\delta}{2}$, $\mathcal{R}_{n-1}\left(\mathcal{F}_1\right)+\mathcal{R}_n\left(\mathcal{F}_2\right)\leq\hat{\mathcal{R}}_{n-1,S}\left(\mathcal{F}_1\right)+\mathcal{R}_S\left(\mathcal{F}_2\right)+2M\left(\frac{2}{n}\log\frac{2}{\delta}\right)^{\frac{1}{2}}$. Combining this with the previous results shows that, with probability at least $1-\delta$, $\mathbb{E}f\leq\hat{\mathbb{E}}_{S,\lambda}f+2\left(\hat{\mathcal{R}}_{n-1,S}\left(\mathcal{F}_1\right)+\mathcal{R}_S\left(\mathcal{F}_2\right)\right)+(2M+2\cdot 2M)\left(\frac{2}{n}\log\frac{2}{\delta}\right)^{\frac{1}{2}}=\hat{\mathbb{E}}_{S,\lambda}f+2\left(\hat{\mathcal{R}}_{n-1,S}\left(\mathcal{F}_1\right)+\mathcal{R}_S\left(\mathcal{F}_2\right)\right)+6M\left(\frac{2}{n}\log\frac{2}{\delta}\right)^{\frac{1}{2}}$ and $\hat{\mathbb{E}}_{S,\lambda}f\leq\mathbb{E}f+\lambda M_\Lambda+2\left(\hat{\mathcal{R}}_{n-1,S}\left(\mathcal{F}_1\right)+\mathcal{R}_S\left(\mathcal{F}_2\right)\right)+6M\left(\frac{2}{n}\log\frac{2}{\delta}\right)^{\frac{1}{2}}$, yielding the second pair of inequalities.

Finally, if the elements of $\mathcal{F}$ are symmetric, so that $\forall_{f\in\mathcal{F},x,x'\in\mathcal{X}}f\left(x,x'\right)=f\left(x',x\right)$, if $g\in\mathcal{F}_1$ then $\exists_{f\in\mathcal{F},x'\in\mathcal{X}}\forall_{x\in\mathcal{X}}\left(g(x)=f\left(x',x\right)=f\left(x,x'\right)\right)$, so $g\in\mathcal{F}_2$ as well and $\mathcal{F}_1\subseteq\mathcal{F}_2$. Likewise, if $g\in\mathcal{F}_2$ then $\exists_{f\in\mathcal{F},x'\in\mathcal{X}}\forall_{x\in\mathcal{X}}\left(g(x)=f\left(x,x'\right)=f\left(x',x\right)\right)$, so $g\in\mathcal{F}_1$ as well and $\mathcal{F}_2\subseteq\mathcal{F}_1$. Thus, $\mathcal{F}_1=\mathcal{F}_2$.

$\square$

**Corollary 4.** *The inequalities in Lemma 3 can be strengthened to the following:*

$$\mathbb{E}f\leq\frac{1}{n(n-1)}\sum_{i,j=1,i\neq j}^{n}f\left(x_i,x_j\right)+\lambda\Lambda\left[f,\theta_f\right]$$

$$+2\mathbb{E}_X\left(\mathcal{R}_{n-1}\left(\mathcal{F}_{1,X}\right)+\mathcal{R}_n\left(\mathcal{F}_{2,X}\right)\right)+2M\left(\frac{2}{n}\log\frac{1}{\delta}\right)^{\frac{1}{2}}$$

$$\frac{1}{n(n-1)}\sum_{i,j=1,i\neq j}^{n}f\left(x_i,x_j\right)+\lambda\Lambda\left[f,\theta_f\right]\leq\mathbb{E}f+\lambda M_\Lambda$$

$$+2\mathbb{E}_X\left(\mathcal{R}_{n-1}\left(\mathcal{F}_{1,X}\right)+\mathcal{R}_n\left(\mathcal{F}_{2,X}\right)\right)+2M\left(\frac{2}{n}\log\frac{1}{\delta}\right)^{\frac{1}{2}}$$

$$\mathbb{E}f\leq\frac{1}{n(n-1)}\sum_{i,j=1,i\neq j}^{n}f\left(x_i,x_j\right)+\lambda\Lambda\left[f,\theta_f\right]$$

$$+2\mathbb{E}_X\left(\hat{\mathcal{R}}_{n-1,S}\left(\mathcal{F}_{1,X}\right)+\mathcal{R}_S\left(\mathcal{F}_{2,X}\right)\right)+6M\left(\frac{2}{n}\log\frac{2}{\delta}\right)^{\frac{1}{2}}$$

$$\frac{1}{n(n-1)}\sum_{i,j=1,i\neq j}^{n}f\left(x_i,x_j\right)+\lambda\Lambda\left[f,\theta_f\right]\leq\mathbb{E}f+\lambda M_\Lambda$$

$$+2\mathbb{E}_X\left(\hat{\mathcal{R}}_{n-1,S}\left(\mathcal{F}_{1,X}\right)+\mathcal{R}_S\left(\mathcal{F}_{2,X}\right)\right)+6M\left(\frac{2}{n}\log\frac{2}{\delta}\right)^{\frac{1}{2}}$$

$\mathcal{F}_{1,x}=\{g_x\mid\exists_{f\in\mathcal{F}}\forall_{x'\in\mathcal{X}}\left[g_x\left(x'\right)=f\left(x,x'\right)\right]\}$, $\mathcal{F}_{2,x}=\{g_x\mid\exists_{f\in\mathcal{F}}\forall_{x'\in\mathcal{X}}\left[g_x\left(x'\right)=f\left(x',x\right)\right]\}$
*If the elements of $\mathcal{F}$ are symmetric, then $\mathcal{F}_{1,x}=\mathcal{F}_{2,x}$.*

*Proof.* This follows directly from the proof of Lemma 3 in which we show that
$\mathbb{E}_S\phi^{(1)}(S)\leq 2\mathbb{E}_X\mathbb{E}_{S_{-n},\epsilon}\sup_{f\in\mathcal{F}}\frac{1}{n-1}\sum_{i=1}^{n-1}\epsilon_i f\left(X,x_i\right)=2\mathbb{E}_X\mathcal{R}_{n-1}\left(\mathcal{F}_{1,X}\right)$
and $\mathbb{E}_S\phi^{(2)}(S)\leq 2\mathbb{E}_{S,\epsilon}\sup_{f\in\mathcal{F}}\frac{1}{n}\sum_{i=1}^{n}\epsilon_i E_X f\left(x_i,X\right)\leq 2\mathbb{E}_X\mathbb{E}_{S,\epsilon}\sup_{f\in\mathcal{F}}\frac{1}{n}\sum_{i=1}^{n}\epsilon_i f\left(x_i,X\right)$
$=2\mathbb{E}_X\mathcal{R}_n\left(\mathcal{F}_{2,X}\right)$. Using these sharper bounds in the expressions (obtained from McDiarmid's inequality) in Lemma 3 (and using $2\delta$ in place of $\delta$) yields the first pair of equations.

In order to obtain the second pair of expressions, we again need to apply McDiarmid's inequality. Using results from the proof of lemma 3,

$$\left| \mathbb{E}_X \left( \hat{\mathcal{R}}_{n-1,S_i} \left( \mathcal{F}_{1,X} \right) + \mathcal{R}_{S_i} \left( \mathcal{F}_{2,X} \right) \right) - \mathbb{E}_X \left( \hat{\mathcal{R}}_{n-1,S} \left( \mathcal{F}_{1,X} \right) + \mathcal{R}_S \left( \mathcal{F}_{2,X} \right) \right) \right|$$

$$= \left| \mathbb{E}_X \left( \hat{\mathcal{R}}_{n-1,S_i} \left( \mathcal{F}_{1,X} \right) - \hat{\mathcal{R}}_{n-1,S} \left( \mathcal{F}_{1,X} \right) \right) + \mathbb{E}_X \left( \mathcal{R}_{S_i} \left( \mathcal{F}_{2,X} \right) - \mathcal{R}_S \left( \mathcal{F}_{2,X} \right) \right) \right|$$

$$\leq \mathbb{E}_X \left| \hat{\mathcal{R}}_{n-1,S_i} \left( \mathcal{F}_{1,X} \right) - \hat{\mathcal{R}}_{n-1,S} \left( \mathcal{F}_{1,X} \right) \right| + \mathbb{E}_X \left| \mathcal{R}_{S_i} \left( \mathcal{F}_{2,X} \right) - \mathcal{R}_S \left( \mathcal{F}_{2,X} \right) \right|$$

$$= \mathbb{E}_X 2 M n^{-1} + \mathbb{E}_X 2 M n^{-1} = 2 M n^{-1} + 2 M n^{-1}$$

$$= 4 M n^{-1}$$

Since this is the same value of $c_i$ we obtained in lemma 3, McDiarmid's Inequality holds for $\epsilon = 2M \left( \frac{2}{n} \log \frac{2}{\delta} \right)^{\frac{1}{2}}$, so that, with probability at least $1 - \frac{\delta}{2}$, $\mathbb{E}_X \left( \mathcal{R}_{n-1} \left( \mathcal{F}_{1,X} \right) + \mathcal{R}_n \left( \mathcal{F}_{2,X} \right) \right) \leq \mathbb{E}_X \left( \hat{\mathcal{R}}_{n-1,S} \left( \mathcal{F}_{1,X} \right) + \mathcal{R}_S \left( \mathcal{F}_{2,X} \right) \right) + 2M \left( \frac{2}{n} \log \frac{2}{\delta} \right)^{\frac{1}{2}}$. Then, the second pair of expressions is obtained by replacing $\delta$ by $\frac{\delta}{2}$ in the first pair of expressions and combining the result with the above inequality. Since each of these expressions hold with probability at least $1 - \frac{\delta}{2}$, their combination will hold with probability at least $1 - 2 \cdot \frac{\delta}{2} = 1 - \delta$, as claimed.

If the elements of $\mathcal{F}$ is symmetric, so that $\forall_{f \in \mathcal{F}, x, x' \in \mathcal{X}} f(x, x') = f(x', x)$, if $g \in \mathcal{F}_{1,x}$ then $\exists_{f \in \mathcal{F}} \forall_{x' \in \mathcal{X}} \left[ g_x(x') = f(x, x') = f(x', x) \right]$, so $g \in \mathcal{F}_{2,x}$ as well and $\mathcal{F}_{1,x} \subseteq \mathcal{F}_{2,x}$. Likewise, if $g \in \mathcal{F}_{2,x}$ $\exists_{f \in \mathcal{F}} \forall_{x' \in \mathcal{X}} \left[ g_x(x') = f(x', x) = f(x, x') \right]$, so $g \in \mathcal{F}_{1,x}$ as well and $\mathcal{F}_{2,x} \subseteq \mathcal{F}_{1,x}$. Thus, $\mathcal{F}_{1,x} = \mathcal{F}_{2,x}$.

$\square$

**Lemma 5.** *Let* $h \in \mathcal{H} : \mathcal{A} \times \mathcal{W} \times \mathcal{X} \rightarrow [-M, M]$ *such that if* $h \in \mathcal{H}$, $-h \in \mathcal{H}$, $\mathcal{Y} \subseteq [-M, M]$, $k : (\mathcal{A} \times \mathcal{Z} \times \mathcal{X})^2 \rightarrow [-M_k, M_k]$, $\forall_{a,a' \in \mathcal{A}, x, x' \in \mathcal{X}, z, z' \in \mathcal{Z}} \left( k \left( (a, x, z), (a', x', z') \right) = k \left( (a', x', z'), (a, x, z) \right) \right)$, *and* $\Xi = (A, W, X, Y, Z)$. *Additionally, let* $f_{h,k} \left( (a, w, x, y, z), (a', w', x', y', z') \right) = (y - h(a, w, x)) \times (y' - h(a', w', x')) k \left( (a, x, z), (a', x', z') \right)$ *Then,*

$$\mathbb{E}_\Xi \mathcal{R}_n \left( \mathcal{F}_{1,\Xi} \right) = \mathbb{E}_\Xi \mathcal{R}_n \left( \mathcal{F}_{2,\Xi} \right)$$

$$\leq 2M \mathbb{E}_{A,X,Z,S,\epsilon} \sup_{h \in \mathcal{H}} \frac{1}{n} \sum_{i=1}^n \epsilon_i h(a_i, w_i, x_i) k \left( (a_i, x_i, z_i), (A, X, Z) \right)$$

$$+ (2 \log 2)^{\frac{1}{2}} M^2 M_k n^{-\frac{1}{2}}$$

$$= 2M \mathbb{E}_{A,X,Z} \mathcal{R}_n \left( \mathcal{F}'_{A,X,Z} \right) + (2 \log 2)^{\frac{1}{2}} M^2 M_k n^{-\frac{1}{2}}$$

$$\leq 2M \mathbb{E}_{S,\epsilon} \sup_{h \in \mathcal{H}, a \in \mathcal{A}, x \in \mathcal{X}, z \in \mathcal{Z}} \frac{1}{n} \sum_{i=1}^n \epsilon_i h(a_i, w_i, x_i) k \left( (a_i, x_i, z_i), (a, x, z) \right)$$

$$+ (2 \log 2)^{\frac{1}{2}} M^2 M_k n^{-\frac{1}{2}}$$

$$= 2M \mathcal{R}_n \left( \mathcal{F}' \right) + (2 \log 2)^{\frac{1}{2}} M^2 M_k n^{-\frac{1}{2}}$$

$$\mathcal{F}'_{a,x,z} = \left\{ f_{a,x,z} \mid \exists_{h \in \mathcal{H}} \forall_{a' \in \mathcal{A}, x' \in \mathcal{X}, z' \in \mathcal{Z}} f_{a,x,z} \left( a', w', x', z' \right) = h \left( a', w', x' \right) k \left( (a', x', z'), (a, x, z) \right) \right\}$$

$$\mathcal{F}' = \left\{ f \mid \exists_{h \in \mathcal{H}, a \in \mathcal{A}, x \in \mathcal{X}, z \in \mathcal{Z}} \forall_{a' \in \mathcal{A}, x' \in \mathcal{X}, z' \in \mathcal{Z}} f \left( a', w', x', z' \right) = h \left( a', w', x' \right) k \left( (a', x', z'), (a, x, z) \right) \right\}$$

*Proof.* Let $\mathcal{F} = \{ g \mid \exists_{h \in \mathcal{H}} (g = f_{h,k}) \}$. Since, for all $h \in \mathcal{H}$, $f_{h,k}$ is manifestly symmetric, we have $\mathcal{F}_1 = \mathcal{F}_2$ and $\mathcal{F}_{1,\xi} = \mathcal{F}_{2,\xi}$, where these classes are defined in lemma 3 and corollary 4, respectively. We now compute $\mathbb{E}_\Xi \mathcal{R}_n \left( \mathcal{F}_{1,\Xi} \right) = \mathbb{E}_\Xi \mathcal{R}_n \left( \mathcal{F}_{2,\Xi} \right)$.

$$\mathbb{E}_\Xi \mathcal{R}_n\left(\mathcal{F}_{1,\Xi}\right) = \mathbb{E}_\Xi \mathcal{R}_n\left(\mathcal{F}_{2,\Xi}\right) = \mathbb{E}_\Xi \mathbb{E}_{S,\epsilon} \sup_{f \in \mathcal{F}} \frac{1}{n} \sum_{i=1}^{n} \epsilon_i f\left(x_i, \Xi\right)$$

$$= \mathbb{E}_{A,W,X,Y,Z,S,\epsilon} \sup_{f \in \mathcal{F}} \frac{1}{n} \sum_{i=1}^{n} \epsilon_i f\left(\left(a_i, w_i, x_i, y_i, z_i\right), \left(A, W, X, Y, Z\right)\right)$$

$$= \mathbb{E}_{A,W,X,Y,Z,S,\epsilon} \sup_{h \in \mathcal{H}} \frac{1}{n} \sum_{i=1}^{n} \epsilon_i \left(y_i - h\left(a_i, w_i, x_i\right)\right)\left(Y - h\left(A, W, X\right)\right)$$
$$\times k\left(\left(a_i, x_i, z_i\right), \left(A, X, Z\right)\right)$$

$$= \mathbb{E}_{A,W,X,Y,Z,S,\epsilon} \sup_{h \in \mathcal{H}} \left(Y - h\left(A, W, X\right)\right) \cdot \frac{1}{n} \sum_{i=1}^{n} \epsilon_i \left(y_i - h\left(a_i, w_i, x_i\right)\right)$$
$$\times k\left(\left(a_i, x_i, z_i\right), \left(A, X, Z\right)\right)$$

$$\leq \mathbb{E}_{A,W,X,Y,Z,S,\epsilon} \sup_{h \in \mathcal{H}} Y \cdot \frac{1}{n} \sum_{i=1}^{n} \epsilon_i y_i k\left(\left(a_i, x_i, z_i\right), \left(A, X, Z\right)\right)$$

$$+ \mathbb{E}_{A,W,X,Y,Z,S,\epsilon} \sup_{h \in \mathcal{H}} -Y \cdot \frac{1}{n} \sum_{i=1}^{n} \epsilon_i h\left(a_i, w_i, x_i\right) k\left(\left(a_i, x_i, z_i\right), \left(A, X, Z\right)\right)$$

$$+ \mathbb{E}_{A,W,X,Y,Z,S,\epsilon} \sup_{h \in \mathcal{H}} -h\left(A, W, X\right) \cdot \frac{1}{n} \sum_{i=1}^{n} \epsilon_i y_i k\left(\left(a_i, x_i, z_i\right), \left(A, X, Z\right)\right)$$

$$+ \mathbb{E}_{A,W,X,Y,Z,S,\epsilon} \sup_{h \in \mathcal{H}} h\left(A, W, X\right) \cdot \frac{1}{n} \sum_{i=1}^{n} \epsilon_i h\left(a_i, w_i, x_i\right) k\left(\left(a_i, x_i, z_i\right), \left(A, X, Z\right)\right)$$

We analyze each of these four terms separately.

$$\mathbb{E}_{A,W,X,Y,Z,S,\epsilon} \sup_{h \in \mathcal{H}} Y \cdot \frac{1}{n} \sum_{i=1}^{n} \epsilon_i y_i k\left(\left(a_i, x_i, z_i\right), \left(A, X, Z\right)\right)$$

$$= \mathbb{E}_{A,X,Y,Z,S,\epsilon} Y \cdot \frac{1}{n} \sum_{i=1}^{n} \epsilon_i y_i k\left(\left(a_i, x_i, z_i\right), \left(A, X, Z\right)\right)$$

$$= \mathbb{E}_{A,X,Y,Z,S} Y \cdot \frac{1}{n} \sum_{i=1}^{n} \mathbb{E}_\epsilon \epsilon_i y_i k\left(\left(a_i, x_i, z_i\right), \left(A, X, Z\right)\right)$$

$$= \mathbb{E}_{A,X,Y,Z,S} Y \cdot \frac{1}{n} \sum_{i=1}^{n} 0 \cdot y_i k\left(\left(a_i, x_i, z_i\right), \left(A, X, Z\right)\right) = \mathbb{E}_Y Y \cdot 0$$

$$= 0$$

$$\mathbb{E}_{A,W,X,Y,Z,S,\epsilon} \sup_{h \in \mathcal{H}} -Y \cdot \frac{1}{n} \sum_{i=1}^{n} \epsilon_i h\left(a_i, w_i, x_i\right) k\left(\left(a_i, x_i, z_i\right), \left(A, X, Z\right)\right)$$

$$\leq \mathbb{E}_{A,X,Y,Z,S,\epsilon} \sup_{h \in \mathcal{H}} |Y| \left|\frac{1}{n} \sum_{i=1}^{n} \epsilon_i h\left(a_i, w_i, x_i\right) k\left(\left(a_i, x_i, z_i\right), \left(A, X, Z\right)\right)\right|$$

$$\leq \mathbb{E}_{A,X,Z,S,\epsilon} \sup_{h \in \mathcal{H}} M \left|\frac{1}{n} \sum_{i=1}^{n} \epsilon_i h\left(a_i, w_i, x_i\right) k\left(\left(a_i, x_i, z_i\right), \left(A, X, Z\right)\right)\right|$$

$$= M \mathbb{E}_{A,X,Z,S,\epsilon} \sup_{h \in \mathcal{H}} \frac{1}{n} \sum_{i=1}^{n} \epsilon_i h\left(a_i, w_i, x_i\right) k\left(\left(a_i, x_i, z_i\right), \left(A, X, Z\right)\right)$$

where the final equality is due to the fact that $h \in \mathcal{H}$ if and only if $-h \in \mathcal{H}$, so that the supremum of the sum will be equal to the supremum of its absolute value.

$$\mathbb{E}_{A,W,X,Y,Z,S,\epsilon} \sup_{h \in \mathcal{H}} h(A,W,X) \cdot \frac{1}{n} \sum_{i=1}^{n} \epsilon_i h(a_i, w_i, x_i) k((a_i, x_i, z_i), (A, X, Z))$$

$$\leq \mathbb{E}_{A,W,X,Z,S,\epsilon} \sup_{h,h' \in \mathcal{H}} h'(A,W,X) \cdot \frac{1}{n} \sum_{i=1}^{n} \epsilon_i h(a_i, w_i, x_i) k((a_i, x_i, z_i), (A, X, Z))$$

$$\leq \mathbb{E}_{A,W,X,Z,S,\epsilon} \sup_{h,h' \in \mathcal{H}} |h'(A,W,X)| \left| \frac{1}{n} \sum_{i=1}^{n} \epsilon_i h(a_i, w_i, x_i) k((a_i, x_i, z_i), (A, X, Z)) \right|$$

$$\leq \mathbb{E}_{A,X,Z,S,\epsilon} \sup_{h \in \mathcal{H}} M \left| \frac{1}{n} \sum_{i=1}^{n} \epsilon_i h(a_i, w_i, x_i) k((a_i, x_i, z_i), (A, X, Z)) \right|$$

$$= M \mathbb{E}_{A,X,Z,S,\epsilon} \sup_{h \in \mathcal{H}} \frac{1}{n} \sum_{i=1}^{n} \epsilon_i h(a_i, w_i, x_i) k((a_i, x_i, z_i), (A, X, Z))$$

where the final equality follows, as above, because $h \in \mathcal{H}$ if and only if $-h \in \mathcal{H}$.

$$\mathbb{E}_{A,W,X,Y,Z,S,\epsilon} \sup_{h \in \mathcal{H}} -h(A,W,X) \cdot \frac{1}{n} \sum_{i=1}^{n} \epsilon_i y_i k((a_i, x_i, z_i), (A, X, Z))$$

$$\leq \mathbb{E}_{A,W,X,Z,S,\epsilon} \sup_{h \in \mathcal{H}} |h(A,W,X)| \left| \frac{1}{n} \sum_{i=1}^{n} \epsilon_i y_i k((a_i, x_i, z_i), (A, X, Z)) \right|$$

$$\leq \mathbb{E}_{A,X,Z,S,\epsilon} M \left| \frac{1}{n} \sum_{i=1}^{n} \epsilon_i y_i k((a_i, x_i, z_i), (A, X, Z)) \right|$$

$$= M \mathbb{E}_{A,X,Z,S,\epsilon} \sup_{h \in \{-1,1\}} h \cdot \frac{1}{n} \sum_{i=1}^{n} \epsilon_i y_i k((a_i, x_i, z_i), (A, X, Z))$$

$$= M \mathbb{E}_{A,X,Z,S} \mathbb{E}_{\epsilon} \sup_{h \in \{-1,1\}} \frac{1}{n} \sum_{i=1}^{n} \epsilon_i h y_i k((a_i, x_i, z_i), (A, X, Z))$$

$$\leq M \cdot \mathbb{E}_{A,X,Z,S} M M_k (2 \log 2)^{\frac{1}{2}} n^{-\frac{1}{2}} = M \cdot M M_k (2 \log 2)^{\frac{1}{2}} n^{-\frac{1}{2}}$$

$$= (2 \log 2)^{\frac{1}{2}} M^2 M_k n^{-\frac{1}{2}}$$

where the final inequality follows from Massart's Finite Lemma using $|yk| \leq M M_k$. Combining these results gives,

$$\mathbb{E}_\Xi \mathcal{R}_n \left(\mathcal{F}_{1,\Xi}\right) = \mathbb{E}_\Xi \mathcal{R}_n \left(\mathcal{F}_{2,\Xi}\right)$$

$$\leq 0 + (2\log 2)^{\frac{1}{2}} M^2 M_k n^{-\frac{1}{2}}$$

$$+ M\mathbb{E}_{A,X,Z,S,\epsilon} \sup_{h\in\mathcal{H}} \frac{1}{n} \sum_{i=1}^n \epsilon_i h\left(a_i, w_i, x_i\right) k\left(\left(a_i, x_i, z_i\right), (A, X, Z)\right)$$

$$+ M\mathbb{E}_{A,X,Z,S,\epsilon} \sup_{h\in\mathcal{H}} \frac{1}{n} \sum_{i=1}^n \epsilon_i h\left(a_i, w_i, x_i\right) k\left(\left(a_i, x_i, z_i\right), (A, X, Z)\right)$$

$$= 2M\mathbb{E}_{A,X,Z,S,\epsilon} \sup_{h\in\mathcal{H}} \frac{1}{n} \sum_{i=1}^n \epsilon_i h\left(a_i, w_i, x_i\right) k\left(\left(a_i, x_i, z_i\right), (A, X, Z)\right)$$

$$+ (2\log 2)^{\frac{1}{2}} M^2 M_k n^{-\frac{1}{2}}$$

$$= 2M\mathbb{E}_{A,X,Z}\mathbb{E}_{S,\epsilon} \sup_{h\in\mathcal{H}} \frac{1}{n} \sum_{i=1}^n \epsilon_i h\left(a_i, w_i, x_i\right) k\left(\left(a_i, x_i, z_i\right), (A, X, Z)\right)$$

$$+ (2\log 2)^{\frac{1}{2}} M^2 M_k n^{-\frac{1}{2}}$$

$$= 2M\mathbb{E}_{A,X,Z}\mathbb{E}_{S,\epsilon} \sup_{f_{A,X,Z}\in\mathcal{F}'_{A,X,Z}} \frac{1}{n} \sum_{i=1}^n \epsilon_i f_{A,X,Z}\left(a_i, w_i, x_i, z_i\right)$$

$$+ (2\log 2)^{\frac{1}{2}} M^2 M_k n^{-\frac{1}{2}}$$

$$= 2M\mathbb{E}_{A,X,Z}\mathcal{R}_n\left(\mathcal{F}'_{A,X,Z}\right) + (2\log 2)^{\frac{1}{2}} M^2 M_k n^{-\frac{1}{2}}$$

$$\leq 2M\mathbb{E}_{S,\epsilon} \sup_{h\in\mathcal{H}, a\in\mathcal{A}, x\in\mathcal{X}, z\in\mathcal{Z}} \frac{1}{n} \sum_{i=1}^n \epsilon_i h\left(a_i, w_i, x_i\right) k\left(\left(a_i, x_i, z_i\right), (a, x, z)\right)$$

$$+ (2\log 2)^{\frac{1}{2}} M^2 M_k n^{-\frac{1}{2}}$$

$$= 2M\mathbb{E}_{S,\epsilon} \sup_{f\in\mathcal{F}'} \frac{1}{n} \sum_{i=1}^n \epsilon_i f\left(a_i, w_i, x_i, z_i\right) + (2\log 2)^{\frac{1}{2}} M^2 M_k n^{-\frac{1}{2}}$$

$$= 2M\mathcal{R}_n\left(\mathcal{F}'\right) + (2\log 2)^{\frac{1}{2}} M^2 M_k n^{-\frac{1}{2}}$$

$$\mathcal{F}'_{a,x,z} = \left\{ f_{a,x,z} \mid \exists_{h\in\mathcal{H}} \forall_{a'\in\mathcal{A}, x'\in\mathcal{X}, z'\in\mathcal{Z}} f_{a,x,z}\left(a', w', x', z'\right) = h\left(a', w', x'\right) k\left(\left(a', x', z'\right), (a, x, z)\right) \right\}$$

$$\mathcal{F}' = \left\{ f \mid \exists_{h\in\mathcal{H}, a\in\mathcal{A}, x\in\mathcal{X}, z\in\mathcal{Z}} \forall_{a'\in\mathcal{A}, x'\in\mathcal{X}, z'\in\mathcal{Z}} f\left(a', w', x', z'\right) = h\left(a', w', x'\right) k\left(\left(a', x', z'\right), (a, x, z)\right) \right\}$$

$$\square$$

**Lemma 6.** *Let $\mathcal{X}$ be a measurable space, $\mu$ be a $\sigma$-finite measure on $\mathcal{X}$, $\mathcal{F}_0$ be a collection of $\mu$-measurable functions, with $f\in\mathcal{F}_0 : \mathcal{X} \to \mathbb{R}$, $k : \mathcal{X}^2 \to \mathbb{R}$ be symmetric and measurable with respect to the product measure $(\mu \times \mu)$, and $\mathcal{F}$ be the quotient space of $\mathcal{F}_0$ in which functions are identified if they are equal $\mu$-almost everywhere. For $f, g \in \mathcal{F}_0$, define the bilinear form $\langle f, g \rangle_k = \int f(x)k(x, y)g(y)d\mu(x, y)$, where $\mu$ is the product measure. Then, $\langle \rangle_k$ is an inner product on $\mathcal{F}$ if and only if $k$ is an Integrally Strictly Positive Definite (ISPD) kernel, so that, for all $f \in \mathcal{F}_0$ such that $f \neq 0$ $\mu$-almost everywhere, $\int f(x)k(x, y)f(y)d\mu(x, y) > 0$. Further, if $k$ is ISPD, then it defines a metric on $\mathcal{F}$ by $d_k(f, g) = \|f - g\|_k = \langle f - g, f - g \rangle_k^{\frac{1}{2}}$.*

*Proof.* For $f, g \in \mathcal{F}_0$,

$$\langle cf + g, h \rangle_k = \int (cf + g)(x)k(x,y)h(y)d\mu(x,y)$$

$$= c \int f(x)k(x,y)h(y)d\mu(x,y) + \int g(x)k(x,y)h(y)d\mu(x,y)$$

$$= c\langle f, h \rangle_k + \langle g, h \rangle_k$$

$$\langle f, g \rangle_k = \int f(x)k(x,y)g(y)d\mu(x,y) = \int g(y)k(y,x)f(x)d\mu(x,y)$$

$$= \int g(x)k(x,y)f(y)d\mu(x,y)$$

$$= \langle g, f \rangle_k$$

so $\langle \rangle_k$ is a bilinear form, as claimed.

To see that $\langle \rangle_k$ is well defined on $\mathcal{F}$. Note, that, if $f = f'$ a.e., then

$$|\langle f, g \rangle_k - \langle f', g \rangle_k| = \left| \int f(x)k(x,y)g(y)d\mu(x,y) - \int f'(x)k(x,y)g(y)d\mu(x,y) \right|$$

$$\leq \int |f(x) - f'(x)| \, |k(x,y)| \, |g(y)| \, d\mu(x,y)$$

$$= \int |(f - f')(x)| \, |k(x,y)| \, |g(y)| \, d\mu(x,y)$$

$$= \int \int |(f - f')(x)| \, |k(x,y)| \, |g(y)| \, d\mu(x)d\mu(y) = \int 0 d\mu(y)$$

$$= 0$$

where, in the fourth line, we use Tonelli's Theorem and the fact that $f - f' = 0$ a.e., so $|f' - f| \, |k| |g| = 0$ $\mu_x$-a.e. and, thus, the inner integral is 0. Thus, if $f = f'$ a.e., $\langle f, g \rangle_k = \langle f', g \rangle_k$. By the symmetry of the bilinear form, if $g = g'$ a.e. $\langle f', g \rangle_k = \langle f', g' \rangle_k$, so that, if $f = f', g = g'$ a.e. $\langle f, g \rangle_k = \langle f', g' \rangle_k$, so $\langle \rangle_k$ is well defined on $\mathcal{F}$.

If $k$ is also ISPD, then, for $f \neq 0$ $\mu$-almost everywhere, $\langle f, f \rangle_k = \int f(x)k(x,y)f(y)d\mu(x,y) > 0$, so that, combined with the above results, $\langle \rangle_k$ is an inner product on $\mathcal{F}$. Conversely, if $\langle \rangle_k$ is an inner product, then, for $f \neq 0$ $\mu$-almost everywhere, $\langle f, f \rangle_k > 0$, so $k$ is ISPD, by definition.

Since $\langle \rangle_k$ is an inner product, it defines a norm $\| \|_k$ on $\mathcal{F}$ by $\|f\|_k = \langle f, f \rangle_k^{\frac{1}{2}}$. Let $d_k(f, g) = \|f - g\|_k$. Since $\| \|_k$ is a norm, $d_k(f, g) = \|f - g\|_k = |-1| \|g - f\|_k = \|g - f\|_k = d_k(g, f)$ and, if $f \neq g$ a.e., then, $d_k(f, g) = \|f - g\|_k > 0$, while $d_k(f, f) = \|f - f\|_k = \|0\|_k = 0$. Finally, $d_k(f, h) = \|f - h\|_k = \|f - g + g - h\|_k \leq \|f - g\|_k + \|g - h\|_k = d_k(f, g) + d_k(g, h)$ by the subadditivity of the norm, so that the triangle inequality holds and $d_k$ is a metric on $\mathcal{F}$, as claimed.

$\square$

**Theorem 1.** *Let $\tilde{h}_k$ minimize $R_k(h)$ and $\hat{h}_{k,U,\lambda,n}$ minimize $\hat{R}_{k,U,\lambda,n}(h)$ for $h \in \mathcal{H}$, $k : (\mathcal{A} \times \mathcal{X} \times \mathcal{Z})^2 \to [-M_k, M_k]$, $\Lambda : \mathcal{H} \times \Theta_h \to [-0, M_\lambda]$, and let $h^* : \mathcal{A} \times \mathcal{W} \times \mathcal{X} \to \mathbb{R}$ satisfy $\mathbb{E}[Y - h^*(A, W, X)|A, X, Z] = 0$ $\mathrm{P}_{A,X,Z}$-almost surely, where*

$$R_k(h) = \mathbb{E}\left[(Y - h(A, W, X))(Y' - h(A', W', X'))k((A, X, Z), (A', X', Z'))\right]$$

$$\hat{R}_{k,U,\lambda,n}(h) = \frac{1}{n(n-1)} \sum_{i,j=1, i\neq j}^{n} [(y_i - h(a_i, w_i, x_i))(y_j - h(a_j, w_j, x_j))$$

$$\times \, k((a_i, x_i, z_i), (a_j, x_j, z_j))] + \lambda \Lambda[h, \theta_h]$$

*Also let,*

$$d_k^2 \left(h, h'\right) = \mathbb{E}\left[\left(h\left(A, W, X\right) - h'\left(A, W, X\right)\right)\left(h\left(A', W', X'\right) - h'\left(A', W', X'\right)\right)\right.$$
$$\left. \times\, k\left(\left(A, X, Z\right), \left(A', X', Z'\right)\right)\right]$$

*Then, $d_k^2 \left(h^*, h\right) = R_k(h)$ and, with probability at least $1 - \delta$,*

$$d_k^2 \left(h^*, \hat{h}_{k, U, \lambda, n}\right) \leq d_k^2 \left(h^*, \tilde{h}_k\right) + \lambda M_\lambda + 8M \mathbb{E}_{A, X, Z}\left(\mathcal{R}_{n-1}\left(\mathcal{F}'_{A, X, Z}\right) + \mathcal{R}_n\left(\mathcal{F}'_{A, X, Z}\right)\right)$$
$$+ 16M^2 M_k \left(\frac{2}{n}\log\frac{2}{\delta}\right)^{\frac{1}{2}} + 10\left(2\log 2\right)^{\frac{1}{2}} M^2 M_k n^{-\frac{1}{2}}$$
$$\leq d_k^2 \left(h^*, \tilde{h}_k\right) + \lambda M_\lambda + 8M\left(\mathcal{R}_{n-1}\left(\mathcal{F}'\right) + \mathcal{R}_n\left(\mathcal{F}'\right)\right)$$
$$+ 16M^2 M_k \left(\frac{2}{n}\log\frac{2}{\delta}\right)^{\frac{1}{2}} + 10\left(2\log 2\right)^{\frac{1}{2}} M^2 M_k n^{-\frac{1}{2}}$$

*Further, if Assumption 5 holds, so $k$ is ISPD, then $d_k$ is a metric on $L^2_{\mathcal{A}\mathcal{X}\mathcal{Z}}$ and, if the right hand side of the inequality goes to zero as $n$ goes to infinity,*
$$d_k \left(\mathbb{E}\left[h^* | A, X, Z\right] - \mathbb{E}\left[\hat{h}_{k, \lambda, n} \Big| A, X, Z\right]\right) \xrightarrow{\text{P}} 0 \text{ so } \mathbb{E}\left[\hat{h}_{k, \lambda, n} \Big| A, X, Z\right] \xrightarrow{\text{P}} \mathbb{E}\left[h^* | A, X, Z\right]$$
*in $d_k$. Also,* $\left\| \mathbb{E}\left[h^* | A, X, Z\right] - \mathbb{E}\left[\hat{h}_{k, \lambda, n} \Big| A, X, Z\right]\right\|_{\mathrm{P}_{A, X, Z}} \xrightarrow{\text{P}} 0 \text{ so } \mathbb{E}\left[\hat{h}_{k, \lambda, n} \Big| A, X, Z\right] \xrightarrow{\text{P}}$
$\mathbb{E}\left[h^* | A, X, Z\right]$ *in $L^2\left(\mathrm{P}_{\mathcal{A}, \mathcal{X}, \mathcal{Z}}\right)$-norm.*

$$\mathcal{F}'_{a, x, z} = \left\{ f_{a, x, z} \mid \exists_{h \in \mathcal{H}} \forall_{a' \in \mathcal{A}, x' \in \mathcal{X}, z' \in \mathcal{Z}} f_{a, x, z}\left(a', w', x', z'\right) = h\left(a', w', x'\right) k\left(\left(a', x', z'\right), \left(a, x, z\right)\right)\right\}$$
$$\mathcal{F}' = \left\{ f \mid \exists_{h \in \mathcal{H}, a \in \mathcal{A}, x \in \mathcal{X}, z \in \mathcal{Z}} \forall_{a' \in \mathcal{A}, x' \in \mathcal{X}, z' \in \mathcal{Z}} f\left(a', w', x', z'\right) = h\left(a', w', x'\right) k\left(\left(a', x', z'\right), \left(a, x, z\right)\right)\right\}$$

**Proof.** Let $\Xi = \{A, W, X, Y, Z\}$. Since $\tilde{h}_k$ minimizes $R_k(h)$ and $\hat{h}_{k, U, \lambda, n}$ minimizes $\hat{R}_{k, U, \lambda, n}(h)$ for $h \in \mathcal{H}$, $\hat{R}_{k, U, \lambda, n}\left(\hat{h}_{k, U, \lambda, n}\right) \leq \hat{R}_{k, U, \lambda, n}\left(\tilde{h}_k\right)$.

Taking $f\left(\left(a, w, x, y, z\right), \left(a', w', x', y', z'\right)\right) = \left(y - h\left(a, w, x\right)\right)\left(y' - h\left(a', w', x'\right)\right)$
$k\left(\left(a, x, z\right), \left(a', x', z'\right)\right)$, noting that $|f| \leq \left(M + M\right)^2 \cdot M_k = \left(2M\right)^2 M_k = 4M^2 M_k$, and applying lemma 3 and corollary 4 to $R_k(h) = \mathbb{E}f$ and $\hat{R}_{k, U, \lambda, n}(h) = \hat{E}_{S, \lambda}f$ tells us that, with probability at least $1 - \frac{\delta}{2}$,

$$R_k(h) \leq \hat{R}_{k, U, \lambda, n}(h) + 2\mathbb{E}_\Xi\left(\mathcal{R}_{n-1}\left(\mathcal{F}_{1, \Xi}\right) + \mathcal{R}_n\left(\mathcal{F}_{2, \Xi}\right)\right) + 8M^2 M_k \left(\frac{2}{n}\log\frac{2}{\delta}\right)^{\frac{1}{2}}$$

$$\hat{R}_{k, U, \lambda, n}(h) \leq R_k(h) + \lambda M_\lambda + 2\mathbb{E}_\Xi\left(\mathcal{R}_{n-1}\left(\mathcal{F}_{1, \Xi}\right) + \mathcal{R}_n\left(\mathcal{F}_{2, \Xi}\right)\right) + 8M^2 M_k \left(\frac{2}{n}\log\frac{2}{\delta}\right)^{\frac{1}{2}}$$

so, with probability at least $1 - \delta$,

$$R_k\left(\hat{h}_{k, U, \lambda, n}\right) \leq \hat{R}_{k, U, \lambda, n}\left(\hat{h}_{k, U, \lambda, n}\right) + 2\mathbb{E}_\Xi\left(\mathcal{R}_{n-1}\left(\mathcal{F}_{1, \Xi}\right) + \mathcal{R}_n\left(\mathcal{F}_{2, \Xi}\right)\right) + 8M^2 M_k \left(\frac{2}{n}\log\frac{2}{\delta}\right)^{\frac{1}{2}}$$

$$\leq \hat{R}_{k, U, \lambda, n}\left(\tilde{h}_k\right) + 2\mathbb{E}_\Xi\left(\mathcal{R}_{n-1}\left(\mathcal{F}_{1, \Xi}\right) + \mathcal{R}_n\left(\mathcal{F}_{2, \Xi}\right)\right) + 8M^2 M_k \left(\frac{2}{n}\log\frac{2}{\delta}\right)^{\frac{1}{2}}$$

$$\leq R_k\left(\tilde{h}_k\right) + \lambda M_\lambda + 4\mathbb{E}_\Xi\left(\mathcal{R}_{n-1}\left(\mathcal{F}_{1, \Xi}\right) + \mathcal{R}_n\left(\mathcal{F}_{2, \Xi}\right)\right) + 16M^2 M_k \left(\frac{2}{n}\log\frac{2}{\delta}\right)^{\frac{1}{2}}$$

Applying lemma 5 yields,

$$R_k\left(\hat{h}_{k,U,\lambda,n}\right) \le R_k\left(\tilde{h}_k\right) + \lambda M_\lambda + 4\left(2M\mathbb{E}_{A,X,Z}\mathcal{R}_{n-1}\left(\mathcal{F}'_{A,X,Z}\right) + 2M\mathbb{E}_{A,X,Z}\mathcal{R}_n\left(\mathcal{F}'_{A,X,Z}\right)\right)$$

$$+ 4\left((2\log 2)^{\frac{1}{2}} M^2 M_k(n-1)^{-\frac{1}{2}} + (2\log 2)^{\frac{1}{2}} M^2 M_k n^{-\frac{1}{2}}\right)$$

$$+ 16M^2 M_k\left(\frac{2}{n}\log\frac{2}{\delta}\right)^{\frac{1}{2}}$$

$$\le R_k\left(\tilde{h}_k\right) + \lambda M_\lambda + 8M\mathbb{E}_{A,X,Z}\left(\mathcal{R}_{n-1}\left(\mathcal{F}'_{A,X,Z}\right) + \mathcal{R}_n\left(\mathcal{F}'_{A,X,Z}\right)\right)$$

$$+ 4(2\log 2)^{\frac{1}{2}} M^2 M_k n^{-\frac{1}{2}}\left(\left(\frac{n}{n-1}\right)^{\frac{1}{2}} + 1\right) + 16M^2 M_k\left(\frac{2}{n}\log\frac{2}{\delta}\right)^{\frac{1}{2}}$$

$$\le R_k\left(\tilde{h}_k\right) + \lambda M_\lambda + 8M\mathbb{E}_{A,X,Z}\left(\mathcal{R}_{n-1}\left(\mathcal{F}'_{A,X,Z}\right) + \mathcal{R}_n\left(\mathcal{F}'_{A,X,Z}\right)\right)$$

$$+ 16M^2 M_k\left(\frac{2}{n}\log\frac{2}{\delta}\right)^{\frac{1}{2}} + 4(2\log 2)^{\frac{1}{2}} M^2 M_k n^{-\frac{1}{2}} \cdot \frac{5}{2}$$

$$= R_k\left(\tilde{h}_k\right) + \lambda M_\lambda + 8M\mathbb{E}_{A,X,Z}\left(\mathcal{R}_{n-1}\left(\mathcal{F}'_{A,X,Z}\right) + \mathcal{R}_n\left(\mathcal{F}'_{A,X,Z}\right)\right)$$

$$+ 16M^2 M_k\left(\frac{2}{n}\log\frac{2}{\delta}\right)^{\frac{1}{2}} + 10(2\log 2)^{\frac{1}{2}} M^2 M_k n^{-\frac{1}{2}}$$

$$\le R_k\left(\tilde{h}_k\right) + \lambda M_\lambda + 8M\left(\mathcal{R}_{n-1}\left(\mathcal{F}'\right) + \mathcal{R}_n\left(\mathcal{F}'\right)\right) + 16M^2 M_k\left(\frac{2}{n}\log\frac{2}{\delta}\right)^{\frac{1}{2}}$$

$$+ 10(2\log 2)^{\frac{1}{2}} M^2 M_k n^{-\frac{1}{2}}$$

By assumption, $h^*$ satisfies $\mathbb{E}\left[Y - h^*(A,X,X)|A,X,Z\right] = 0$ $P_{A,X,Z}$-almost surely, so that,

$$\mathbb{E}\left[Y - h(A,W,X)|A,X,Z\right] = \mathbb{E}\left[Y - h^*(A,W,X) + h^*(A,W,X) - h(A,W,X)|A,X,Z\right]$$
$$= \mathbb{E}\left[Y - h^*(A,W,X)|A,X,Z\right] + \mathbb{E}\left[h^*(A,W,X) - h(A,W,X)|A,X,Z\right]$$
$$= 0 + \mathbb{E}\left[h^*(A,W,X) - h(A,W,X)|A,X,Z\right]$$
$$= \mathbb{E}\left[h^*(A,W,X) - h(A,W,X)|A,X,Z\right]$$

$P_{A,X,Z}$-almost surely. Then,

$$R_k(h) = \mathbb{E}\left[(Y - h(A,W,X))(Y' - h(A',W',X')) k((A,X,Z),(A',X',Z'))\right]$$
$$= \mathbb{E}\left[\mathbb{E}\left[(Y - h(A,W,X))(Y' - h(A',W',X'))\right.\right.$$
$$\left.\left. \times\ k((A,X,Z),(A',X',Z'))|(A,X,Z),(A',X',Z')\right]\right]$$
$$= \mathbb{E}\left[\mathbb{E}\left[Y - h(A,W,X)|A,X,Z\right]\right.$$
$$\times\ \mathbb{E}\left[Y' - h(A',W',X')|A',X',Z'\right]$$
$$\left. \times\ k((A,X,Z),(A',X',Z'))\right]$$
$$= \mathbb{E}\left[\mathbb{E}\left[h^*(A,W,X) - h(A,W,X)|A,X,Z\right]\right.$$
$$\times\ \mathbb{E}\left[h^*(A',W',X') - h(A',W',X')|A',X',Z'\right]$$
$$\left. \times\ k((A,X,Z),(A',X',Z'))\right]$$
$$= \mathbb{E}\left[\mathbb{E}\left[(h^*(A,W,X) - h(A,W,X))(h^*(A',W',X') - h(A',W',X'))\right.\right.$$
$$\left.\left. \times\ k((A,X,Z),(A',X',Z'))|(A,X,Z),(A',X',Z')\right]\right]$$
$$= \mathbb{E}\left[(h^*(A,W,X) - h(A,W,X))(h^*(A',W',X') - h(A',W',X'))\right.$$
$$\left. \times\ k((A,X,Z),(A',X',Z'))\right]$$
$$= d_k^2\left(h^*,h\right)$$

Thus,

$$
\begin{aligned}
d_k^2\left(h^*, \hat{h}_{k,U,\lambda,n}\right) &= R_k\left(\hat{h}_{k,U,\lambda,n}\right) \\
&\leq R_k\left(\tilde{h}_k\right) + \lambda M_\lambda + 8M\mathbb{E}_{A,X,Z}\left(\mathcal{R}_{n-1}\left(\mathcal{F}'_{A,X,Z}\right) + \mathcal{R}_n\left(\mathcal{F}'_{A,X,Z}\right)\right) \\
&\quad + 16M^2 M_k\left(\frac{2}{n}\log\frac{2}{\delta}\right)^{\frac{1}{2}} + 10\left(2\log 2\right)^{\frac{1}{2}} M^2 M_k n^{-\frac{1}{2}} \\
&= d_k^2\left(h^*, \tilde{h}_k\right) + \lambda M_\lambda + 8M\mathbb{E}_{A,X,Z}\left(\mathcal{R}_{n-1}\left(\mathcal{F}'_{A,X,Z}\right) + \mathcal{R}_n\left(\mathcal{F}'_{A,X,Z}\right)\right) \\
&\quad + 16M^2 M_k\left(\frac{2}{n}\log\frac{2}{\delta}\right)^{\frac{1}{2}} + 10\left(2\log 2\right)^{\frac{1}{2}} M^2 M_k n^{-\frac{1}{2}} \\
&\leq d_k^2\left(h^*, \tilde{h}_k\right) + \lambda M_\lambda + 8M\left(\mathcal{R}_{n-1}\left(\mathcal{F}'\right) + \mathcal{R}_n\left(\mathcal{F}'\right)\right) + 16M^2 M_k\left(\frac{2}{n}\log\frac{2}{\delta}\right)^{\frac{1}{2}} \\
&\quad + 10\left(2\log 2\right)^{\frac{1}{2}} M^2 M_k n^{-\frac{1}{2}}
\end{aligned}
$$

If the right hand side of this expression goes to zero as $n$ goes to infinity, then, for any $\delta, \epsilon > 0$, we can find $n$ such that the right hand side is less than $\epsilon$. Further, since we can do this for any value of $\delta$, we can choose a sequence of $\delta_n$s decreasing in $n$, so that $\lim_{n\to\infty} \delta_n = 0$, so that the left hand side converges in probability. If $k$ is ISPD, by Lemma 6, $d_k$ is a metric on $L^2_{\mathcal{A}\mathcal{X}\mathcal{Z}}$. Thus, $d_k\left(h^*, \hat{h}_{k,U,\lambda,n}\right) \xrightarrow{\text{P}} 0$, so $\mathbb{E}\left[\hat{h}_{k,U,\lambda,n}\Big| A, X, Z\right] \xrightarrow{\text{P}} \mathbb{E}\left[h^*|A, X, Z\right]$, in $d_k$. Further, the fact that $k$ is Integrally Strictly Positive Definite, implies that $\left\|\mathbb{E}\left[h^*|A,X,Z\right] - \mathbb{E}\left[\hat{h}_{k,U,\lambda,n}\Big| A,X,Z\right]\right\|_{\text{P}_{A,X,Z}} \xrightarrow{\text{P}} 0$, so that $\mathbb{E}\left[\hat{h}_{k,U,\lambda,n}\Big| A, X, Z\right] \xrightarrow{\text{P}} \mathbb{E}\left[h^*|A, X, Z\right]$, in $L^2\left(\text{P}_{\mathcal{A}\mathcal{X}\mathcal{Z}}\right)$-norm, as well.

$\square$

**Lemma 7.** Let $f : \mathcal{X}^2 \to [-M, M]$, $\forall_{x\in\mathcal{X}} f(x, x) \geq 0$, $\hat{U}_n[f] = \frac{1}{n(n-1)}\sum_{i,j=1, j\neq i}^n f(x_i, x_j)$, and $\hat{V}_n[f] = n^{-2}\sum_{i,j=1}^n f(x_i, x_j)$. Then, $(n-1)\hat{U}_n[f] \leq n\hat{V}_n[f] \leq (n-1)\hat{U}_n[f] + M$.

*Proof.*

$$
\begin{aligned}
(n-1)\hat{U}_n[f] &= n^{-1}\sum_{i,j=1, j\neq i}^n f(x_i, x_j) \leq n^{-1}\sum_{i,j=1}^n f(x_i, x_j) \\
&= n\hat{V}_n[f] = n^{-1}\left(\sum_{i,j=1, j\neq i}^n f(x_i, x_j) + \sum_{i,j=1, j=i}^n f(x_i, x_j)\right) \\
&\leq n^{-1} n(n-1)\hat{U}_n[f] + n^{-1}\sum_{i=1}^n f(x_i, x_i) \leq (n-1)\hat{U}_n[f] + M
\end{aligned}
$$

so

$$
(n-1)\hat{U}_n[f] \leq n\hat{V}_n[f] \leq (n-1)\hat{U}_n[f] + M
$$

$\square$

**Corollary 8.** Let $\tilde{h}_k$ minimize $R_k(h)$ and $\hat{h}_{k,V,\lambda,n}$ minimize $\hat{R}_{k,V,\lambda,n}(h)$ for $h \in \mathcal{H}$ and let $h^* : \mathcal{A} \times \mathcal{W} \times \mathcal{X} \to \mathbb{R}$ satisfy $\mathbb{E}\left[Y - h^*(A, W, X)|A, X, Z\right] = 0$ $\text{P}_{A,X,Z}$-almost surely, where

$$
R_k(h) = \mathbb{E}\left[(Y - h(A, W, X))(Y' - h(A', W', X'))k((A, X, Z), (A', X', Z'))\right]
$$

$$
\begin{aligned}
\hat{R}_{k,V,\lambda,n}(h) &= n^{-2}\sum_{i,j=1}^n (y_i - h(a_i, w_i, x_i))(y_j - h(a_j, w_j, x_j))k((a_i, x_i, z_i), (a_j, x_j, z_j)) \\
&\quad + \lambda\Lambda[f, \theta_f]
\end{aligned}
$$

*Also let,*

$$d_k^2 (h, h') = \mathbb{E}\left[(h(A, W, X) - h'(A, W, X))(h(A', W', X') - h'(A', W', X'))\right.$$
$$\left. \times k((A, X, Z), (A', X', Z'))\right]$$

*Then, $d_k^2(h^*, h) = R_k(h)$ and, with probability at least $1 - \delta$,*

$$d_k^2\left(h^*, \hat{h}_{k,V,\lambda,n}\right) \leq d_k^2\left(h^*, \tilde{h}_k\right) + \lambda M_\lambda + 8M\mathbb{E}_{A,X,Z}\left(\mathcal{R}_{n-1}\left(\mathcal{F}'_{A,X,Z}\right) + \mathcal{R}_n\left(\mathcal{F}'_{A,X,Z}\right)\right)$$

$$+ 16M^2 M_k \left(\frac{2}{n}\log\frac{2}{\delta}\right)^{\frac{1}{2}} + \left(4M^2 M_k + \lambda M_\lambda\right)(n-1)^{-1}$$

$$+ 10\left(2\log 2\right)^{\frac{1}{2}} M^2 M_k n^{-\frac{1}{2}}$$

$$\leq d_k^2\left(h^*, \tilde{h}_k\right) + \lambda M_\lambda + 8M\left(\mathcal{R}_{n-1}\left(\mathcal{F}'\right) + \mathcal{R}_n\left(\mathcal{F}'\right)\right) + 16M^2 M_k \left(\frac{2}{n}\log\frac{2}{\delta}\right)^{\frac{1}{2}}$$

$$+ \left(4M^2 M_k + \lambda M_\lambda\right)(n-1)^{-1} + 10\left(2\log 2\right)^{\frac{1}{2}} M^2 M_k n^{-\frac{1}{2}}$$

*Further, if Assumption 5 holds, so $k$ is ISPD, then $d_k$ is a metric on $L^2_{\mathcal{A}\mathcal{X}\mathcal{Z}}$ and, if the right hand side of the inequality goes to zero as $n$ goes to infinity,*

$$d_k\left(\mathbb{E}\left[h^*|A, X, Z\right] - \mathbb{E}\left[\hat{h}_{k,\lambda,n}\Big|A, X, Z\right]\right) \xrightarrow{\mathrm{P}} 0 \;\; so \;\; \mathbb{E}\left[\hat{h}_{k,\lambda,n}\Big|A, X, Z\right] \xrightarrow{\mathrm{P}} \mathbb{E}\left[h^*|A, X, Z\right]$$

*in $d_k$.  Also,* $\left\|\mathbb{E}\left[h^*|A, X, Z\right] - \mathbb{E}\left[\hat{h}_{k,\lambda,n}\Big|A, X, Z\right]\right\|_{\mathrm{P}_{A,X,Z}} \xrightarrow{\mathrm{P}} 0$ *so* $\mathbb{E}\left[\hat{h}_{k,\lambda,n}\Big|A, X, Z\right] \xrightarrow{\mathrm{P}}$
$\mathbb{E}\left[h^*|A, X, Z\right]$ *in $L^2\left(\mathrm{P}_{A,\mathcal{X},\mathcal{Z}}\right)$-norm.*

$$\mathcal{F}'_{a,x,z} = \left\{f_{a,x,z} \mid \exists_{h\in\mathcal{H}} \forall_{a'\in\mathcal{A}, x'\in\mathcal{X}, z'\in\mathcal{Z}} f_{a,x,z}\left(a', w', x', z'\right) = h\left(a', w', x'\right) k\left(\left(a', x', z'\right), (a, x, z)\right)\right\}$$
$$\mathcal{F}' = \left\{f \mid \exists_{h\in\mathcal{H}, a\in\mathcal{A}, x\in\mathcal{X}, z\in\mathcal{Z}} \forall_{a'\in\mathcal{A}, x'\in\mathcal{X}, z'\in\mathcal{Z}} f\left(a', w', x', z'\right) = h\left(a', w', x'\right) k\left(\left(a', x', z'\right), (a, x, z)\right)\right\}$$

*Proof.* Defining $f$ and $\Xi$ as in Theorem 1, then $\hat{R}_{k,U,n}(h) = \frac{1}{n(n-1)}\sum_{i,j=1, j\neq i}^{n} f_h\left(\xi_i, \xi_j\right)$,
$\hat{R}_{k,U,\lambda,n}(h) = \frac{1}{n(n-1)}\sum_{i,j=1, j\neq i}^{n} f_h\left(\xi_i, \xi_j\right) + \lambda\Lambda[f, \theta_f]$, $\hat{R}_{k,V,n}(h) = n^{-2}\sum_{i,j=1}^{n} f_h\left(\xi_i, \xi_j\right)$,
$\hat{R}_{k,V,\lambda,n}(h) = n^{-2}\sum_{i,j=1}^{n} f_h\left(\xi_i, \xi_j\right) + \lambda\Lambda[f, \theta_f]$, Noting that $|f| \leq 4M^2 M_k$ and applying Lemma
7 to $\hat{R}_{k,U,\lambda,n}$ and $\hat{R}_{k,V,\lambda,n}$ yields,

$$\hat{R}_{k,\lambda,n}(h) = \hat{R}_{k,U,n}(h) + \lambda\Lambda[f, \theta_f] \leq \frac{n}{n-1}\hat{R}_{k,V,n}(h) + \lambda\Lambda[f, \theta_f]$$

$$\leq \frac{n}{n-1}\hat{R}_{k,V,n}(h) + \frac{n}{n-1}\lambda\Lambda[f, \theta_f] = \frac{n}{n-1}\hat{R}_{k,V,\lambda,n}(h)$$

$$\leq \hat{R}_{k,U,n}(h) + (n-1)^{-1} 4M^2 M_k + \frac{n}{n-1}\lambda\Lambda[f, \theta_f]$$

From the proof of Theorem 1, with probability at least $1 - \frac{\delta}{2}$, we have,

$$R_k(h) \leq \hat{R}_{k,\lambda,n}(h) + 2\mathbb{E}_\Xi\left(\mathcal{R}_{n-1}\left(\mathcal{F}_{1,\Xi}\right) + \mathcal{R}_n\left(\mathcal{F}_{2,\Xi}\right)\right) + 8M^2 M_k\left(\frac{2}{n}\log\frac{2}{\delta}\right)^{\frac{1}{2}}$$

$$\hat{R}_{k,\lambda,n}(h) \leq R_k(h) + \lambda M^2 + 2\mathbb{E}_\Xi\left(\mathcal{R}_{n-1}\left(\mathcal{F}_{1,\Xi}\right) + \mathcal{R}_n\left(\mathcal{F}_{2,\Xi}\right)\right) + 8M^2 M_k\left(\frac{2}{n}\log\frac{2}{\delta}\right)^{\frac{1}{2}}$$

Recalling that $\hat{h}_{k,V,\lambda,n}$ minimizes $\hat{R}_{k,V,\lambda,n}(h)$ and $\tilde{h}_k$ minimizes $R_k(h)$ over $\mathcal{H}$, so that
$\hat{R}_{k,V,\lambda,n}\left(\hat{h}_{k,V,\lambda,n}\right) \leq \hat{R}_{k,V,\lambda,n}\left(\tilde{h}_k\right)$ and combining the above expressions gives,

$$R_k\left(\hat{h}_{k,V,\lambda,n}\right) \le \hat{R}_{k,U,\lambda,n}\left(\hat{h}_{k,V,\lambda,n}\right) + 2\mathbb{E}_\Xi\left(\mathcal{R}_{n-1}\left(\mathcal{F}_{1,\Xi}\right) + \mathcal{R}_n\left(\mathcal{F}_{2,\Xi}\right)\right) + 8M^2 M_k \left(\frac{2}{n}\log\frac{2}{\delta}\right)^{\frac{1}{2}}$$

$$\le \frac{n}{n-1}\hat{R}_{k,V,\lambda,n}\left(\hat{h}_{k,V,\lambda,n}\right) + 2\mathbb{E}_\Xi\left(\mathcal{R}_{n-1}\left(\mathcal{F}_{1,\Xi}\right) + \mathcal{R}_n\left(\mathcal{F}_{2,\Xi}\right)\right) + 8M^2 M_k \left(\frac{2}{n}\log\frac{2}{\delta}\right)^{\frac{1}{2}}$$

$$\le \frac{n}{n-1}\hat{R}_{k,V,\lambda,n}\left(\tilde{h}_k\right) + 2\mathbb{E}_\Xi\left(\mathcal{R}_{n-1}\left(\mathcal{F}_{1,\Xi}\right) + \mathcal{R}_n\left(\mathcal{F}_{2,\Xi}\right)\right) + 8M^2 M_k \left(\frac{2}{n}\log\frac{2}{\delta}\right)^{\frac{1}{2}}$$

$$\le \hat{R}_{k,n}\left(\tilde{h}_k\right) + (n-1)^{-1}4M^2 M_k + \frac{n}{n-1}\lambda\Lambda[f,\theta_f] + 2\mathbb{E}_\Xi\left(\mathcal{R}_{n-1}\left(\mathcal{F}_{1,\Xi}\right) + \mathcal{R}_n\left(\mathcal{F}_{2,\Xi}\right)\right)$$
$$+ 8M^2 M_k \left(\frac{2}{n}\log\frac{2}{\delta}\right)^{\frac{1}{2}}$$

$$\le R_k\left(\tilde{h}_k\right) + (n-1)^{-1}4M^2 M_k + \frac{n}{n-1}\lambda M_\lambda + 4\mathbb{E}_\Xi\left(\mathcal{R}_{n-1}\left(\mathcal{F}_{1,\Xi}\right) + \mathcal{R}_n\left(\mathcal{F}_{2,\Xi}\right)\right)$$
$$+ 16M^2 M_k \left(\frac{2}{n}\log\frac{2}{\delta}\right)^{\frac{1}{2}}$$

$$\le R_k\left(\tilde{h}_k\right) + \left(4M^2 M_k + \lambda M_\lambda\right)(n-1)^{-1} + \lambda M_\lambda + 4\mathbb{E}_\Xi\left(\mathcal{R}_{n-1}\left(\mathcal{F}_{1,\Xi}\right) + \mathcal{R}_n\left(\mathcal{F}_{2,\Xi}\right)\right)$$
$$+ 16M^2 M_k \left(\frac{2}{n}\log\frac{2}{\delta}\right)^{\frac{1}{2}}$$

Using Lemma 5 and results from the proof of Theorem 1,

$$R_k\left(\hat{h}_{k,V,\lambda,n}\right) \le R_k\left(\tilde{h}_k\right) + \left(4M^2 M_k + \lambda M_\lambda\right)(n-1)^{-1} + \lambda M_\lambda$$
$$+ 8M\left(\mathcal{R}_{n-1}\left(\mathcal{F}'\right) + \mathcal{R}_n\left(\mathcal{F}'\right)\right) + 16M^2 M_k \left(\frac{2}{n}\log\frac{2}{\delta}\right)^{\frac{1}{2}}$$
$$+ 10\left(2\log 2\right)^{\frac{1}{2}} M^2 M_k n^{-\frac{1}{2}}$$

$$= R_k\left(\tilde{h}_k\right) + \lambda M_\lambda + 8M\left(\mathcal{R}_{n-1}\left(\mathcal{F}'\right) + \mathcal{R}_n\left(\mathcal{F}'\right)\right) + 16M^2 M_k \left(\frac{2}{n}\log\frac{2}{\delta}\right)^{\frac{1}{2}}$$
$$+ \left(4M^2 M_k + \lambda M_\lambda\right)(n-1)^{-1} + 10\left(2\log 2\right)^{\frac{1}{2}} M^2 M_k n^{-\frac{1}{2}}$$

Recalling that $d_k^2\left(h^*, h\right) = R_k(h)$, we have,

$$d_k^2\left(h^*, \hat{h}_{k,V,\lambda,n}\right) \le d_k^2\left(h^*, \tilde{h}_k\right) + \lambda M^2 + 8M\left(\mathcal{R}_{n-1}\left(\mathcal{F}'\right) + \mathcal{R}_n\left(\mathcal{F}'\right)\right) + 16M^2 M_k \left(\frac{2}{n}\log\frac{2}{\delta}\right)^{\frac{1}{2}}$$
$$+ \left(4M^2 M_k + \lambda M_\lambda\right)(n-1)^{-1} + 10\left(2\log 2\right)^{\frac{1}{2}} M^2 M_k n^{-\frac{1}{2}}$$

$\square$

**Theorem 2.** *Under Assumption 4, $h^*$ is the unique solution $\mathrm{P}_{A,W,X}$-almost surely. Further, if*
$\mathbb{E}\left[\hat{h}_{k,\lambda,n}\middle| A, X, Z\right] \xrightarrow{\mathrm{P}} \mathbb{E}\left[h^*|A, X, Z\right], \hat{h}_{k,\lambda,n} \xrightarrow{\mathrm{P}} h^*.$

*Proof.* Let $h^*, h^{*\prime}$ both be zeros of $\mathbb{E}\left[Y - h\left(A, W, X\right)|A, X, Z\right]$, $\mathrm{P}_{A,X,Z}$-almost surely, then

$$\mathbb{E}\left[(h^* - h^{*\prime})(A, W, X)|A, X, Z\right] = \mathbb{E}\left[(h^* - Y + Y - h^{*\prime})(A, W, X)|A, X, Z\right]$$
$$= \mathbb{E}\left[Y - h^{*\prime}(A, W, X)|A, X, Z\right] - \mathbb{E}\left[Y - h^*(A, W, X)|A, X, Z\right] = 0 - 0$$
$$= 0$$

$P_{A,X,Z}$-almost surely, so $h^* = h^{*\prime}$ $P_{A,W,X}$-almost surely.

If $\mathbb{E}\left[\hat{h}_{k,\lambda,n}\Big|A, X, Z\right] \xrightarrow{P} \mathbb{E}\left[h^*|A, X, Z\right]$, $\left\|\mathbb{E}\left[\left(\hat{h}_n - h^*\right)(A, W, X)\Big|A, X, Z\right]\right\|_{P_{A,X,Z}} \xrightarrow{P} 0$,

meaning that the convergence occurs $P_{A,X,Z}$-almost surely, so, by assumption, $\left\|\hat{h}_n - h^*\right\|_{P_{A,W,X}} \xrightarrow{P}$

$0$, so $\hat{h}_n \xrightarrow{P} h^*$.

$\square$

# B  Hyperparameter Tuning & Model Architecture

We tuned the architectures of the Naive Net and NMMR models on both the Demand and dSprite experiments. The Naive Net used MSE loss to estimate $Y^a$, while NMMR relied on either the U-statistic or V-statistic.

Within each experiment, the Naive Net and NMMR models used similar architectures. In the Demand experiment, both models consisted of 2-5 ("Network depth" in Table S1) fully connected layers with a variable number ("Network width") of hidden units.

In the dSprite experiment, each model had two VGG-like heads [28] that took in $A$ and $W$ images and applied two blocks of {Conv2D, Conv2D, MaxPool2D} with 3 by 3 kernels. Each Conv2D layer had 64 filters in the first block, then 128 filters in the second block. The output of the second block was flattened, then projected to 256 dimensions. Two subsequent fully connected layers were used, with their number of units determined by the "layer width decay" factor in Table S1. For example, if this factor was 0.5, then the two layers would have 128 and 64 units, respectively.

We performed a grid search over the following parameters:

Table S1: Grid of hyperparameters for our naive neural network and NMMR models.

| Hyperparameter | Demand | dSprite |
|---|---|---|
| Learning rate | {3e-3, 3e-4, 3e-5} | {3e-4, 3e-5, 3e-6} |
| L2 penalty | {3e-5, 3e-6, 3e-7} | {3e-6, 3e-7} |
| # of epochs | 3000 | 500 |
| Batch size | 1000 | 256 |
| Layer width decay | | {0.25, 0.5} |
| Network width | {10, 40, 80} | |
| Network depth | {2, 3, 4, 5} | |

We selected the final hyperparameters by considering the lowest average U-statistic or V-statistic on held-out validation sets for NMMR or the MSE for the Naive Net. For the Demand experiment, we repeated this process 10 times with different random seeds and averaged the statistics. For the dSprite experiment, we had 3 repetitions.

Full hyperparameter choices for all methods used in this work are available in our code. The hyperparameters selected for NMMR were tuned for each dataset:

Table S2: Optimal hyperparameters for NMMR methods

| Hyperparameter | NMMR-U Demand | NMMR-U dSprite | NMMR-V Demand | NMMR-V dSprite |
|---|---|---|---|---|
| Learning rate | 3e-3 | 3e-5 | 3e-3 | 3e-5 |
| L2 penalty | 3e-6 | 3e-6 | 3e-6 | 3e-7 |
| # of epochs | 3,000 | 500 | 3,000 | 500 |
| Batch size | 1,000 | 256 | 1,000 | 256 |
| Layer width decay | — | 0.25 | — | 0.5 |
| Network width | 80 | — | 80 | — |
| Network depth | 4 | — | 3 | — |

The hyperparameters for Naive Net for each dataset were:

Table S3: Optimal hyperparameters for Naive Net model

| Hyperparameter | Naive Net Demand | Naive Net dSprite |
|---|---|---|
| Learning rate | 3e-3 | 3e-5 |
| L2 penalty | 3e-6 | 3e-6 |
| # of epochs | 3,000 | 500 |
| Batch size | 1,000 | 256 |
| Layer width decay | — | 0.25 |
| Network width | 80 | — |
| Network depth | 2 | — |

Another hyperparameter of note is the choice of kernel in the loss function of NMMR. Throughout, we relied on the RBF kernel:

$$k(x_i, x_j) = e^{\frac{-||x_i - x_j||_2^2}{2\sigma^2}}$$

with $\sigma = 1$. For future work, we could consider other choices of the kernel or tune the length scale parameter $\sigma$. In the dSprite experiment, the kernel function is applied to pairs of $Z$ and $A$ data. Since $A$ is an $64 \times 64$ image, we chose to concatenate $(z_i, 0.05a_i)$ as input to the kernel for the $i$-th data point. One could also consider tuning this multiplicative factor, but in practice we found that it allowed for both $Z$ and $A$ to impact the result of the kernel function.

## C  Experiment Details

### C.1  Demand Data Generating Process

The Demand experiment has the following structural equations:

- Demand: $U \sim \mathcal{U}(0, 10)$
- Fuel cost: $[Z_1, Z_2] = [2\sin(2\pi U/10) + \epsilon_1, 2\cos(2\pi U/10) + \epsilon_2]$
- Web page views: $W = 7g(U) + 45 + \epsilon_3$
- Price: $A = 35 + (Z_1 + 3)g(U) + Z_2 + \epsilon_4$
- Sales: $Y = A \times \min(\exp(\frac{W-A}{10}), 5) - 5g(U) + \epsilon_5$
- where $g(u) = 2\left(\frac{(u-5)^4}{600} + e^{-4(u-5)^2} + \frac{u}{10} - 2\right)$, and $\epsilon_i \sim \mathcal{N}(0, 1)$

### C.2  Demand causal DAG

### C.3  Demand Exploratory Data Analysis

In Figure S2, Panels A and B show that $W$ is a more informative proxy for $U$ than $Z$, although neither relationship is one-to-one. Panel C shows that the true potential outcome curve, denoted by the black curve $a \mapsto E[Y^a]$, deviates from the observed $(A, Y)$ distribution due to confounding. In

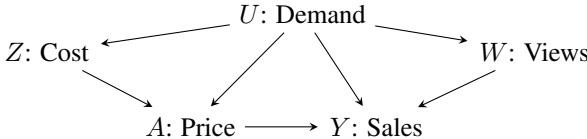

Figure S1: Causal DAG for the Demand experiment.

particular, the largest deviation occurs at smaller values of $A$. The goal of each method is to recover this average potential outcome curve given data on $A, Z, W,$ and $Y$.

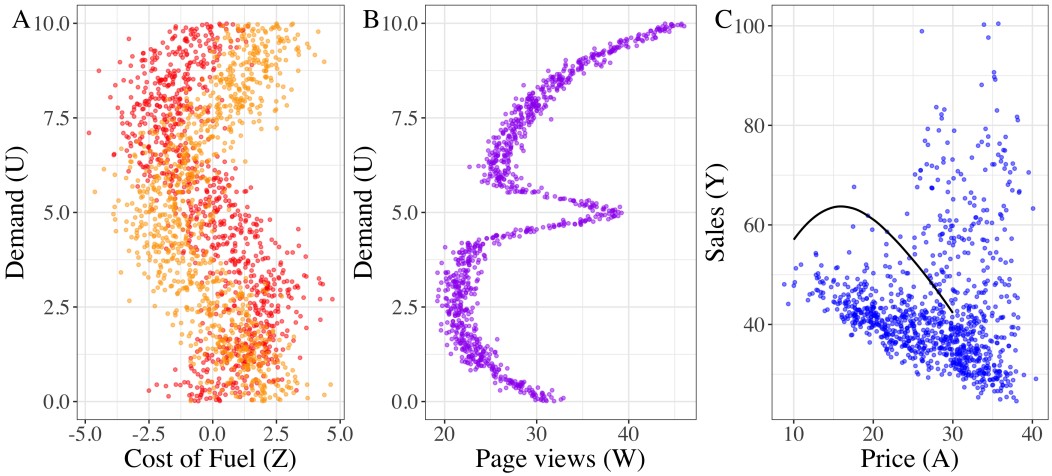

Figure S2: Views of the $A, Z, W, Y, U$ relationships. (A) $Z_1$ (red) and $Z_2$ (orange) have sinusoidal relationships with $U$, (B) $W$ has a far less noisy relationship with $U$, and (C) the observed distribution $(A, Y)$ (blue) deviates from the true average potential outcome curve (black).

## C.4 Demand Boxplot Statistics

Table S4 contains the median and interquartile ranges (in parentheses) of the c-MSE values compiled in the boxplots shown in Figure 2. NMMR demonstrated state of the art performance on the Demand benchmark. We also extended the benchmark to include training set sizes of 10,000 and 50,000 data points, whereas Xu et al. [7] originally included 1,000 and 5,000 data points. We observed that NMMR-U had strong performance across all dataset sizes. PMMR and KPV were unable to run on 50,000 training points due to computational limits on their kernel methods, while DFPV exhibited a large increase in c-MSE as training set size increased.

Table S4: Demand Boxplot Median & (IQR) values

| | Training Set Size | | | |
| --- | --- | --- | --- | --- |
| Method | 1,000 | 5,000 | 10,000 | 50,000 |
| PMMR | 587.51 (40.35) | 466.5 (33.47) | 423.1 (29.26) | — |
| KPV | 469.94 (97.07) | 481.32 (54.8) | 470.62 (29.3) | — |
| Naive Net | 160.35 (33.78) | 186.97 (30.22) | 204.36 (113.71)) | 224.09 (33.17) |
| CEVAE | 180.8 (161.26) | 214.98 (120.88) | 170.58 (176.1) | 171.98 (293.27) |
| 2SLS | 82.08 (18.82) | 83.16 (4.51) | 82.1 (5.55) | 82.01 (2.22) |
| DFPV | 41.83 (11.78) | 48.22 (7.73) | 87.14 (471.59) | 242.15 (464.38) |
| LS | 63.19 (5.82) | 65.14 (2.64) | 64.98 (2.44) | 64.65 (0.74) |
| **NMMR-U (ours)** | 23.68 (8.02) | **16.21 (10.55)** | **14.25 (4.46)** | **14.27 (12.47)** |
| **NMMR-V (ours)** | **23.41 (11.26)** | 30.74 (17.73) | 42.88 (29.45) | 62.18 (16.97) |

## C.5  Demand Prediction Curves

Figures S3-S6 provide the individual predicted potential outcome curves of each method in the Demand experiment. While Figure 2 provides a summary of the c-MSE, this does not give a picture of the model's actual estimate of $\mathbb{E}[Y^a]$. These figures give an insight into ranges of $A$ for which each model provides particularly accurate or inaccurate estimates of the potential outcomes. Across all training set sizes, methods are most accurate in the region where the training observations of $Y$ are closest to the ground truth (see Figure S2).

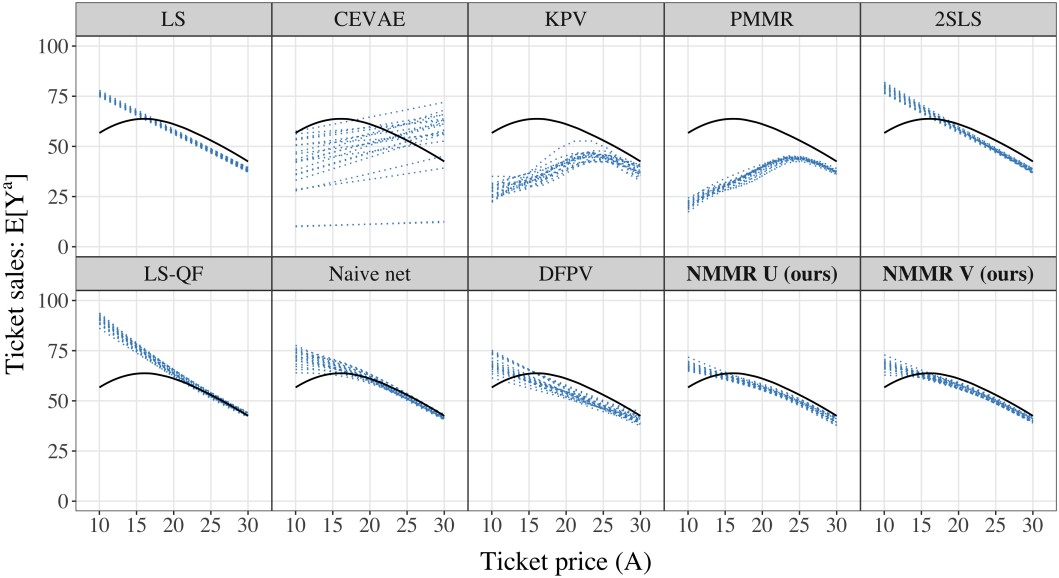

Figure S3: Demand experiment with 1,000 training data points with the true average potential outcome curves (black) and each method's predicted potential outcome curves (blue). Each method was replicated 20 times, generating one predicted curve per replicate. Note that with only a limited amount of data, most methods only recover the true curve in the later half of the range of $A$. See Figure S2.

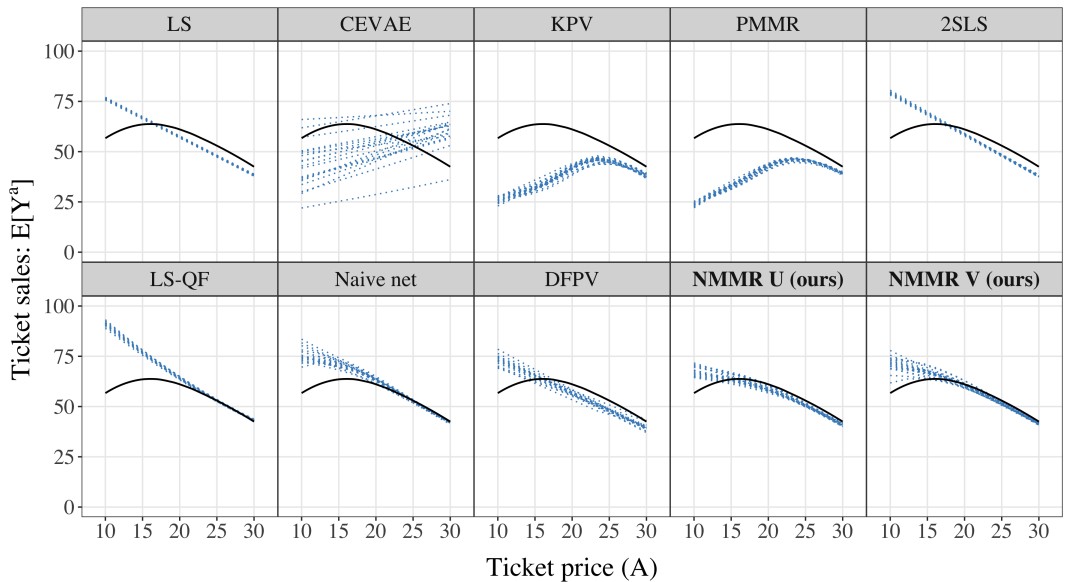

Figure S4: Demand experiment with 5,000 training data points with the true average potential outcome curves (black) and each method's predicted potential outcome curves (blue). Each method was replicated 20 times, generating one predicted curve per replicate. Note that now, NMMR begins to adjust in the range of $A \in [10, 20]$ and bend down towards the true curve. NMMR is empirically accounting for the unobserved confounder $U$ through the proxy variables.

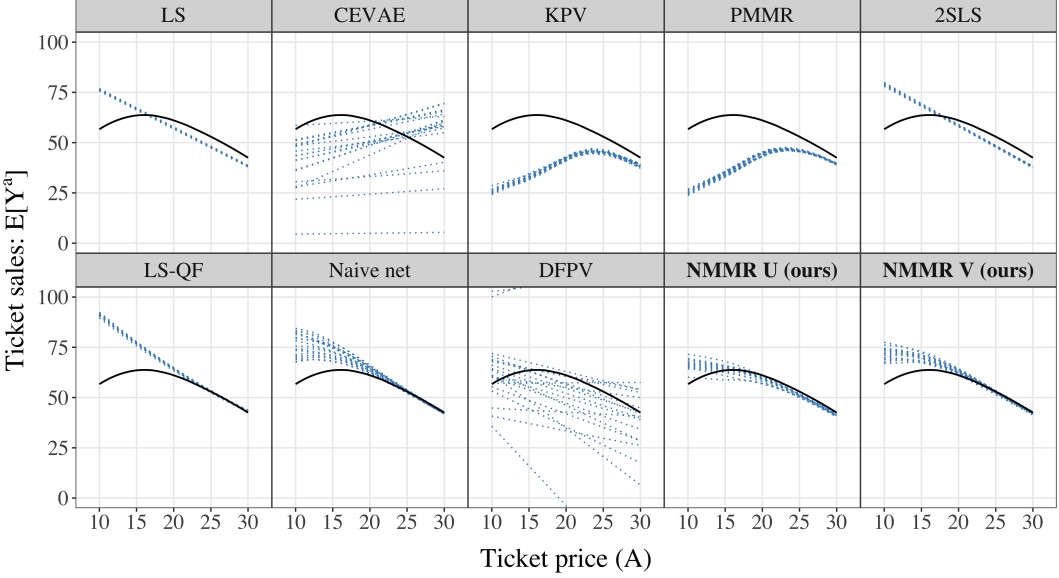

Figure S5: Demand experiment with 10,000 training data points with the true average potential outcome curves (black) and each method's predicted potential outcome curves (blue). Each method was replicated 20 times, generating one predicted curve per replicate. We observed some additional curvature to NMMR prediction curves

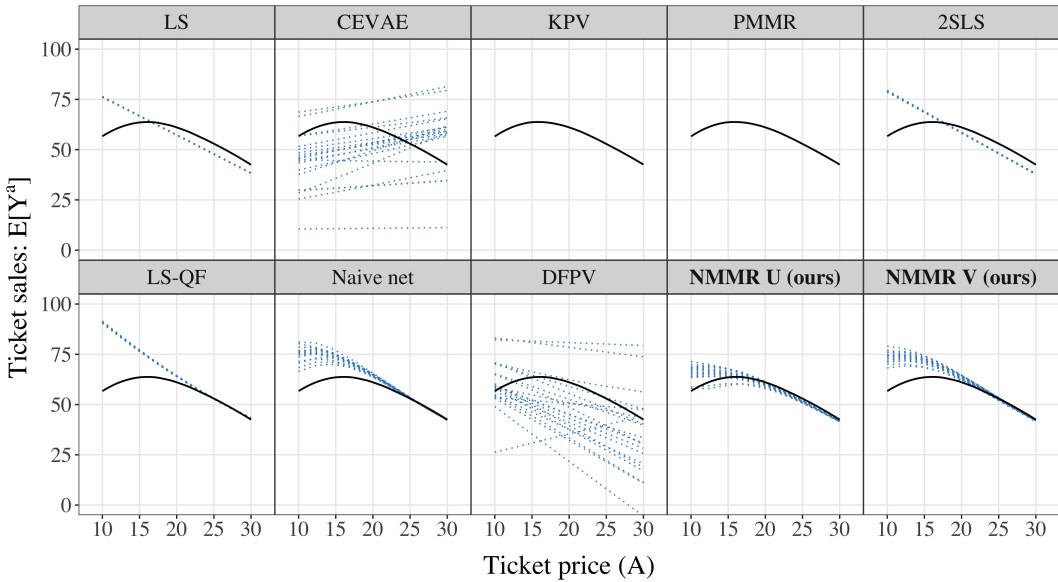

Figure S6: Demand experiment with 50,000 training data points with the true average potential outcome curves (black) and each method's predicted potential outcome curves (blue). Each method was replicated 20 times, generating one predicted curve per replicate. KPV and PMMR timed out due to computational requirements of their kernel methods.

## C.6  dSprite Data Generating Process

The dSprite experiment has a unique data generating mechanism, given that the images $A$ and $W$ are queried from an existing dataset rather than generated on the fly. The dataset is indexed by parameters: shape, color, scale, rotation, posX, and posY. As mentioned in the paper, this experiment fixes shape = heart, color = white. Therefore, to simulate data from this system, we follow the steps:

1. Simulate values for scale, rotation, posX, posY [†].
2. Set $U = \text{posY}$.
3. Set $Z = (\text{scale, rotation, posX})$.
4. Set $A$ equal to the dSprite image with the corresponding (scale, rotation, posX and posY) as found in $Z$ and $U$, then add $\mathcal{N}(0, 0.1)$ noise to each pixel.
5. Set $W$ equal to the dSprite image with (scale=0.8, rotation=0, posX=0.5) and posY from $U$, then add $\mathcal{N}(0, 0.1)$ noise to each pixel.
6. Compute $Y = \frac{\frac{1}{10}||vec(A)^T B||_2^2 - 5000}{1000} \times \frac{(31 \times U - 15.5)^2}{85.25} + \epsilon, \epsilon \sim \mathcal{N}(0, 0.5)$

[†] Let $\mathcal{DU}(a, b)$ denote a Discrete Uniform distribution from $a$ to $b$. Scale is a Discrete Uniform random variable taking values [0.5, 0.6, 0.7, 0.8, 0.9, 1.0] with equal probability. Rotation $\sim \mathcal{DU}(0, 2\pi)$ with 40 equally-spaced values. And both posX, posY $\sim \mathcal{DU}(0, 1)$ with 32 equally-spaced values.

## C.7  dSprite causal DAG

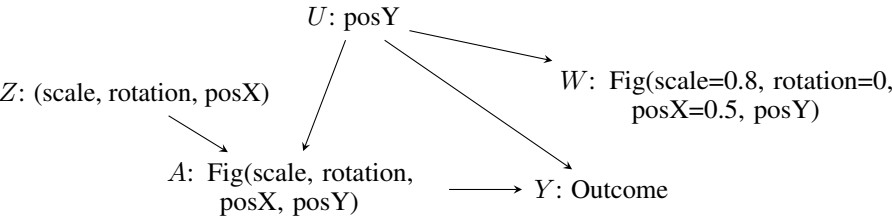

Figure S7: DAG for the dSprite experiment

## C.8 dSprite Exemplar $A$ and $W$

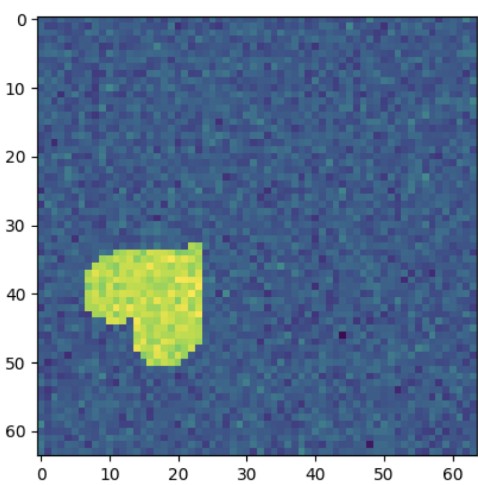
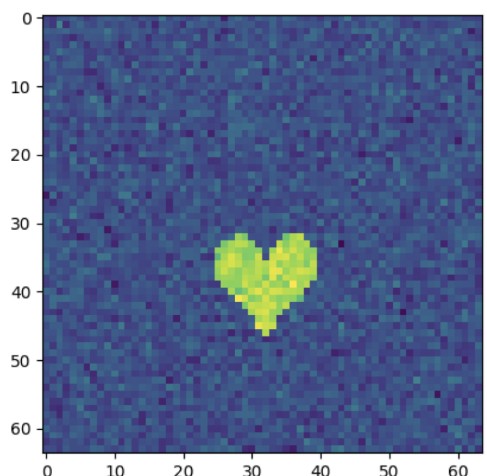

(a) Example of $A$ in dSprite corresponding (scale, rotation, posX and posY) determined from $Z$ and $U$.

(b) Example of $W$ in dSprite, which is always a centered, vertical heart with a fixed scale. The only thing that changes is posY, which is determined by $U$.

Figure S8: Examples of the image based treatment $A$ and outcome-inducing proxy $W$ in the dSprite experiment. Previous proximal inference methods did not take advantage of the inductive bias of image convolutions, which NMMR naturally incorporates into its neural network architecture for the dSprite benchmark.

## C.9 dSprite Test Set

The dSprite test set consists of 588 images $A$ spanning the following grid of parameters:

- posX $\in [0, \frac{5}{31}, \frac{10}{31}, \frac{15}{31}, \frac{20}{31}, \frac{25}{31}, \frac{30}{31}]$
- posY $\in [0, \frac{5}{31}, \frac{10}{31}, \frac{15}{31}, \frac{20}{31}, \frac{25}{31}, \frac{30}{31}]$
- scale $\in [0.5, 0.8, 1.0]$
- rotation $\in [0, 0.5\pi, \pi, 1.5\pi]$

The labels for each test image $A$ are computed as $\mathbb{E}[Y^a] = \frac{\frac{1}{10}||vec(a)^T B||_2^2 - 5000}{1000}$.

## C.10 dSprite Boxplot Statistics

Table S5 contains the median and interquartile ranges (in parentheses) of the c-MSE values compiled in the boxplots shown in Figure 3. NMMR demonstrated state of the art performance on the dSprite benchmark. We also extended the benchmark to include training set sizes of 7,500 data points, whereas Xu et al. [7] originally included 1,000 and 5,000 data points. We observed that NMMR-V had strong performance across all dataset sizes and particularly excelled when training data increased. Most other methods remained relatively consistent as the amount of data increased.

Table S5: dSprite Boxplot Median & (IQR) values

| Method | Training Set Size | | |
| --- | --- | --- | --- |
| | 1,000 | 5,000 | 7,500 |
| PMMR | 17.7 (0.78) | 17.74 (0.64) | 16.2 (0.48) |
| KPV | 23.4 (1.37) | 16.58 (0.93) | 14.46 (1.01) |
| Naive Net | 32.25 (9.72) | 34.24 (11.95) | 34.76 (11.95) |
| CEVAE | 26.34 (0.82) | 26.16 (1.51) | 25.77 (1.45) |
| DFPV | 10.02 (2.95) | 8.81 (2.04) | 8.52 (1.06) |
| **NMMR-U (ours)** | **4.72 (1.1)** | 7.1 (2.74) | 7.52 (2.05) |
| **NMMR-V (ours)** | 11.8 (1.88) | **1.82 (0.67)** | **1.53 (0.68)** |

## C.11 dSprite DFPV vs. NMMR Evaluation

In order to assess whether our improved performance on dSprite was due to the fact that NMMR leveraged convolutional neural networks while DFPV relied on multi-layer perceptrons with a spectral-norm regularization. We modified DFPV to include the same VGG-like heads mentioned in Appendix B. We performed a grid search over the same learning rates and L2 penalties in Table S1. Figure S9 shows that DFPV with CNNs actually performed slightly worse than the original, published-version of DFPV. We report several different results for DFPV CNN since the results were so close after cross validation. Figure S9 shows results for only the 1,000 data point evaluation – DFPV had trouble scaling in practice as dataset size increased.

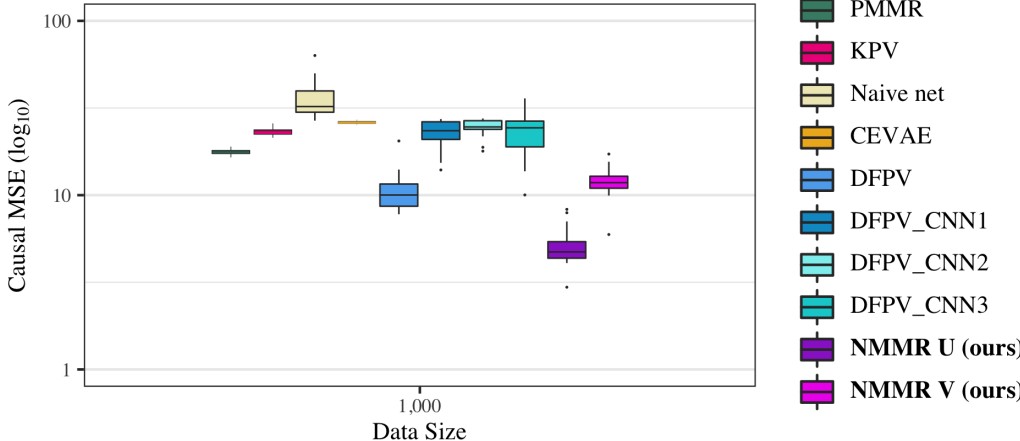

Figure S9: Performance of DFPV with CNNs compared to other evaluated methods on the dSprite benchmark. DFPV with CNNs performed worse compared to the published version of DFPV across a variety of hyperparameters

| Model | Learning Rate | Weight Decay |
| --- | --- | --- |
| DFPV_CNN1 | 3e-6 | 3e-6 |
| DFPV_CNN2 | 3e-6 | 3e-7 |
| DFPV_CNN3 | 3e-5 | 3e-7 |

Table S6: Hyperparameters for reported DFPV_CNN models

## D   Batched Loss Function

When computing the unregluarized version of the loss function of NMMR:

$$\mathcal{L} = (Y - h(A, W, X))^T K (Y - h(A, W, X))$$

we either had to compute the kernel matrix $K$ for all points in the training set once, or dynamically calculate this matrix per batch. The latter approach would require many, many more calculations since we'd be repeating this process every batch and every epoch.

Our solution relied on batching the V-statistic and U-statistic. Recall we can write the V-statistic as:

$$\hat{R}_V(h) = n^{-2} \sum_{i,j=1}^{n} (y_i - h_i)(y_j - h_j) k_{ij}$$

We can vectorize this double sum as a series of vector and matrix multiplciations:

$$(y_1 - h(a, w_1, x_1), \ldots, y_n - h(a, w_n, x_n)) \begin{pmatrix} k_{1,1} & \cdots & k_{1,n} \\ \vdots & \ddots & \vdots \\ k_{n,1} & \cdots & k_{n,n} \end{pmatrix} \begin{pmatrix} y_1 - h(a, w_1, x_1) \\ \cdots \\ y_n - h(a, w_n, x_n) \end{pmatrix}$$

and for the U-statistic, we can simply set the main diagonal of $K$ to be 0 to eliminate $i = j$ terms from this double sum.

However, calculating $K$ for large datasets in a tensor-friendly manner resulted in enormous GPU allocation requests, on the order of 400GBs in the dSprite experiments. We implemented a batched version of the matrix multiplication above to circumvent this issue:

```python
def NMMR_loss_batched(model_output, target, kernel_inputs,
        kernel_function, batch_size: int, loss_name: str):
    residual = target - model_output
    n = residual.shape[0]

    loss = 0
    for i in range(0, n, batch_size):
        # return the i-th to i+batch_size rows of K
        partial_kernel_matrix = calculate_kernel_matrix_batched(
            kernel_inputs,
            (i, i+batch_size),
            kernel_function)
        if loss_name == "V_statistic":
            factor = n ** 2
        if loss_name == "U_statistic":
            factor = n * (n-1)
            # zero out the main diagonal of the full matrix
            for row_idx in range(partial_kernel_matrix.shape[0]):
                partial_kernel_matrix[row_idx, row_idx+i] = 0
        # partial matrix multiplication
        temp_loss = residual[i:(i+batch_size)].T @
            partial_kernel_matrix @ residual / factor
        loss += temp_loss[0, 0]
    return loss
```

## E   Noise Figures

In the Demand noise experiment, we tested each method's ability to estimate $\mathbb{E}[Y^a]$ given varying levels of noise in the proxies $Z$ and $W$. Specifically, we varied the variance on the Gaussian noise terms $\epsilon_1$, $\epsilon_2$, and $\epsilon_3$ from the Demand structural equations described in Appendix C.1. We will refer to these variances as $\sigma_{Z_1}^2$, $\sigma_{Z_2}^2$, and $\sigma_W^2$, respectively, and we set $\sigma_{Z_1}^2 = \sigma_{Z_2}^2$ throughout. We refer to the pair $(\sigma_{Z_1}^2, \sigma_{Z_2}^2)$ as "Z noise" and $\sigma_W^2$ as "W noise". In Xu et al. [7], these variances were all equal to 1. We evaluated each method†on 5000 samples from the Demand data generating process with the following $Z$ and $W$ noise levels:

$$\sigma_{Z_1}^2, \sigma_{Z_2}^2 \in \{0, 0.01, 0.1, 0.5, 1, 2, 4, 8, 16\}$$
$$\sigma_W^2 \in \{0, 0.01, 0.1, 0.5, 1, 16, 64, 150\}$$

In total there are 9 x 8 = 72 noise levels. From Appendix Figure S2, Panels A and B, we can see that $Z_1$ and $Z_2$ lie approximately within the interval $[-4, 4]$, whereas $W$ lies approximately in the interval $[20, 45]$. Accordingly, we chose the maximum value of $Z$ and $W$ noise to be the square of half of the variable's range. So for $W$, half of this range is approximately 12.5 units, therefore the maximum value for $\sigma_W^2$ is $12.5^2 \approx 150$. Similarly, half of the range of $Z$ is 4, and so the maximum value of both $\sigma_{Z_1}^2$ and $\sigma_{Z_2}^2$ is $4^2 = 16$. This maximum level of noise is capable of completely removing any information on $U$ contained in $Z$ and $W$.

Intuitively, as the noise on $Z$ and $W$ is increased, they become less informative proxies for $U$. We would expect that greater noise levels will degrade each method's performance in terms of c-MSE. This experiment provides a way of evaluating how efficient each method is at recovering information about $U$, given increasingly corrupted proxies. It also provides some initial insights into the relative importance of each proxy, $Z$ and $W$.

Figure S10 contains a 72-window grid plot with 1 window for each combination of $Z$ and $W$ noise and Figures S11-S18 show each method's individual potential outcome prediction curves at each of the 72 noise levels. We can see that NMMR-V is notably more robust to noise than NMMR-U, and also appears to be the most efficient method at higher noise levels. NMMR also consistently outperforms Naive Net, supporting the utility of the U- and V-statistic loss functions. However, we also note that kernel-based methods, such as KPV and PMMR, rank increasingly well with increased noise level, likely due to their lack of data adaptivity. We also observe that less data adaptive methods are less prone to large errors. Finally, we see a surprising trend that as the $Z$ noise is increased, several methods achieve lower c-MSE. We believe this stems from the fact that $W$ is a more informative proxy, so it is possible that noising $Z$ aids methods in relying more strongly on the better proxy for $U$.

†We used the optimal hyperparameters for NMMR-U, NMMR-V and Naive Net found through tuning, as described in Appendix B.

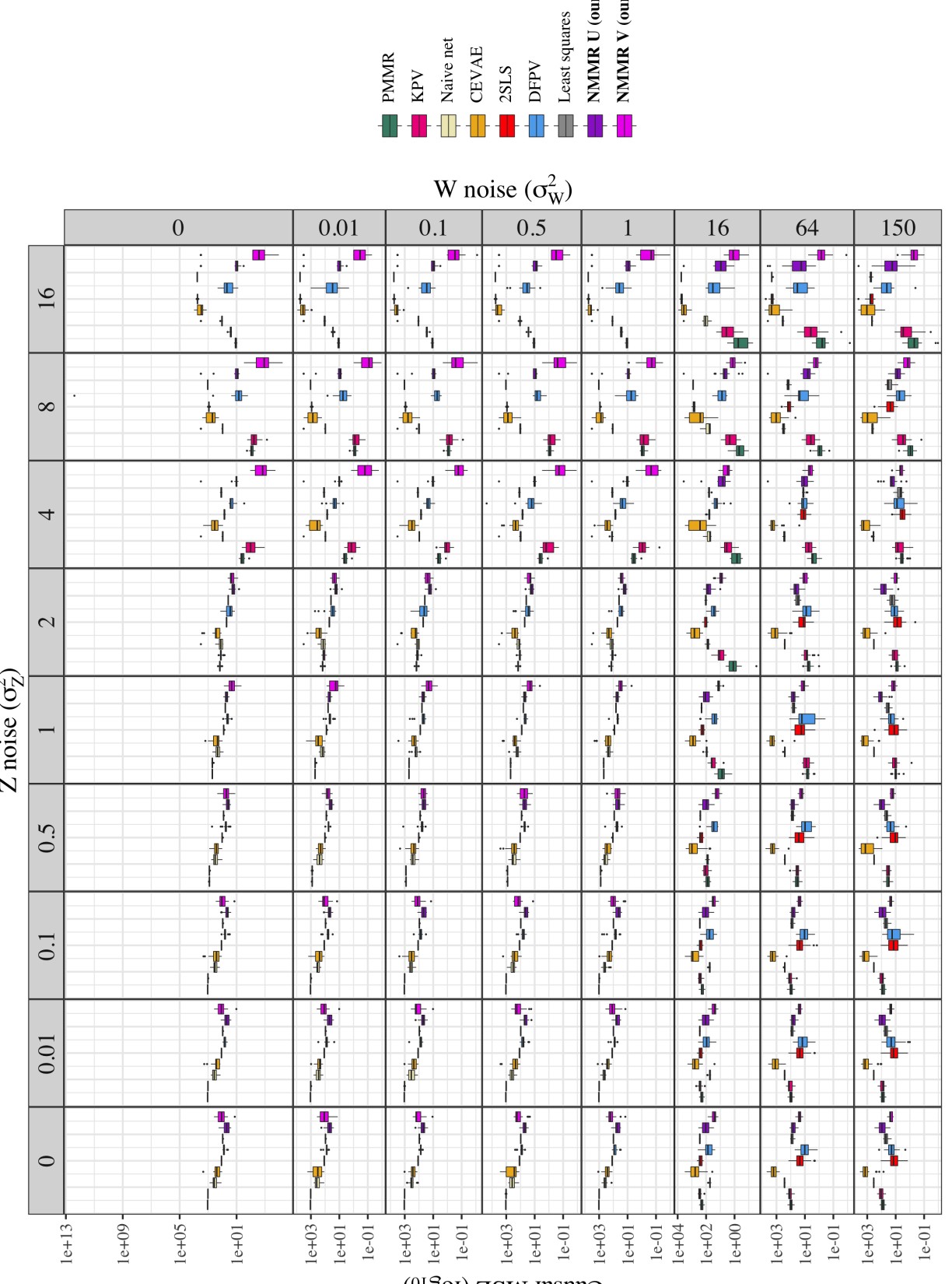

Figure S10: Causal MSE (c-MSE) of NMMR and baseline methods across 72 noise levels in the Demand experiment. Each method was replicated 20 times and evaluated on the same 10 test values of $\mathbb{E}[Y^a]$ each replicate. Each individual box plot represents 20 values of c-MSE.

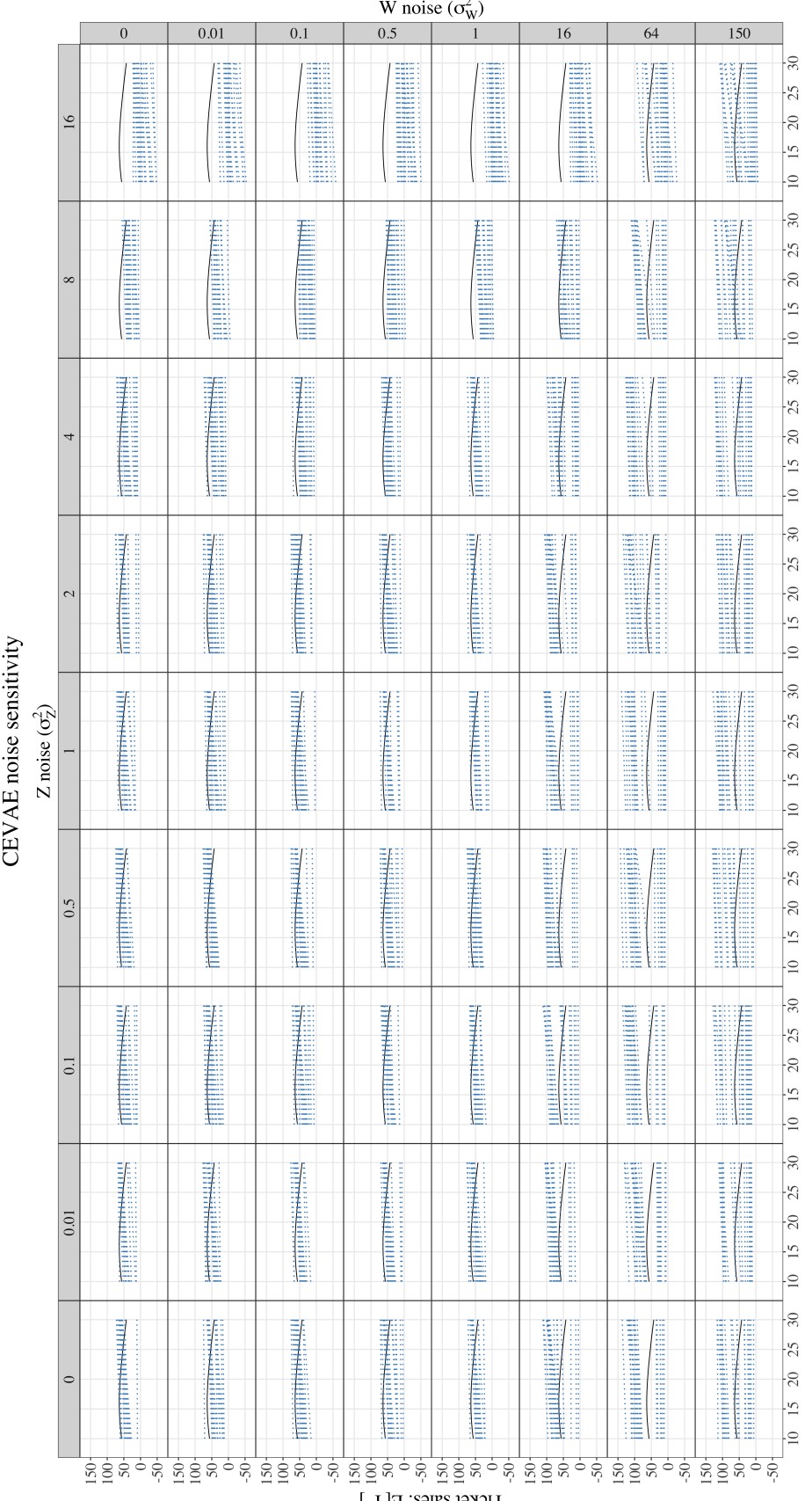

Figure S11: Predicted potential outcome curves for 20 replicates of CEVAE. Black curve is the ground truth $\mathbb{E}[Y^a]$.

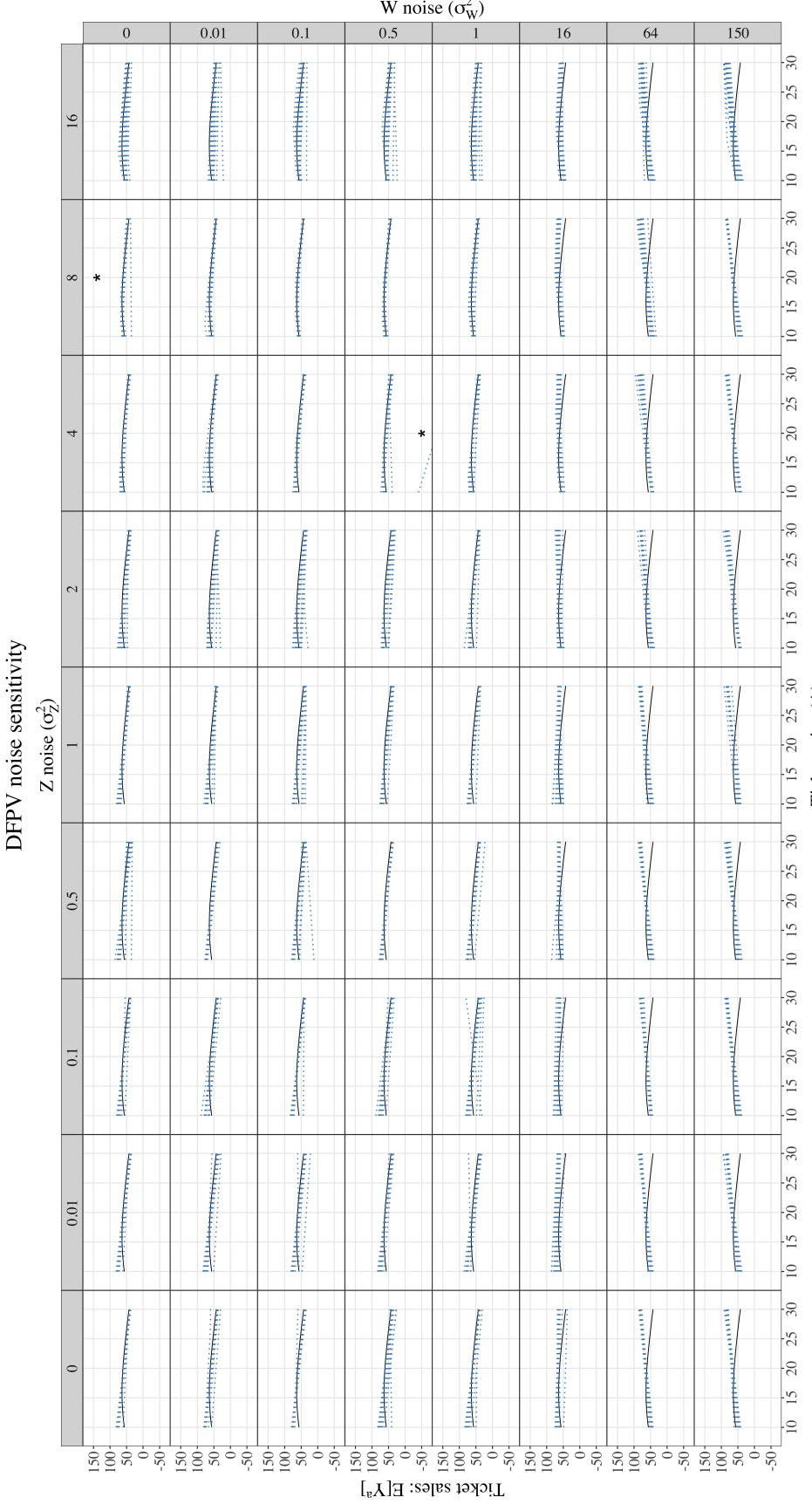

Figure S12: Predicted potential outcome curves for 20 replicates of DFPV. Black curve is the ground truth $\mathbb{E}[Y^a]$. Asterisks each indicate a replicate that lies outside of the plot's range.

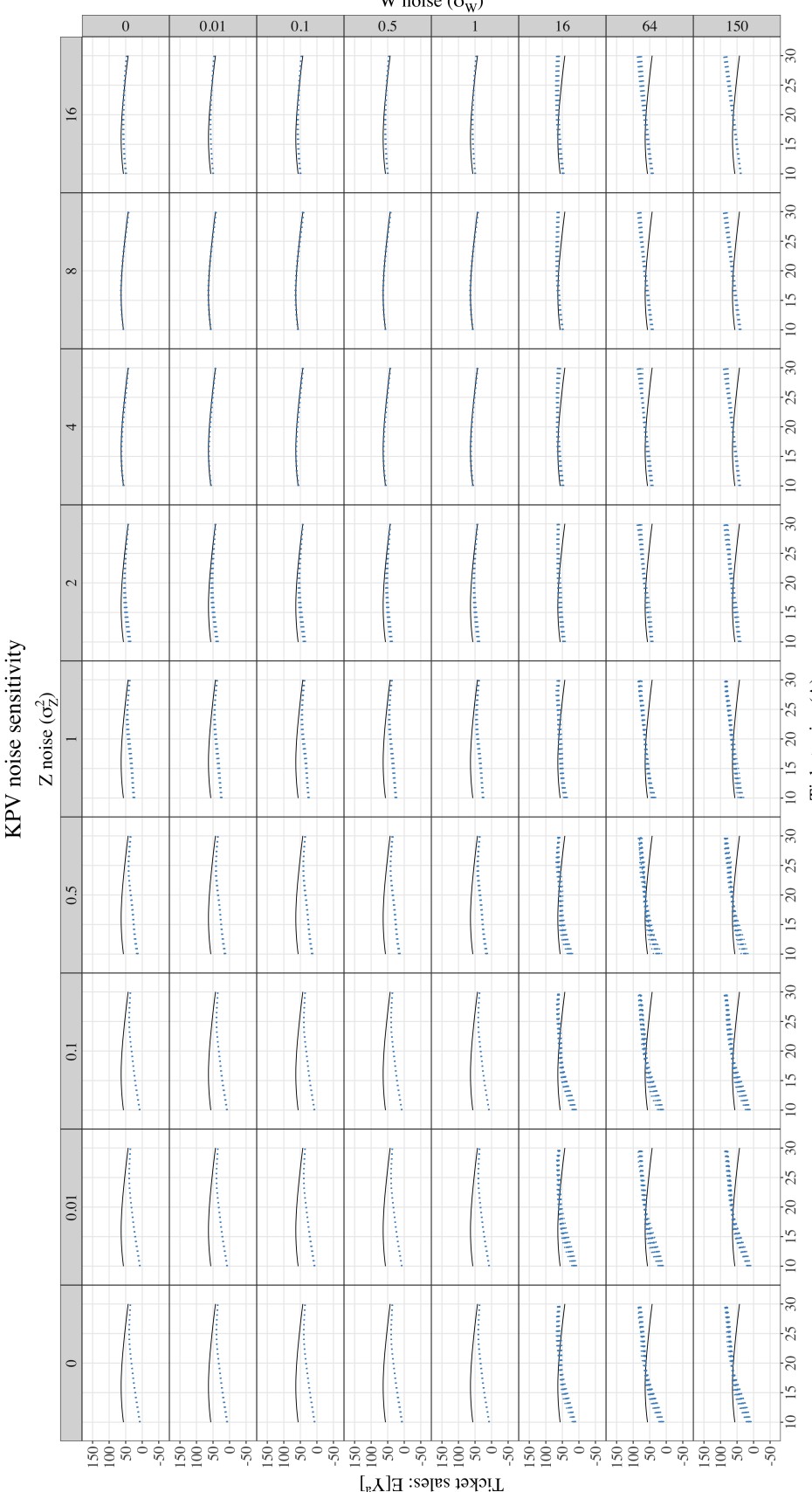

Figure S13: Predicted potential outcome curves for 20 replicates of KPV. Black curve is the ground truth $\mathbb{E}[Y^a]$.

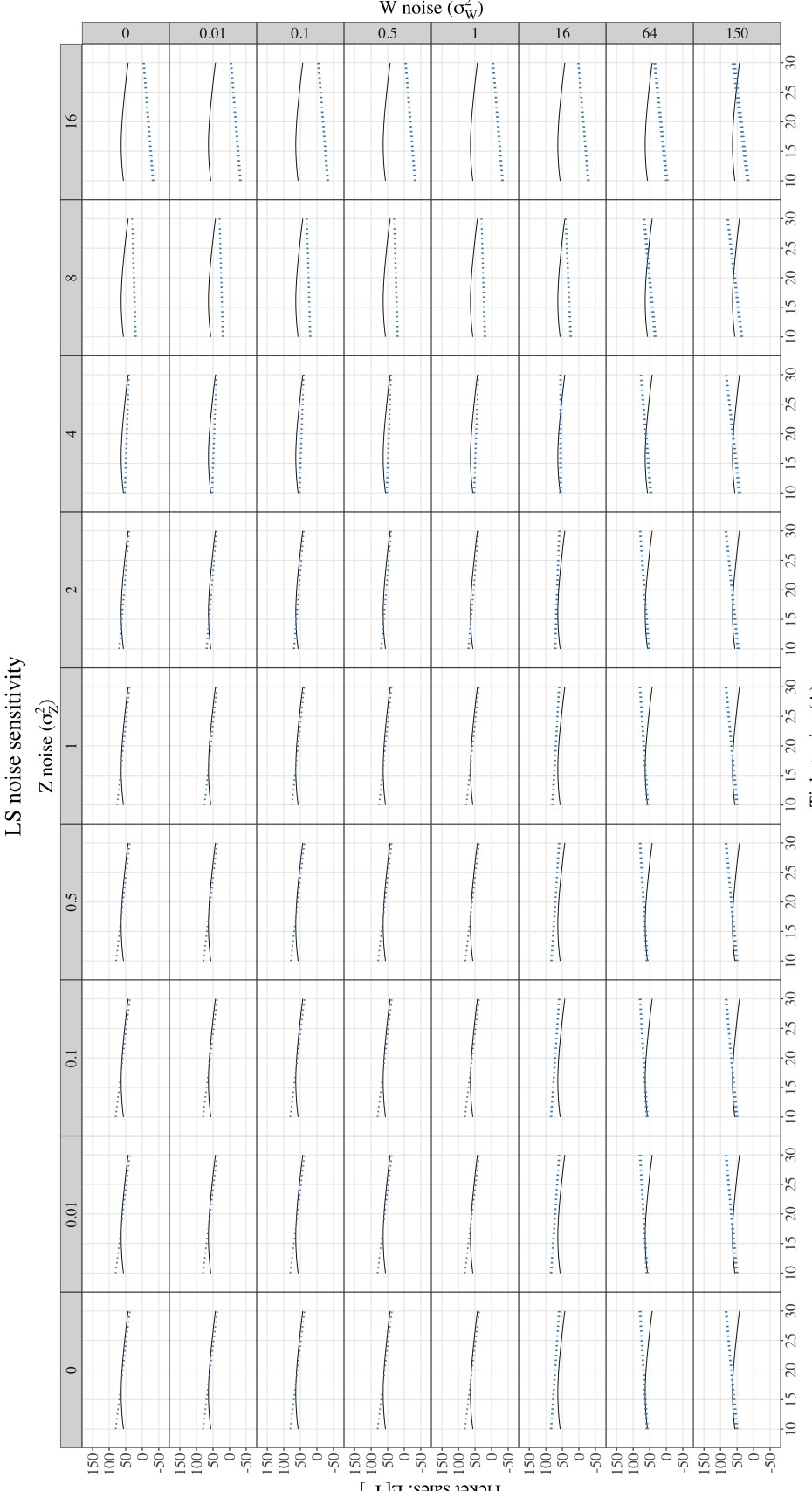

Figure S14: Predicted potential outcome curves for 20 replicates of Least Squares. Black curve is the ground truth $\mathbb{E}[Y^a]$.

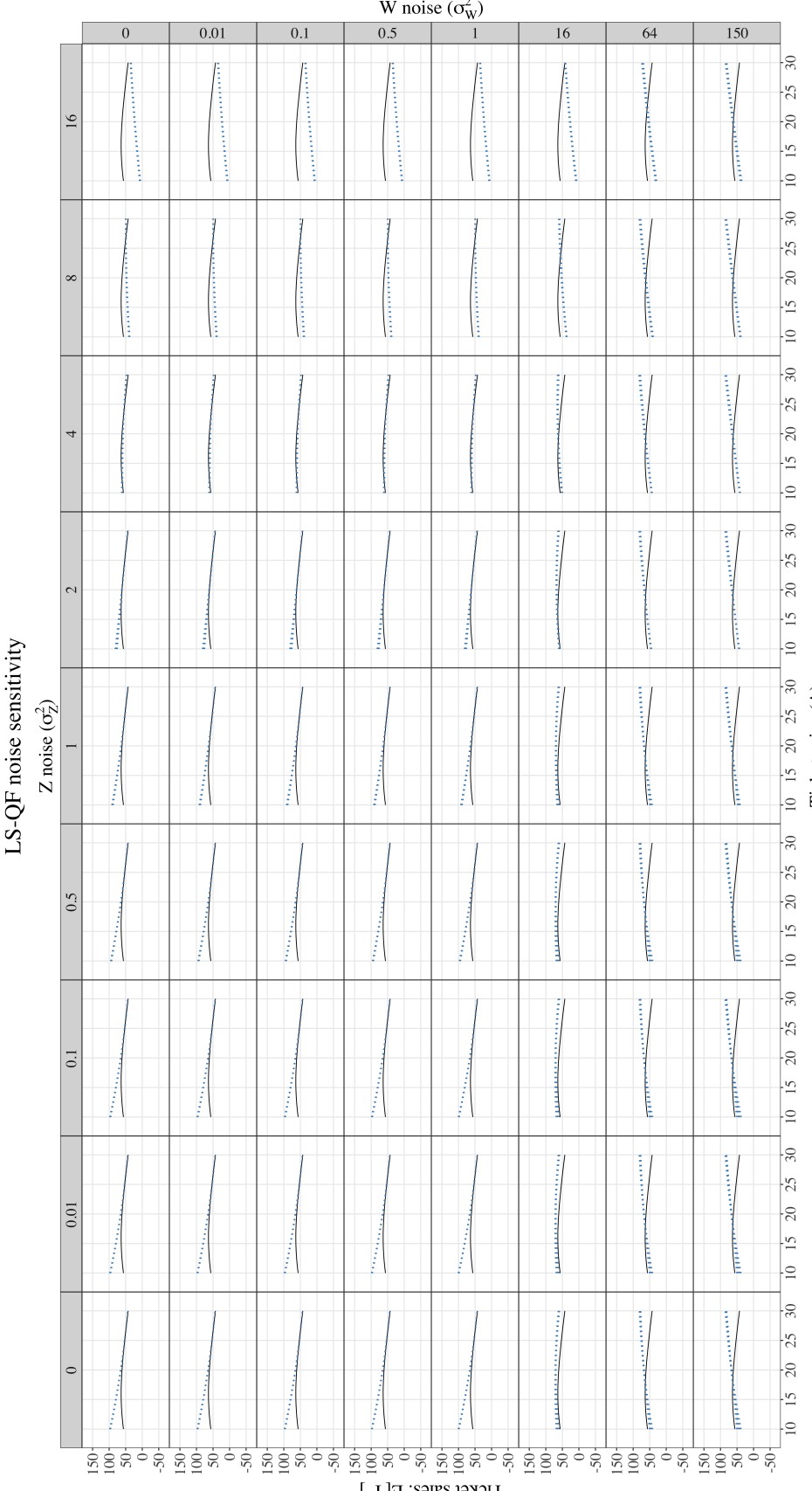

Figure S15: Predicted potential outcome curves for 20 replicates of LS-QF. Black curve is the ground truth $\mathbb{E}[Y^a]$.

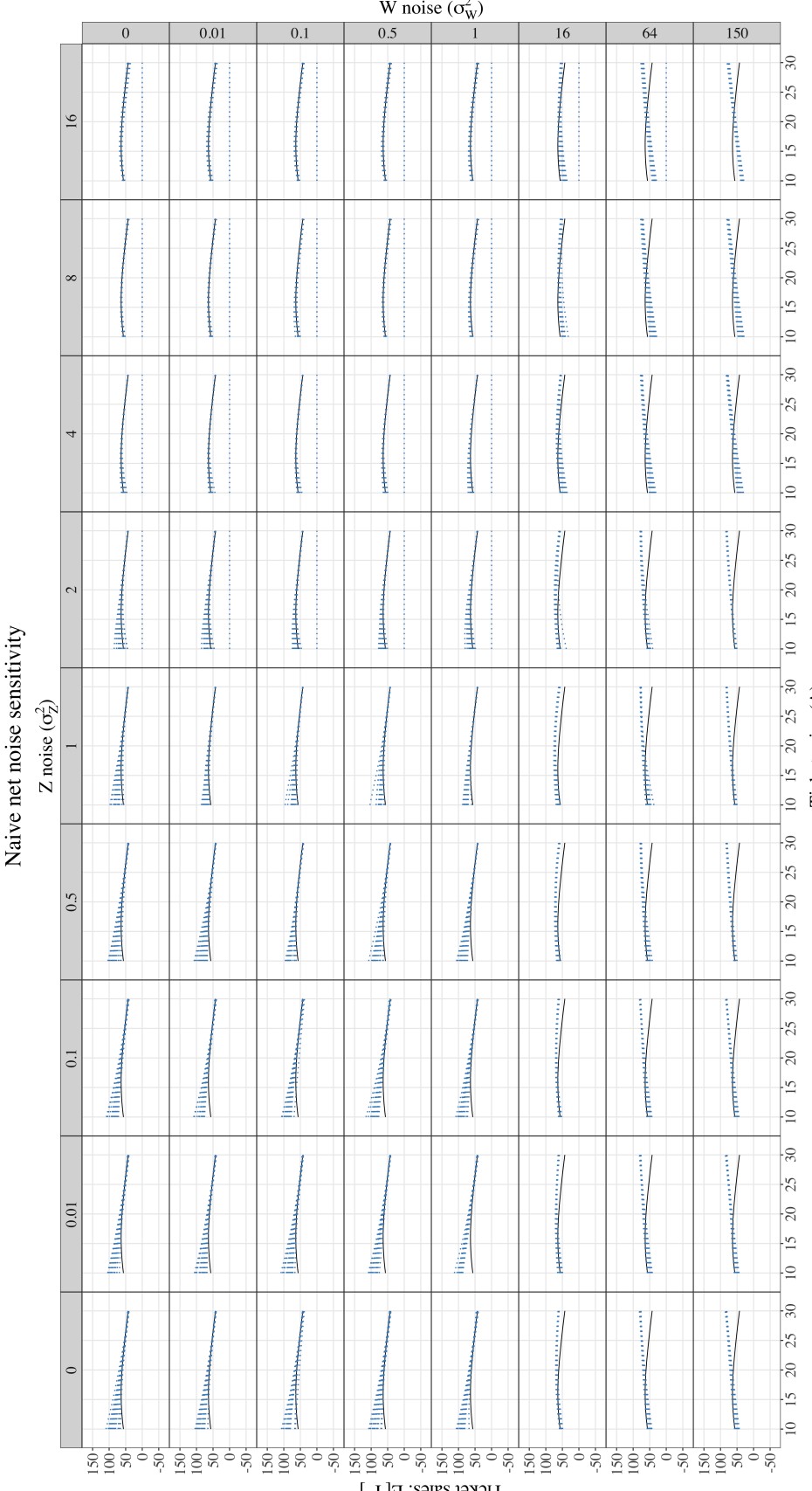

Figure S16: Predicted potential outcome curves for 20 replicates of Naive Net. Black curve is the ground truth $\mathbb{E}[Y^a]$.

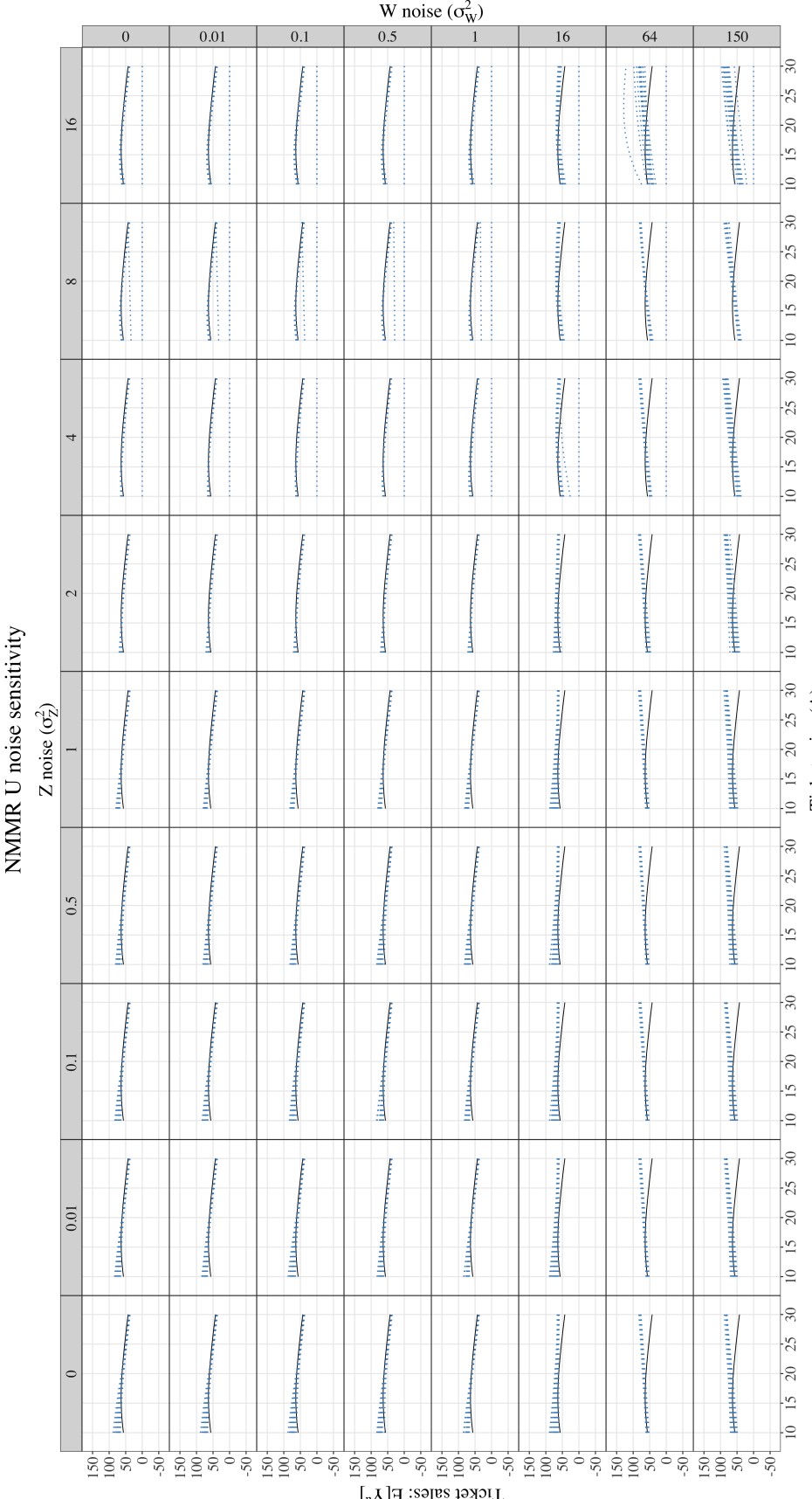

Figure S17: Predicted potential outcome curves for 20 replicates of NMMR-U. Black curve is the ground truth $\mathbb{E}[Y^a]$.

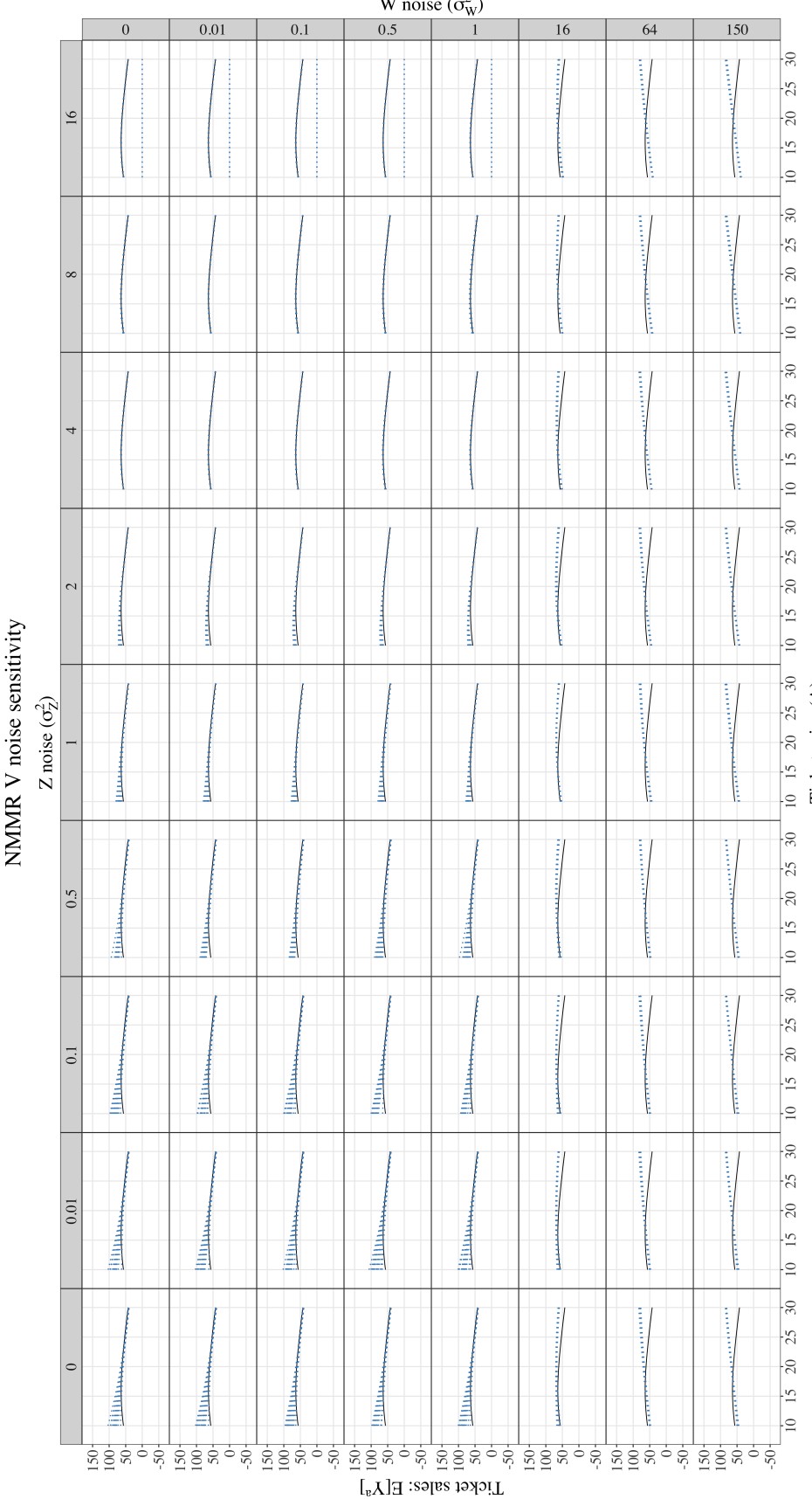

Figure S18: Predicted potential outcome curves for 20 replicates of NMMR-V. Black curve is the ground truth $\mathbb{E}[Y^a]$.