# OpenReview forum: "Deep Learning Methods for Proximal Inference via Maximum Moment Restriction"
_NeurIPS.cc/2022/Conference — NeurIPS 2022 Accept_

### Official Review · Reviewer_ujm9 · 2022-07-07

**Rating:** 6
**Confidence:** 5
**Soundness:** 4 excellent
**Presentation:** 3 good
**Contribution:** 2 fair

**Summary:**

The paper proposes the extension of PMMR, called NMMR-V and NMMR-U, which learn the bridge function in the proximal inference represented by neural net models. Although NMMR-V minimizes the same empirical loss as PMMR, which can be understood as the V-statistic of population loss $R(h)$,  NMMR-U minimizes U-statistics of $R(h)$.  The paper provides the theoretical results and demonstrates the superior performance over existing methods.

**Questions:**

- Why the function $g$ is assumed to be RKHS?  I wonder whether we can learn $g$ by maximizing the loss $R_U(h)$. A similar trick is used for the IV setting in the following work.

Deep Generalized Method of Moments for Instrumental Variable Analysis
Andrew Bennett, Nathan Kallus, Tobias Schnabel

- It seems that convolutional NN is used for NMMP methods in dSprite setting, but not for the DFPV method. How does DFPV perform when it is run with the CNN structure?



**Limitations:**

I think the adequately addressed the limitations and potential negative societal impact of their work.

**Strengths And Weaknesses:**

Strength:
- The extension of PMMR to NMMR is novel though it seems slightly straightforward.
- The analysis of consistent results seems novel and important to me. This might be of independent interest since it can be used in the setting where the U-statistic loss is used, such as preference learning.

Weakness:
- Assuming "discriminator" function $g$ to be RKHS seems slightly restricting to me. When the data becomes high-dimensional, it is known that kernels based on Euclid norm can be highly sensitive to the noise and degenerate the performance.
- It seems that the different network structures are used in the experiments, and it might be difficult to assess whether the NMMR method really contributes to the performance.
- The presentation can be improved. For example, it repeats the long inline equations in the text, which makes me difficult to go through the paper. Also, in the Demand design setting, it reports the performance of methods that uses Z information in the test time, which is not supposed to be available in the proximal setting. This might mislead the reader, who might think non-causal methods perform better than baseline methods in the proximal method.

---

> ### Author Response · Authors · 2022-08-02
> **Response to Reviewer ujm9 1/2**
>
> Thank you again for your review our of work. Please see our Summary comment posted separately, but we’ve include a point-by-point discussion of your specific comments:
>
> - “Assuming "discriminator" function g to be RKHS seems slightly restricting to me. When the data becomes high-dimensional, it is known that kernels based on Euclid norm can be highly sensitive to the noise and degenerate the performance.”
>     - This was a compromise made in order to improve upon several successful estimators previously presented in the literature, which do require $g$ come from a known RKHS, and thus, naturally, use a fixed kernel, while avoiding some potentially challenging issues that can arise from making $g$ more flexible.
>     - In future work we will use adversarial methods, which would involve estimating $g$.  However, it is well known from the optimization literature, that minimax problems and saddle point methods can have convergence issues even in the ideal case that the function is strongly convex/strongly concave and that stable estimators require careful optimization choices.  In the case of neural network problems, this is exemplified by the difficulty inherent in training GANs.
>     - Using a fixed kernel avoids this additional source of instability, although, perhaps, at the cost of increased efficiency and sensitivity to noise, as the reviewer points out.  For these reasons, we believe methods using fixed kernels are worth considering and we are planning to, in future work, compare their performance to adversarial methods in which we directly estimate $g$.
> - “It seems that the different network structures are used in the experiments, and it might be difficult to assess whether the NMMR method really contributes to the performance.”
>     - Thank for examining our work so carefully, we have updated our description of “naive network” in section 6.1 to correctly reflect our experiments. For both our “naive network” (NN) and NMMR, rather than using a single fixed architecture, we did an architecture search as well as hyperparameter optimization across a shared grid of configurations as described in Supplementary Table S1. This is a stronger proof of NMMR’s improvement over the baseline than fixing a single architecture, since a single architecture may be more well-suited to NMMR than NN or vice versa. Instead, we deliberately chose the best performing member from each method class (within the rather expansive limits of the grid).
>     - Additionally, a desirable property of NMMR is the ease with which an arbitrary network structure can be implemented as well as our demonstration that this tailoring of architectures results in greater performance for NMMR. For instance, on the dSprite experiment, we incorporate a VGG-like CNN into NMMR and observe its excellent performance on the task. We argue that this is a benefit of the approach rather than a mischaracterization.
>     - See below for a discussion and demonstration of incorporating the same CNN into DFPV (a baseline method of comparison), which does not result in an improved performance.
>     - See below also for a discussion of baselines that use $Z$ data and how we have rerun/are running experiments to update them.
> - “The presentation can be improved. For example, it repeats the long inline equations in the text, which makes me difficult to go through the paper.”
>     - We will use the additional page of space in the camera ready version of this manuscript to expand the equations for readability and improve the presentation.
>
> -----Continued Below-----

---

> > ### Author Response · Authors · 2022-08-02
> > **Response to Reviewer ujm9 2/2**
> >
> > - “Also, in the Demand design setting, it reports the performance of methods that uses Z information in the test time, which is not supposed to be available in the proximal setting. This might mislead the reader, who might think non-causal methods perform better than baseline methods in the proximal method.”
> >     - Thank you for this excellent point and we have rerun many experiments to remove $Z$ from test time evaluations. Here is a new Figure showing the direct comparison between the two versions of linear regression: [link](https://imgur.com/j6jNIkZ) and here is a new Figure showing them in context with other methods: [link](https://imgur.com/EYMWKVq)
> >     - Originally, we intentionally included Z in these Least Squares and Naive Nets in order to demonstrate that NMMR nonetheless outperforms them with significantly less access to information about U, the unmeasured confounder. However, in order to be consistent with other methods, we are changing this. We’ve also removed the line “Importantly, Least Squares consistently outperforms all other methods besides NMMR” because after rerunning the experiments, this is not absolutely true and it was not our intention to mislead anybody!
> >     - For the Demand experiment, we repeated all of our experiments with LS and LS-QF removing any Z-term from the regression in order to visualize the difference. See the figures here ([S14](https://imgur.com/Q5Fyod5), [S15](https://imgur.com/Q5Fyod5)) or in the revised Supplement. As expected, LS AW performs worse than LS AWZ, however LS AW still outperforms nearly all of the other “causal” methods with the exception of DFPV on sample sizes 1000 and 5000. It does appear to be quite a strong baseline and we believe it is important to include such naive baselines as least squares in our experiments because they serve as an important benchmark to which we compare the other far more complex methods.
> >     - For the sake of completeness, we have also recreated all of the figures in Supplementary Section C.5 (Demand prediction curves/surfaces) and E (model performance in the presence of varying degrees of noise in Z and W) replacing LS AWZ with LS AW, removing Z. See the Figures towards the end of [this album](https://imgur.com/a/UjEucIN) or in our revised manuscript. We can see from the prediction curves that LS AW and LS AW2 (the quadratic version) is not as capable of learning the curvature in the true structural function as LS was when using Z.
> >     - We have also repeated every experiment with naive neural networks, now without using $Z$ data. This involved redoing the HP search and re-evaluating models the Demand, dSprite, and Demand noise experiments. We observe that NN without $Z$ actually do slightly better. See the side by side comparison: [Demand link](https://imgur.com/SxPnmvv) and [dSprite link](https://imgur.com/Gf3M6gW). This is consistent with our observations in the Demand Noise experiment (see: [link](https://imgur.com/TfqxNtw) or Figure S10).
> > - "Why the function g is assumed to be RKHS? I wonder whether we can learn g by maximizing the loss RU(h). A similar trick is used for the IV setting in the following work. Deep Generalized Method of Moments for Instrumental Variable Analysis Andrew Bennett, Nathan Kallus, Tobias Schnabel.”
> >     - Please refer to our response to the first comment above.  In future work, we plan to take a similar adversarial approach, which would allow $g$ to be drawn from a more flexible function class than an RKHS, in particular, $g$ would also be modeled as a neural network. However, this is beyond the scope of our current work. We’ve also made sure to include a reference to Bennett et al., thank you for making us aware of the work.
> > - “It seems that convolutional NN is used for NMMR methods in dSprite setting, but not for the DFPV method. How does DFPV perform when it is run with the CNN structure?”
> >     - Thank you for suggesting this important experiment. We modified Xu et al.’s DFPV method to incorporate a CNN architecture. We used the same architecture as the CNN in NMMR and did a grid search over learning rates and weight decay penalties using the same grid as we used for NMMR.
> >     - We found that these hyperparameters had an important impact on the performance of the method. We discovered 3 hyperparameter configurations (see Table S6 for details) that performed similarly well and included them in the dSprite figure from our paper directly adjacent to the results for DFPV (without CNN) and NMMR U and V. We performed the dSprite experiment with DFPV+CNN for sample size n=1000. See the new Figure S9 at: [link](https://imgur.com/27AcbtN) or in the updated Supplement Appendix C.11.
> >     - The DFPV+CNN method has scalability issues on larger sample sizes and due to time constraints in the rebuttal period we were not able to fully revise DFPV in order to solve these scalability issues. We also believe that this is a point in favor of NMMR, which we designed to scale more easily than DFPV.

---

### Official Review · Reviewer_nVyc · 2022-07-09

**Rating:** 6
**Confidence:** 4
**Soundness:** 2 fair
**Presentation:** 3 good
**Contribution:** 2 fair

**Summary:**

The authors propose a new estimator for the proximal inference problem. When the treatment is continuous, the object of interest is a function of treatment. The most closely related work appears to be [9, Zhang et al 2020], which studies the related problem of nonparametric instrumental variable regression. Like [9, Zhang et al 2020], the “first stage” is eliminated by introducing a kernel matrix into the loss—operationalizing an insight from [8, Muandet et al 2020]. All that remains is the “second stage” of minimizing the loss, for which the authors propose a neural network. The loss either contains cross terms (V statistic) or omits them (U statistic). The authors demonstrate state of the art performance in simulations. Theoretically, the authors outline the argument for consistency, but some terms in the bound are not analyzed.

**Questions:**

Please address the issues under the heading Quality for an improved score

**Limitations:**

--

**Strengths And Weaknesses:**

Originality

The authors emphasize that their implementation and analysis of the U statistic loss with neural networks departs from previous work. Indeed, the performance in simulations is impressive relative to previous work. However, the algorithmic innovation is not very substantial given that previous works mention the U statistic loss (though without fully developing the results like this work).

The authors also emphasize that the theoretical proof of consistency is original. I agree that a strong guarantee is missing from the previous work on U and V statistic losses for neural networks, and such a result would be original and significant. However, I have some issues with the quality of the result the authors provide.

Quality

I have some substantial concerns about the main result, Theorem 1, and some minor concerns about the subsequent result, Theorem 2.

In Theorem 1, has M_k been defined? Is ||h||^2_2 the empirical norm E_n[h(…)^2]? And what norm is ||…||_{P}. Is this the L2 norm?

The bound of Theorem 1 contains Rademacher complexities. The authors write “we cannot compute the scaling of the Rademacher complexity directly”. Do I understand correctly that we therefore do not know that the bound vanishes as n increases? What worries me is the possibility that the quantities in the bound are may not be the right ones for analysis. In ill posed inverse problems, finding the right quantities for analysis is the heart of the problem. Can the authors provide at least some sketch of why they are the right quantities?

In Theorem 2, are the norms of convergence L2? I am surprised that this result has no mention of ill posedness or properties of the kernels for A,X,Z, which are described in [17, Dikkala et al 2020], [11, Ghassami et al 2022], and [Kallus et al 2021]. Without any further assumptions, are we allowing the rate to be arbitrarily slow for h, e.g. even logarithmic rates? This point warrants discussion.

Clarity

Lines 9-10 describe how prior work has pre-specified kernel functions which are not data-adaptive and which struggle to scale. Doesn’t the proposed method also have pre-specified kernel functions that manifest in the kernel matrix in the loss?

Line 30 says positivity can be empirically verified. This is only partially true; please revise the language.

Line 39 says the groups are mutually exclusive. This is not precisely true; some variables in the graph may be classified as either Z or X, for example. See [5, Tchetgen Tchetgen et al 2020]

In the related work, please include the additional references of [Kallus et al 2021] in the paragraph that mentions doubly robust methods for proximal inference and [Singh 2020] in the paragraph that mentions two-stage procedures.

Line 177 “has a slight amount of bias, but in practice is much less biased…” this is a bit imprecise. Do you mean in simulations?

Line 180 is very nicely written.

Significance

On the one hand, this paper provides convincing simulations that demonstrate that a U statistic loss for neural networks outperforms previous work. On the other hand, the proposed algorithm is a straightforward extension of previous work, and the main theoretical question in this literature remains unanswered. If my main theoretical concerns can be ameliorated, then I will increase the score.

---

> ### Author Response · Authors · 2022-08-02
> **Response to Reviewer nVyc 1/3**
>
> Thank you again for your review our of work. Please see our Summary comment posted separately, but we’ve include a point-by-point discussion of your specific comments:
>
> - “In Theorem 1, has M_k been defined? ”
>     - Thank you for noticing that the definition of $M_k$ was missing.  We have now added it to the statement of the theorem.
> - “Is ||h||^2_2 the empirical norm E_n[h(…)^2]?”
>     - We thank the reviewer for identifying this issue. Our theory was initially done using a penalty on the $L^2$-norm of $h$, while, in practice, we actually used weight decay (written as an $\ell^2$-penalty on the neural network weights, although implemented directly in the Adam optimizer).  We have changed the penalty term to reflect this and have updated the proof.
>     - In future work we plan to use the empirical function norm in our methods and will, thus, include the square of empirical function norm, $\|h\|_{2, \mathcal{P}, n}^2 = \mathrm{E}_n h^2$, as you correctly believed we intended to do, potentially in combination with additional terms for other regularization techniques (as we do here with weight decay).
> - “And what norm is ||…||_{P}. Is this the L2 norm?”
>     - This is the $L^2$-norm with respect to the probability measure $\mathcal{P}$.  We have added a note earlier in the paper at the end of Section 2 to state that all norms are $L^2$ unless otherwise stated and to clarify our notation more generally.
>         - “In what follows, all norms will be $L^2$ with respect to the relevant probability measure, unless otherwise noted. If necessary, we will explicitly denote the norm of $f$ with respect to a probability measure...”
> - “The bound of Theorem 1 contains Rademacher complexities. The authors write “we cannot compute the scaling of the Rademacher complexity directly”. Do I understand correctly that we therefore do not know that the bound vanishes as n increases? What worries me is the possibility that the quantities in the bound may not be the right ones for analysis. In ill posed inverse problems, finding the right quantities for analysis is the heart of the problem. Can the authors provide at least some sketch of why they are the right quantities?”
>     - Our statement that “we cannot compute the scaling of the Rademacher complexity directly” is only meant to indicate that we do not explicitly compute the Rademacher complexity, but, like previous authors including Xu, et al., appeal to results, such as from Bartlett, et al. and Neyshabur, et al., that it should scale like $n^{- \frac{1}{2}}$.
>     - We previously included a discussion in the main text, that we had to cut for space, that noted that even though explicit results are not known for the Rademacher complexity of the product of a fixed function and a neural network, we can instead consider the case in which the entire function is represented by a neural network which would then have a Rademacher complexity that is at least as large as our function.
>     - Since, under the results cited above, this will scale like $n^{- \frac{1}{2}}$, the Rademacher complexity in our bound will go to zero at least that fast, meaning that it is $\mathcal{O} \left( n^{- \frac{1}{2} } \right)$.  Thus, we believe that Rademacher complexity is an appropriate quantity to demonstrate consistency, although, in future work we would consider a framework like that in Dikkala, et al. or Kallus, et. al, which would allow us to better control the rate of convergence.  However, one advantage of using Rademacher complexity rather than these other measures is that it is more familiar to readers and is, in some sense, more intuitive than the alternative tools used in those works.
> - “In Theorem 2, are the norms of convergence L2?”
>     - See above, we have added a note earlier in the paper at the end of Section 2 that all norms are $L^2$ unless otherwise stated.
>
> -----Continued Below-----

---

> > ### Author Response · Authors · 2022-08-02
> > **Response to Reviewer nVyc 2/3**
> >
> > - “I am surprised that this result has no mention of ill posedness or properties of the kernels for A,X,Z, which are described in [17, Dikkala et al 2020], [11, Ghassami et al 2022], and [Kallus et al 2021]. Without any further assumptions, are we allowing the rate to be arbitrarily slow for h, e.g. even logarithmic rates? This point warrants discussion.”
> >     - Our methods do not provide explicit rates in terms of standard norms, therefore our results do not account for the degree of ill-posedness, since there is not a clear relationship between the metric induced by the kernel, $k$, on $L^2_\mathcal{AXZ}$ and the standard $L^2$ metrics on either $L^2_\mathcal{A, W, X}$ or $L^2_\mathcal{A, X, Z}$,  However under Assumptions 3 and 4 together if $\tau = \sup_{ h \in \mathcal{H} } \| h - h^* \|_2 \| \mathrm{E} \left[ h - h^* \middle| A, Z, X \right] \|_2^{-1}$ is finite, the convergence rate in Theorem 2 should be the same as the convergence rate in Theorem 1 (under the standard $L^2$-norm), up to a factor of $\tau$.  We have added a short discussion in the text starting at line 234 (which was all we could fit) regarding this point.
> >     - “We can also compare the convergence of the estimated bridge function to that of its projection onto $L^2_\mathcal{AXZ}$. Prior literature has focused on a measure of ``ill-posedness'' $\tau = \sup_{h \in \mathcal{H}} \| h - h^* \|_2 \| \mathrm{E} \left[ h - h^* \middle| A, Z, X \right]_2^{-1}$. If $\tau$ is finite, then the rate of convergence of the estimated bridge function will be at worst $\tau$ times that of its projection (it will be slower by a factor of $\tau$). This will be the case whether we measure convergence using $\| \mathrm{E} \left[ h - h^* \middle| A, Z, X \right]_2$ or the metric induced by $k$."
> >     - Our results do guarantee that the quadratic form $\mathrm{E} \left[ \left( h - h^* \right)^t K \left( h - h^* \right) \right]$ converges say as fast as $n^{ - \frac{1}{2} }$, this corresponds to $n^{- \frac{1}{4} }$ convergence under the metric induced by the kernel, $d_k$, where $d_k^2 \left( h^*, \hat{h} \right) = \mathrm{E}\left[ \left( h - h^* \right)^t K \left( h - h^* \right) \right]$.  However, due to the structure of the kernel, this doesn’t give us an explicit guarantee on the rate of convergence of $\| h - h^* \|_2$, only that $\mathrm{E} \left[ \hat{h} \middle| A, X, Z \right] \to \mathrm{E} \left[ h^* \middle| A, X, Z \right]$ approximately like $n^{ - \frac{1}{4} }$, but not uniformly, due to the nature of the kernel. We have also ensured that we’ve cited these papers in our Related Works Section 3.
> > - "Lines 9-10 describe how prior work has pre-specified kernel functions which are not data-adaptive and which struggle to scale. Doesn’t the proposed method also have pre-specified kernel functions that manifest in the kernel matrix in the loss?”
> >     - The kernel is used here only in an auxiliary capacity to help estimate the bridge function, which is modeled as a scalable neural net.  While we might gain efficiency by allowing the kernel to take a more flexible form, it would not change the way in which the bridge function is modeled, nor its data-adaptivity.
> >     - In future work we plan to replace the kernel with a flexible adversary modeled by a neural network and will compare the performance of fixed kernel vs. adversarial methods, however, this is beyond the scope of our current submission.
> > - “Line 30 says positivity can be empirically verified. This is only partially true; please revise the language.”
> >     - We have revised the language of the paper in Section 1 to reflect this important point. Thank you for making this distinction. The line now reads:
> >         - “The assumptions are typically unverifiable for continuous data.”
> > - "Line 39 says the groups are mutually exclusive. This is not precisely true; some variables in the graph may be classified as either Z or X, for example. See [5, Tchetgen Tchetgen et al 2020].”
> >     - Thank you bringing up this point. When we wrote the sentence, we intended it as the groups must be mutually exclusive after categorizing the variables. We see now how it can be read another way. If groups must be mutually exclusive *before* categorizing, then there is only one correct partition. This is, in general, not true, as you pointed out.
> >     - Hopefully how the text reads now will prevent any miscommunication. Please let us know if you have a better wording in mind:
> >         - Section 1 “Proximal inference requires categorizing the measured covariates into three groups: treatment-inducing proxy variables $Z$, outcome-inducing proxy variables $W$, and “backdoor” variables $X$ that affect both $A$ and $Y$ (i.e. typical confounders)"
> >
> > ----Continued Below----

---

> > > ### Author Response · Authors · 2022-08-02
> > > **Response to Reviewer nVyc 3/3**
> > >
> > > - “In the related work, please include the additional references of [Kallus et al 2021] in the paragraph that mentions doubly robust methods for proximal inference and [Singh 2020] in the paragraph that mentions two-stage procedures.”
> > >     - We’ve updated the text to include these citations. It now reads in Section 3:
> > >         - “Singh and colleagues also consider two stage kernel models in the IV setting and RKHS techniques for proximal inference”
> > >         - “Cui et al. introduced an Inverse Probability Weighted (IPW)-bridge function and corresponding IPW and Doubly Robust (DR) estimators as well as their influence functions for the case of binary treatment, which were further extended in Ghassami et al. and also explored by Kallus et al., who provide alternative identification assumptions and provide results for general treatments.”
> > > - “Line 177 “has a slight amount of bias, but in practice is much less biased…” this is a bit imprecise. Do you mean in simulations?”
> > >     - Thank you for catching this — indeed we meant in simulations. The text in Section 4 now reads:
> > >         - “In practice, $\hat{R}_{k, U, \lambda, n}(h)$ is slightly biased, but, in simulations, is much less biased...”
> > > - “Line 180 is very nicely written.”
> > >     - Thank you!

---

> > > > ### Comment · Reviewer_nVyc · 2022-08-09
> > > > **Thank you for the responses**
> > > >
> > > > Thank you for the thorough responses to my review. Many items have been successfully addressed.
> > > >
> > > > The item that remains for debate it how rigorous Theorem 1 really is. Here is my updated assessment based on the updated discussion.
> > > >
> > > > On Rademacher complexities
> > > >
> > > > i. The Rademacher complexities over these function spaces remain without formal bounds.
> > > >
> > > > ii. Obtaining those formal bounds would be involved, but probably doable with fast rates.
> > > >
> > > > iii.Those formal bounds would probably require some restrictions on the kernels for A,X,Z.
> > > >
> > > > iv. papers by Xu et al. also leave the Rademacher complexities without formal bounds.
> > > >
> > > > On ill posedness
> > > >
> > > > i. The main bound is in a non-standard metric, which is why ill posedness does not arise in main bound.
> > > >
> > > > ii. Convergence in this metric implies convergence for projections of the bridge at half the rate.
> > > >
> > > > iii. The rate for the bridge is the rate for the bridge projections times the measure of ill posedness.
> > > >
> > > > So I now understand how the Theorem 1 rate would eventually translate into rates in more familiar metrics, and at which step ill posedness arises. The paper is better after clarifying these points.
> > > >
> > > > I also see that one step in the argument is plausible yet missing. In the reviewer discussion period, I will reconsider the score.

---

> > > > > ### Author Response · Authors · 2022-08-09
> > > > > **Thank you for your reconsideration and a few questions**
> > > > >
> > > > > Hi Reviewer nVyc,
> > > > >
> > > > > Thank you so much for considering our revised version of our manuscript. Could you elaborate on the plausible but missing step? We will endeavor to improve our results during the upcoming weeks. Additionally, do you have any recommendations/references for bounds on Rademacher complexities?
> > > > >
> > > > > We'd really like to improve these results, even if that must stretch into future work, because we agree that having hard bounds is important
> > > > >
> > > > > Thanks again,
> > > > >
> > > > > The Authors of Paper 11442

---

> > > > > > ### Comment · Reviewer_nVyc · 2022-08-09
> > > > > > **Clarification**
> > > > > >
> > > > > > To clarify, "one step in the argument is plausible yet missing" refers to "obtaining those formal bounds [on Rademacher complexities] would be involved, but probably doable with fast rates".
> > > > > >
> > > > > > I will see if I can find some references whose combination might give the desired result.
> > > > > >
> > > > > > Thanks for your hard work on this.

---

> ### Author Response · Authors · 2022-08-08
> **Final Check In**
>
> Hi Reviewer nVyc,
>
> We just wanted to check in and ask if you had any follow up thoughts on our response to your comments. The author-reviewer discussion window is closing in about 24 hours and we'd love to clarify any remaining questions. Either way, thank you again for your review of our submission!
>
> Best,
>
> The Authors of Paper 11442

---

### Official Review · Reviewer_wpTs · 2022-07-14

**Rating:** 8
**Confidence:** 4
**Soundness:** 3 good
**Presentation:** 3 good
**Contribution:** 3 good

**Summary:**

The authors introduced flexible and scalable deep learning methods to estimate causal effects in the presence of unmeasured confounding using proximal inference.

**Questions:**

* The authors proposed two methods, U-statistic and V-statistic, for proximal inference. The authors should comment on the pros and cons of each method. For example, in simulations, is there a reason why we would expect V outperforms in the former experiment and U outperforms in the latter one?

* Recently there was inverse probability weighting and doubly robust estimators proposed for proximal inference (Cui et al, 2020). (Kallus et al, 2021; Ghassami et al, 2021) then considered kernel methods for doubly robust estimators. Would the proposed deep learning approach extend to inverse probability weighting and doubly robust estimators? Some comments on this would make the proposed approach more general.

* The authors provide a consistency result for their deep learning approach. What kind of convergence rate results such as that of generalized method of moments (Dikkala et al, 2020) could be expected?

**Limitations:**

See weakness and questions.

**Strengths And Weaknesses:**

Strengths: The authors proposed a new deep learning method for proximal inference: neural maximum moment restriction; the authors show desirable numerical performance of their methods; the authors provide a consistency result as a theoretical justification.
Weakness: The authors could give some practical guidance on how to choose their proposed two methods; the authors could potentially strengthen their theoretical results.

---

> ### Author Response · Authors · 2022-08-02
> **Response to Reviewer wpTS**
>
> Thank you again for your review our of work. Please see our Summary comment posted separately, but we’ve include a point-by-point discussion of your specific comments:
>
> - “The authors proposed two methods, U-statistic and V-statistic, for proximal inference. The authors should comment on the pros and cons of each method. For example, in simulations, is there a reason why we would expect V outperforms in the former experiment and U outperforms in the latter one?”
>     - Why the V statistic loss outperforms the U statistic loss in the demand experiment, but is slightly worse in the dSprite experiment, is a very interesting question and one that we believe would make for an excellent follow up investigation. We hypothesize that while superficially the sample sizes in both experiments (i.e. the demand vs. dSprite experiment) are the same, they in truth do not represent and apples-to-apples comparison as they represent functions with potentially very different properties. Thus, while the U-statistic is unbiased and so we would expect it to have lower MSE asymptotically, it may be the case the we have not yet observed a larger sample size for the U-statistic to dominate the V.
>     - We’ve updated Section 6.2 to include a more precise discussion of the pros and cons of each method in our simulation experiments.
>         - “In Figure S10, we can see that NMMR-V is more robust to noised proxies than NMMR-U. This could derive from the fact that the U-statistic is an unbiased, but higher variance, estimator than the V-statistic, so when the proxies are less reliable, the estimate of the risk function $R_k(h)$ is correspondingly less stable.”
> - “Recently there was inverse probability weighting and doubly robust estimators proposed for proximal inference (Cui et al, 2020). (Kallus et al, 2021; Ghassami et al, 2021) then considered kernel methods for doubly robust estimators. Would the proposed deep learning approach extend to inverse probability weighting and doubly robust estimators? Some comments on this would make the proposed approach more general.”
>     - When we initially wrote this paper we were unaware of the work on IPW and doubly robust proximal estimators, which utilize an additional IPW bridge function in addition to the original regression bridge function; we now include a discussion of these previous papers in the Related Work in Section 3, beginning at line 117.
>         - “Cui et al. introduced an Inverse Probability Weighted (IPW)-bridge function and corresponding IPW and Doubly Robust (DR) estimators as well as their influence functions for the case of binary treatment, which were further extended in Ghassami et al. and also explored by Kallus et al., who provide alternative identification assumptions and provide results for general treatments.”
>     - The theory extends naturally to these settings and, so, we believe that our methods could be extended to encompass these models and hope to do so in future work, but this is beyond the scope of our current submission which focuses on the original, regression-only case.
> - “The authors provide a consistency result for their deep learning approach. What kind of convergence rate results such as that of generalized method of moments (Dikkala et al, 2020) could be expected?”
>     - Recent papers including Dikkala, et al., 2020 and Kallus, et al. 2021, use a different approach to their consistency proofs that relies on localized Rademacher complexity and the corresponding critical radius as opposed to using global Rademacher complexity, as we do.  Additionally, because of differences in their estimators, they are able to state their consistency results in terms of $\| \mathrm{E} \left[ h(A, W, X) - h^*(A, W, X) \middle| A, X, Z \right] \|_2$, and, under additional assumptions, sometimes $\| h(A, W, X) - h^*(A, W, X) \|_2$ directly, while our consistency result uses the quadratic form $\mathrm{E}\left[ \left( h - h^* \right)^t K \left( h - h^* \right) \right]$.
>     - We provide our convergence results in terms of the metric induced by the kernel, $k$.  Specifically, $d_k^2 \left( h^*, \hat{h} \right) = \mathrm{E}\left[ \left( h - h^* \right)^t K \left( h - h^* \right) \right]$, which we show is a valid metric on $L^2_\mathcal{AXZ}$, but differs from the standard metric in that it is not isotropic and tends to value some directions over others.  Due to the fact that many kernels, in particular exponential kernels, have a spectrum that is not bounded away from zero, it is difficult to concretely restate our result in terms of the norm. Thus, we cannot directly provide a convergence rate for the standard $L^2_\mathcal{AXZ}$-norm, although we expect that the quadratic form will converge at rate-$n^\frac{1}{2}$, which implies an $n^\frac{1}{4}$ rate for the bridge function with respect to the metric $d_k$ (although this will not be uniform over all directions).

---

> ### Author Response · Authors · 2022-08-08
> **Final Check In**
>
> Hi Reviewer wpTs,
>
> We just wanted to check in and ask if you had any follow up thoughts on our response to your comments. The author-reviewer discussion window is closing in about 24 hours and we'd love to clarify any remaining questions. Either way, thank you again for your review of our submission!
>
> Best,
>
> The Authors of Paper 11442

---

### Author Response · Authors · 2022-08-02
**A Summary of Our Strengths and Improvements 1/2**

**A Summary of Our Strengths and Improvements**

We thank the reviewers for their time spent considering our paper. To help facilitate additional discussion we’ve summarized the major points across all 3 reviews and a summary of our response. We’ve also included a point-by-point reply under each of the respective reviews. We have revised our manuscript and supplement, but, where possible, we have including direct quotes or [links](https://imgur.com/a/UjEucIN) to updated figures to facilitate an easy review of the changes we’ve made. After reading our responses, we ask the each reviewer to consider revising their scores if we adequately addressed their points.

**Strengths:**

- **Performance:** NMMR has state of the art performance on the Demand and dSprite experiments. Reviewer wpTs noted “the authors show desirable numerical performance of their methods” and Reviewer nVyc wrote “the authors demonstrate state of the art performance in simulations” Figures 2 and 3 show that NMMR is a significant improvement on previous methods.
- **Novelty**: All reviewers noted that NMMR is novel and that our method represents a new approach to an important problem in proximal inference with several key improvements.
- **Scalability:** We also scaled both experiments past what was previously considered in the literature. Reviewer wpTs remarked “The authors introduced flexible and scalable deep learning methods to estimate causal effects in the presence of unmeasured confounding using proximal inference.” Previous work considered 1K and 5K data points on both benchmarks. We trained NMMR and existing methods on up to 50,000 data points in the Demand experiment and 7,500 data points in the dSprite experiment. KPV and PMMR could not scale to 50,000 points in our experiments, while NMMR is written as a batched method that easily scales with the training data.
- **Theoretical Guarantees:** Every reviewer commented on the importance of the new theoretical guarantees for NMMR presented in our work. Reviewer ujm9 noted “The analysis of consistent results seems novel and important to me. This might be of independent interest since it can be used in the setting where the U-statistic loss is used, such as preference learning,” while Reviewer wpTs stated “the authors provide a consistency result as a theoretical justification.” We appreciate the insight of Reviewer nVyc: “I agree that a strong guarantee is missing from the previous work on U and V statistic losses for neural networks, and such a result would be original and significant” and have worked to address their points of potential improvement for our proof. Please see the “Improvements to Theory” point below and our reply to Reviewer nVyc.

-----Continued in the comment below-----

---

> ### Author Response · Authors · 2022-08-02
> **A Summary of Our Strengths and Improvements 2/2**
>
> **Weakness and summary of our improvements**
> - **Improvements to Theory:** We have revamped our Section 5 to have new proofs that work with weight decay rather than the function norm. Reviewer wpTs wrote “the authors could potentially strengthen their theoretical result,” while Reviewer nVyc commented “the main theoretical question in this literature remains unanswered.” We’ve included detailed responses to Reviewer nVyc’s questions about our theory and strengthened our theoretical result to closely match our empirical practice. Our new statement of Theorem 1 is too long to include below, but please see Section 5 and Appendix A.
> - **Relationship to existing literature**. The reviewers commented that NMMR is a “straightforward” advance to existing methods. In particular, Reviewer ujm9 noted “NMMR is novel though it seems slightly straightforward” while Reviewer nVyc wrote “algorithmic innovation is not very substantial given that previous works mention the U statistic loss (though without fully developing the results like this work).”  We believe that our work represents a natural, though non-trivial, extension of previous methods, and improves upon existing methods in several important ways. Though the previous work of Zhang et al. offered a theoretical treatment of the U statistic loss in an IV setting, there was no empirical evaluation of the U statistic loss and it had not been proposed for use in proximal settings such as we consider. Additionally, our work is the first to considered incorporating task specific inductive biases (e.g. CNNs in the dSprite experiment). We thus believe our work is a novel contribution since it extends and combines several ideas from the literature including NNs for proximal inference, MMR, and V/U-statistics, and provides compelling empirical evidence of their efficacy on several important benchmark tasks.
> - **Comparisons across methods:** We have conducted extensive additional analysis to address several questions raised by the reviewers:
>     - **NMMR vs DFPV:** We implemented a version of DFPV with the same CNN architecture we used in NMMR in place of the MLPs used in the published version of DFPV. We found that after a thorough hyperparameter search of DFPV with CNNs, NMMR still outperformed DFPV with CNNs. See here: [link](https://imgur.com/27AcbtN) or Appendix C.11 in the updated supplement.
>     - **Access to $Z$:** Reviewer ujm9 noted that the LS & NN (MLP) methods we evaluated in the paper had access to the $Z$ variable in the Demand experiment, while in practice this would not be the case for proximal inference. We have rerun LS with only access to $A$ and $W$ and have remade 6 Demand-related figures in our paper that reference the original version of LS. We have also rerun NN on Demand and dSprite. See here: [link](https://imgur.com/a/UjEucIN) or the updated text.
>     - **NMMR vs MLP:** We originally performed a model architecture and hyperparameter search over the same grid for NMMR and MLP and optimized models based on held-out U or V statistic loss. The MLP models we compared NMMR to in the paper may not have the exact same architecture, but they are the best MLPs could do on the same grid of hyperparameters. Combined with rerunning the Naive Networks (see above), we have thoroughly demonstrated the lift gained by using NMMR.
> - **Presentation:** We will utilize the extra page for a potential camera ready version to expand our inline math equations and make additional edits to improve the readability of our manuscript.
>
> Thank you again for your (re)consideration of our work!

---

### Author Response · Authors · 2022-08-05
**Additional Feedback Requested**

Hi Reviewers wpTs, nVyc, and ujm9,

We're excited to discuss our paper with you and answer any additional questions you may have! Your comments have already improved our work quite a bit. We've revamped our theoretical section and run many more experiments to investigate the questions raised in your reviews. Our Summary comment is a quick overview of our major changes and we've added individual comments to each of your reviews. The reviewer-author discussion period is nearly over and we'd love to know if we've adequately addressed your comments on our paper.

Best,
The Authors of Paper 11442

---

### Meta-Review · Area_Chair_Vh9w · 2022-08-29

**Recommendation:** Accept
**Confidence:** Certain

**Metareview:**

Reviewers agreed that the paper proposes a new and valuable method for proximal causal inference with a solid theoretical analysis. The reviewers pointed out several ways in which the theory might be improved, some of which have already been undertaken by the authors, while some remain open. A potential drawback is that the authors were originally unaware of highly related work (especially Cui et al. 2020 and Kallus et al. 2021); while the authors have now addressed this work, I suggest that in the final version they add further details about the differences between their results and those of the above papers.
Having said all that, we all view the contribution positively and believe its merits outweigh its drawbacks .

**Award:**

No

---

### Decision · Program_Chairs · 2022-09-14

Accept